# Accurate Detection of Urothelial Bladder Cancer Using Targeted Deep Sequencing of Urine DNA

**DOI:** 10.3390/cancers15102868

**Published:** 2023-05-22

**Authors:** Dongin Lee, Wookjae Lee, Hwang-Phill Kim, Myong Kim, Hyun Kyu Ahn, Duhee Bang, Kwang Hyun Kim

**Affiliations:** 1Department of Chemistry, Yonsei University, Seoul 03722, Republic of Korea; egun1229@gmail.com; 2IMBdx, Seoul 08506, Republic of Korea; 3Department of Urology, Ewha Womans University Seoul Hospital, Seoul 07804, Republic of Korea

**Keywords:** bladder cancer, non-invasive cancer detection, urine tumor DNA, target deep sequencing, early cancer detection

## Abstract

**Simple Summary:**

Accurate, non-invasive methods that can diagnose bladder cancer (BC) are needed to detect BC in its early stages and reduce the burden on patients with hematuria. We developed uAL100, a method that accurately diagnoses BC by identifying tumor DNA in urine samples using targeted deep sequencing. We demonstrate that uAL100 accurately detects putative BC driver mutations in the urine without information about the tumor.

**Abstract:**

Patients with hematuria are commonly given an invasive cystoscopy test to detect bladder cancer (BC). To avoid the risks associated with cystoscopy, several urine-based methods for BC detection have been developed, the most prominent of which is the deep sequencing of urine DNA. However, the current methods for urine-based BC detection have significant levels of false-positive signals. In this study, we report on uAL100, a method to precisely detect BC tumor DNA in the urine without tumor samples. Using urine samples from 43 patients with BC and 21 healthy donors, uAL100 detected BC with 83.7% sensitivity and 100% specificity. The mutations identified in the urine DNA by uAL100 for BC detection were highly associated with BC tumorigenesis and progression. We suggest that uAL100 has improved accuracy compared to other urine-based methods for early BC detection and can reduce unnecessary cystoscopy tests for patients with hematuria.

## 1. Introduction

Hematuria is a prominent symptom in the early stages of bladder cancer (BC) and is present in 80–90% of patients with BC [1]. However, only 10–22% of patients with hematuria are eventually diagnosed with BC [2,3]. Cystoscopy, the gold-standard method for detecting BC, is an invasive and costly procedure that carries risks of complication, and its accuracy is highly dependent on the skills of the operator. Therefore, there is a need for non-invasive tests to detect BC with reliable accuracy.

Several non-invasive urine-based tests have been developed to complement invasive cystoscopy for BC diagnosis utilizing materials in the urine (e.g., urine cytology, BTA, UroVysion, uCyt+, and NMP22) [4,5,6,7]. However, these tests have poor accuracy in detecting BC, with sensitivities between 56% and 83% and specificities between 64% and 88%. Additionally, these urine-based tests do not provide genotyping information to aid clinical decisions [5,6]. Therefore, there is a need for the development of alternative non-invasive methods with enhanced accuracy and the genetic information of BC.

As an alternative non-invasive method for detecting BC, the targeted deep sequencing of cell-free DNA (cfDNA) in the blood or DNA in the urine (supernatant or cell pellet) has been developed, as these methods can detect genomic mutations at low allele frequencies, enabling the diagnosis and genotyping of BC in the early stages of the disease [8,9,10,11,12]. In particular, the deep sequencing of urine is more advantageous than blood in BC detection since urine contains relatively more enriched tumor-derived materials than blood due to a lower amount of leukocyte-derived materials in patients with BC [12,13]. Additionally, since urine can be sampled entirely non-invasively with a practically unlimited sample volume, urine sampling is more advantageous than blood sampling. Therefore, the targeted deep sequencing of DNA in the supernatant or cell pellet of urine is a promising approach for clinical applications in BC, although the accurate detection of BC is challenging due to the presence of significant levels of false-positive signals (sensitivity: 83–87% and specificity: 85~96%) [10,11,14,15].

In a previous study, we detected circulating tumor DNA in the blood of patients with colorectal cancer using our AlphaLiquid100 target capture panel and its paired analysis pipeline, which we can refer to collectively as AL100 [16]. The analysis pipeline suppresses errors in deep sequencing data for 118 cancer-associated gene regions and detects single-nucleotide variants (SNVs), insertions/deletions (INDELs), copy-number variations (CNVs), and gene fusions with variant allele frequencies (VAFs) as low as 0.1%. The AlphaLiquid100 panel includes genes associated with BC; therefore, we decided to utilize AL100 to accurately detect urine tumor DNA (utDNA) in the urine supernatant for the diagnosis of BC.

In this study, we describe uAL100, a urine-based BC detection method that accurately detects utDNA without matched tumor information. When applied to urine samples from our cohort, uAL100 successfully suppressed technical errors, resulting in a specificity of 100% for BC detection with a sensitivity of 83.7%. We expected that uAL100 would be able to more reliably detect early BC than other urine-based methods, thereby reducing unnecessary tests for patients with hematuria. Additionally, uAL100 is expected to be helpful in making clinical decisions for patients with BC since uAL100 provides genotypes of BC, including mutations that are significant in BC tumorigenesis and progression. Collectively, we anticipate that uAL100 has the potential to greatly improve the efficiency and efficacy of BC detection and treatment. 

## 2. Materials and Methods

### 2.1. Patient Enrollment and Sample Collection

We prospectively collected clinical information, including tissue and urine samples from patients who underwent surgical treatment or intervention for malignant or benign urological disease. This study was approved by the Institutional Review Board of Ewha Woman’s University Seoul Hospital (IRB No. 2019-08-008). All patients provided informed consent for tissue banking and genetic testing. Urine cytology was performed at the time of cystoscopy according to the Paris System [17]. Tissue samples were obtained at the time of surgery. The DNA from tumor tissue was extracted using a QIAamp DNA Minikit (Qiagen, Hilden, Germany). Urine was collected prior to surgery in a urine preservation tube (Norgen Biotek, Thorold, ON, Canada). For the DNA isolation from urine, urine samples were centrifuged at 200× *g* for 10 min, followed by 3000× *g* for 20 min to remove cellular materials. Urine DNA was isolated from the supernatants using a MagMax^TM^ Cell-Free DNA Isolation Kit (Applied Biosystems, Thermo Fisher Scientific, Foster City, CA, USA) or a Quick-DNA^TM^ Urine Kit (Zymo Research, Irvine, CA, USA).

### 2.2. Analysis of DNA from Urine and Bladder Tumors

Next-generation sequencing libraries of urine and tumor DNA were constructed using the IMBdx NGS DNA Library Prep Kit (IMBdx, Seoul, Republic of Korea). For urine DNA, 6–30 ng of DNA was used to construct each DNA library. Hybridization-based target enrichment was performed using the AlphaLiquid^®^ 100 target capture panel (IMBdx, Seoul, Republic of Korea), which was designed to cover the entire exons of 118 cancer-related genes (Appendix A). The target-enriched DNA libraries were sequenced using the Illumina Novaseq 6000 platform (Illumina, San Diego, CA, USA) to create 150 bp paired-end reads, with median sequencing depths of 6094× and 7732× for targeted regions in urine and tumor samples, respectively (Appendix A). A primary variant calling was conducted in the same manner as in our previous study [16].

To detect mutations in the urine’s DNA, cutoffs of VAF ≥ 1.5% and an altered high-quality unique sequence count ≥4 were applied to filter out noise, contamination, and sequencing errors. For CNVs, gene amplifications were determined when the copy number was higher than 4 with a *p*-value < 0.001. Candidate gene fusions were detected by a dual fusion caller composed of GeneFuse and SViCT [18,19]. Mutations identified in urine DNA were used for BC detection if they satisfied at least one of the following criteria (Appendix A): (1) one of the two *TERT* promoter hotspot SNVs, (2) annotated in the OncoKB database as oncogenic or likely oncogenic [20], (3) annotated in the COSMIC database as oncogenic [21], or (4) included in the Cancer Hotspots database (splice acceptor and donor mutations only) [22,23]. These criteria were not applied to the mutational concordance study.

Single-nucleotide polymorphism databases were used at the primary variant calling step to filter out putative germline mutations. Residual putative germline mutations were excluded by the following criteria. For the mutational concordance study, mutations found in both the tumor and urine were considered germline if they both had a VAF between 40% and 60% or higher than 95%. For BC detection, mutations in urine DNA with a VAF between 40% and 60% or higher than 95% were considered germline. These criteria were not applied to the two *TERT* promoter hotspot SNVs, which were reported as somatic mutations.

### 2.3. Concordance Analysis of Tumor and Urine DNA

For the mutational concordance analysis, mutations in the tumor DNA were excluded when the VAF was below 3%. The mutations in the urine DNA used in the concordance analysis were selected before the data were filtered against the cancer mutation databases. SNVs, INDELs, and CNVs in urine and tumor DNA from each of the seven patients with BC were annotated as “both tumor and urine”, “tumor only”, or “urine only”. Pearson’s correlation coefficient (r) was used to assess correlations between mutations in the matched tumor and urine samples. The annotated SNVs and INDELs of all seven patients were then merged, and the ratio of mutations for one given sample type that detected an in-paired sample type (detection ratio) was calculated using the following formula:$$\frac{\mathrm{Number}\;\mathrm{of}\;\mathrm{mutations}\;\mathrm{detected}\;\mathrm{in}\;\mathrm{both}\;\mathrm{sample}\;\mathrm{types}}{\mathrm{Number}\;\mathrm{of}\;\mathrm{mutations}\;\mathrm{detected}\;\mathrm{one}\;\mathrm{given}\;\mathrm{sample}\;\mathrm{type}}\times 100.$$

### 2.4. BC Detection without Tumor Samples

All statistical tests were conducted using the R system version 4. A *T*-test was used to test for differences between the two groups, and ANOVA was used for more than two groups. For BC detection using uAL100 without tumor samples, putative driver mutations were identified in urine DNA using the criteria described in Section 2.2. The mutation with the highest VAF in each patient was selected, and the respective VAFs were then used as the percentage of utDNA in the entire sample of urine DNA. The sensitivity was the proportion of BC cases with BC driver mutations in their urine DNA among the 43 cases tested. The specificity for this was the proportion of the 21 controls that were correctly identified as not having BC driver mutations in their urine DNA. The R library pROC was used to calculate the area under the curve (AUC) levels, and ggplot was used to plot the matched receiver operating characteristic (ROC) curves.

## 3. Results

### 3.1. Patient Characteristics and Study Design

This study included 43 patients with urothelial BC and 21 patients with benign disease. Patients with a benign disease did not have any history of malignant disease and were considered healthy controls without cancer. The patients’ characteristics are summarized in Table 1 and Figure 1a. Of the 43 patients with BC, 36 (83.7%) had non-muscle-invasive BC (NMIBC), and 7 (16.3%) had muscle-invasive BC (MIBC). Urine samples were available from all 43 patients with BC, and tumor samples were available from 7 of the 43 patients with BC. All urine and tumor samples collected from patients with BC and healthy donors were subjected to uAL100. The workflow of uAL100 is depicted in Figure 1b.

### 3.2. Mutational Concordance in Paired Samples of Tumor and Urine DNA

Before assessing the ability of uAL100 to detect BC, we investigated mutational concordance between the matched tumor and urine samples of seven patients with BC to confirm that the mutations detected by uAL100 were derived from tumors. We detected a total of 50 mutations and two gene amplifications in the urine and tumor DNA of the seven patients (Figure 2a; Appendix A). Of the 50 mutations, 37 were shared between paired tumor and urine samples, whereas 13 were detected only in tumor samples or urine samples (tumor only = 4, urine only = 9; Figure 2a and Appendix A). As a result, 90.2% of tumor mutations were also detected in paired urine samples, and 80.4% of urine mutations were also detected in paired tumor samples (Figure 2b). Shared mutations in urine and tumor DNA were detected in six of the seven patients. Only one mutation, TP53 M237I, was detected in the other patient, and this mutation was detected only in the urine DNA. The VAFs of the shared mutations were highly correlated between the tumor and urine samples in five out of the six patients that had shared mutations (Appendix A). Within urine samples, the mutations that were shared with the matched tumor samples had a higher median VAF than those that were not shared with the matched tumor samples (median VAF = 17.18% vs. 2.06%; *p* = 0.025; Figure 2c). Although tumors were available from only seven patients, as most of the patients (six of seven) had NMIBC, our method was reliable regardless of the tumor stage. We concluded that most of the mutations detected in the urine samples by uAL100 were true positives, suggesting that uAL100 accurately detected utDNA in the urine. Nevertheless, to obtain the statistical significance of high mutational concordance between the paired samples, additional sample pairs should be assayed, which is one of our future tasks.

### 3.3. Bladder Cancer Detection without Tumor Samples

Next, we applied uAL100 to call putative BC driver mutations in urine samples from 43 patients with BC and 21 healthy controls to validate the accuracy of uAL100 for the detection of BC (Appendix A). We detected a total of 170 putative BC driver mutations across 46 genes in the urine samples from patients with BC, whereas we did not detect any putative BC driver mutations in the urine samples of healthy controls (Figure 3a and Appendix A). We detected at least one putative BC driver mutation in the urine samples of 36 out of the 43 patients with BC, with a median mutation count of 3.5 per patient. The number of detected mutations increased with the tumor stage (*p* < 0.001; Figure 3b) and tended to be higher in high-grade BCs than in low-grade BCs, although this difference was not statistically significant (Figure 3c). The estimated proportion of utDNA within the urine samples increased with both the tumor stage (*p* < 0.001; Figure 3d) and the tumor grade (*p* < 0.001; Figure 3e). Next, we evaluated the performance of BC detection using the identified mutations. The overall sensitivity and specificity of BC detection based on the mutations identified by uAL100 were 83.7% and 100%, respectively (Figure 3f,g). In a subgroup analysis according to tumor stage, the sensitivity of detection in the subgroups with stage Ta+Tis, T1, and T2–T4 disease was 71.4%, 90.9%, and 85.7%, respectively, with a specificity of 100% in each subgroup (Figure 3g). When subgroups with stages that corresponded to NMIBC were combined, the sensitivity of detection was 83.3%, which is similar to those of MIBC (T2–T4). These results indicate that uAL100 had high sensitivity and 100% specificity for the detection of BC without information about the tumor, even in the early stages of the disease.

### 3.4. Bladder Cancer Genotypes Determined Using uAL100

We analyzed the genotypes of 43 patients with BC based on the somatic mutations detected in urine DNA (Figure 3a). The most frequently mutated gene was *TERT* (60% of cases), and all 26 *TERT* mutations were located at one of two previously reported *TERT* promoter single-nucleotide hotspots (Figure 4a and Appendix A) [24,25,26]. Other frequently mutated genes were *TP53*, *ARID1A*, *PIK3CA*, *ERBB2*, *ERBB3*, *FGFR3*, *KDM6A*, and *TSC1* (>10% of cases), which were observed as BC driver genes in previous studies [10,26]. Additionally, when compared to the TCGA study that investigated BC tumors, the frequency of cases with mutations in each gene detected by uAL100 showed a high correlation with those detected in BC tumors reported in the TCGA study (*R* = 0.91; Figure 4b) [27], which indicated that BC genotyping using uAL100 significantly reflected tumor information. When considering the tumor stage, the most frequently mutated sites in both the NMIBC subgroup and the MIBC subgroup were the *TERT* promoter, *TP53*, *ARID1A*, and *PIK3CA* (Figure 4c,d). The *TERT* promoter and *PIK3CA* were also the most frequently mutated sites regardless of cancer grade (Figure 4e,f); however, *TP53* and *ARID1A* were less mutated in low-grade BC compared with high-grade BC (0% vs. 52.78% and 14.29% vs. 30.56%, respectively), which is consistent with previous studies [26,28].

We next investigated the CNVs identified by uAL100 in the 43 urine samples from patients with BC. We observed nine different gene amplifications across 10 patients (Figure 3a). Six of these nine amplified genes were observed in BCs in previous studies [27,28,29,30]. Each of the three remaining gene amplifications was detected in a different single patient. Among them, *MYCN* amplification was also detected in the matched tumor sample (patient BC009), indicating that *MYCN* amplification in the urine was a true positive result. The other two amplifications in *CDK12* and *MAPK1*, respectively, were considered tumorigenic in other cancer types, suggesting that may also be tumorigenic in BC. In addition to the nine amplifications, we identified an intra-chromosomal translocation on chromosome 4 involving *FGFR3* and *TACC3* in a single patient (patient BC038; Figure 4g). The breakpoint occurred in the intron17 of *FGFR3* and intron10 of *TACC3*, and the resulting fusion gene contained essential domains of both genes for the tumorigenesis of BC [27]. Overall, the mutations identified by uAL100 were related to BC and included genes that played a crucial role in tumorigenesis and in the progression of the disease. These findings indicate the potential of mutations detected by uAL100 in urine DNA as biomarkers for BC. Additionally, the genotyping provided by uAL100 offered a deeper understanding of genetic changes in BC, which could inform future diagnostic and therapeutic approaches.

### 3.5. Information from Peripheral Blood Mononuclear Cells in Urine-Based Bladder Cancer Detection

Peripheral blood mononuclear cells (PBMCs) are required to distinguish between tumor variants and germline and clonal hematopoiesis (CH) variants in cfDNA from plasma. However, in this study, we did not analyze PBMCs, because such an analysis would have caused additional inconvenience and discomfort for the patients. Instead, to remove putative germline mutations, we applied a simple germline filter that removed mutations with a VAF of 40~60% or >95%. This filter removed 21 mutations in urine samples from 15 patients with BC and 5 mutations in urine samples from 3 healthy controls (Appendix A). As a result, all three healthy controls were correctly identified as not having BC, whereas no patients with BC were incorrectly identified as not having BC (Appendix A).

Because hematopoietic cells are not prevalent in urine, DNA in urine contains a relatively low amount of CH variants and a relatively high fraction of tumor DNA compared with cfDNA in the blood [13]. Therefore, we hypothesized that CH variants were unlikely to have VAFs > 1.5% in urine DNA. Previous studies identified CH variants in PBMCs and cfDNA from the blood in >90% of healthy controls [31,32]. By contrast, we did not identify any mutations in the urine samples of healthy controls (Figure 3a). Therefore, we concluded that CH variants were not significant in the analysis of cfDNA in urine. Based on these results, we suggest that PBMCs are not necessary for the detection of BC using urine samples; however, further studies are required with larger numbers of urine samples and matched PBMCs for the precise validation of our hypothesis. 

## 4. Discussion

In this study, we described uAL100 and validated its performance in BC diagnosis. There was a high concordance between the mutations detected by uAL100 in matched urine and tumor samples, although the number of matched pairs was only seven in our cohorts. This indicated that the mutations detected in urine by uAL100 were derived from BC tumors. We then showed that uAL100 had 83.7% sensitivity and 100% specificity in detecting BC without tumor samples, suggesting that our method had a high accuracy regardless of the tumor stage. Previous urine-based methods using deep sequencing for BC detection showed significant levels of false-positive signals with a similar or lower sensitivity compared with our method (Table 2) [10,11]. A genotyping analysis showed that the mutations identified by uAL100 were highly related to BC tumorigenesis. Additionally, we showed that accurate utDNA detection was possible without PBMCs which enables the entire non-invasiveness of uAL100. However, since several mutations were removed by the germline filter in some patients that were diagnosed with BC, it was possible that the germline filter could lead to imperfect clinical decision making based on the mutation profiles that it produced. Therefore, although the germline filter had little effect on the sensitivity of BC detection without tumor samples, information from PBMCs may still be needed to support optimal clinical decisions.

For the clinical utilization of liquid biopsy for bladder cancer diagnosis, factors such as cost-effectiveness, test standardization and diagnostic accuracy should be taken into consideration. In this study, we described the process of sample preparation, DNA extraction and genomic analysis for our test and validated the diagnostic accuracy of uAL100 for bladder cancer. During the validation of uAL100, we showed that uAL100 accurately detected utDNA without matched PBMCs, which enabled an increase in convenience and a reduction in costs. Regarding cost-effectiveness, although sequencing costs have decreased, the cost of sequencing may still be more expensive than conventional urine tests for bladder cancer. Therefore, we do not expect uAL100 to fully replace cystoscopy, which is the standard procedure for the diagnosis of bladder cancer. However, uAL100 has high diagnostic accuracy, and we expect that uAL100 could identify patients who need cystoscopy and reduce the number of unnecessary procedures in patients who may have undergone invasive procedures without our test.

As in our cohort, NMIBC is the most common type of bladder cancer at the time of diagnosis. However, unlike similar clinical and histopathological characteristics of NMIBC tumors, their genomic landscapes are variable, resulting in largely different disease progression and responses to treatment [33,34]. Consequently, about 50–70% of NMIBC patients experience recurrence and progression to MIBC, necessitating continuous cystoscopy and diverse therapeutic interventions [35,36]. Therefore, accurate capture of the genomic landscape of NMIBC, including its tumor cell heterogeneity, is necessary for complete cancer remission in the early stages of the disease. However, the amount of released tumor-derived materials in bodily fluids for NMIBC is small, and therefore, reflecting the integrative information of NMIBC is challenging. To overcome such challenges, we envision that uAL100, with its high accuracy, has the potential to capture the genomic landscape of NMIBC containing information from subclonal or metastatic tumors at low levels. Thus, uAL100 is expected to be helpful in early clinical decisions and lifelong surveillance by preventing excessive cystoscopy and therapeutic interventions, which we need to validate in future studies.

We found that uAL100 performed well at detecting BC without tumor or blood samples; however, our study had several limitations. First, our cohort was small, and further validation of our method with larger samples is needed. Specifically, more tumor and urine sample pairs are needed for analysis to obtain statistical significance for the mutational concordance study since we only analyzed seven sample pairs, which is a small number to provide statistical significance to the results. Second, since the analysis pipeline of uAL100 was originally developed for plasma samples, error suppression may not be sufficient to remove all errors in urine DNA. Therefore, although the current version of error suppression in uAL100 is reliable, an error suppression method specifically for urine samples is needed to improve the accuracy of uAL100. Third, since our target capture panel was not solely composed of genes related to BC, the cost of our method was not fully optimized for BC detection. Furthermore, some known BC-related genes, such as *PLEKHS1*, were not contained in the target capture panel. Consequently, it might be possible to increase the sensitivity for BC detection while reducing the error rate and cost by adjusting the gene panel. Finally, as we mentioned, the surveillance of NMIBC patients is important due to their high recurrence rate. Therefore, we need to validate the surveillance ability of uAL100 to make our method more meaningful in real clinical situations.

## 5. Conclusions

The uAL100 method had 83.7% sensitivity and 100% specificity in the detection of BC tumor DNA in the urine without blood and tumor samples. Mutations identified by uAL100 were highly associated with BC tumorigenesis and progression. The uAL100 method has the potential to improve the accuracy of BC diagnosis and reduce the need for invasive procedures and additional samples.

## Figures and Tables

**Figure 1 cancers-15-02868-f001:**
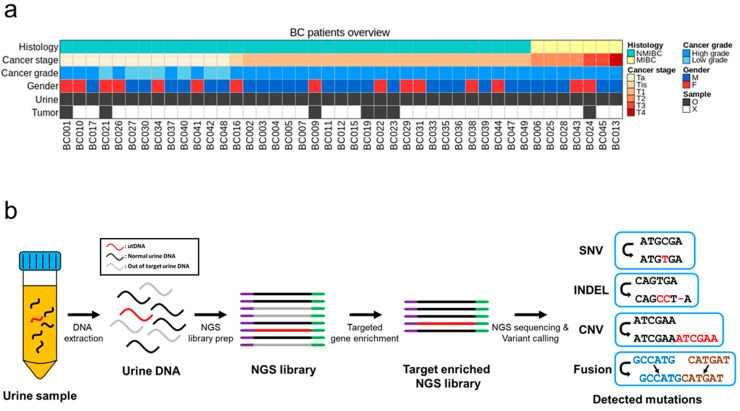
Patient information and study design. (**a**) Information about patients with BC are depicted graphically. Availability of urine and tumor samples for each patient is indicated by the grey color in the corresponding rows. M: male, F: female. (**b**) The scheme of uAL100 is described graphically.

**Figure 2 cancers-15-02868-f002:**
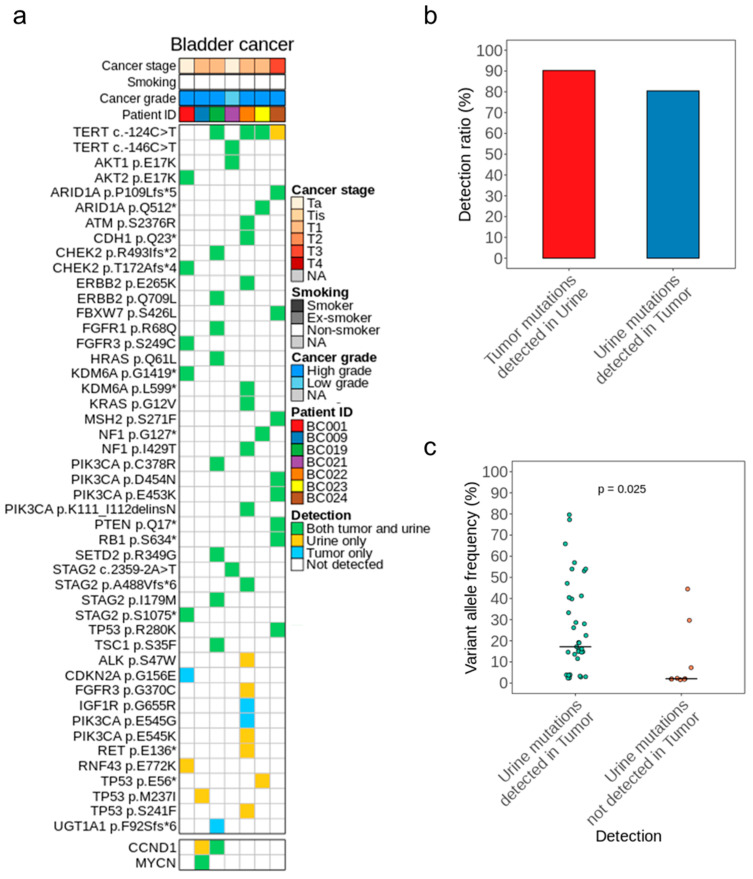
Mutational landscape of tumor and urine pairs. (**a**) Single-nucleotide variants (SNVs), insertions/deletions (INDELs), and copy-number variants (CNVs) detected in seven pairs of tumor and urine samples are summarized. Filled colors denote the detection status of each mutation. (**b**) The proportions of mutations identified in each sample type that were also found in the paired sample type are depicted. The red column denotes the proportion of mutations in tumor samples that were also detected in paired urine samples, and vice versa for the blue column. (**c**) Comparison of variants allele frequencies (VAFs) between mutations in urine samples that were detected or not detected in paired tumor samples. *p* value calculated by Student’s *t*-test.

**Figure 3 cancers-15-02868-f003:**
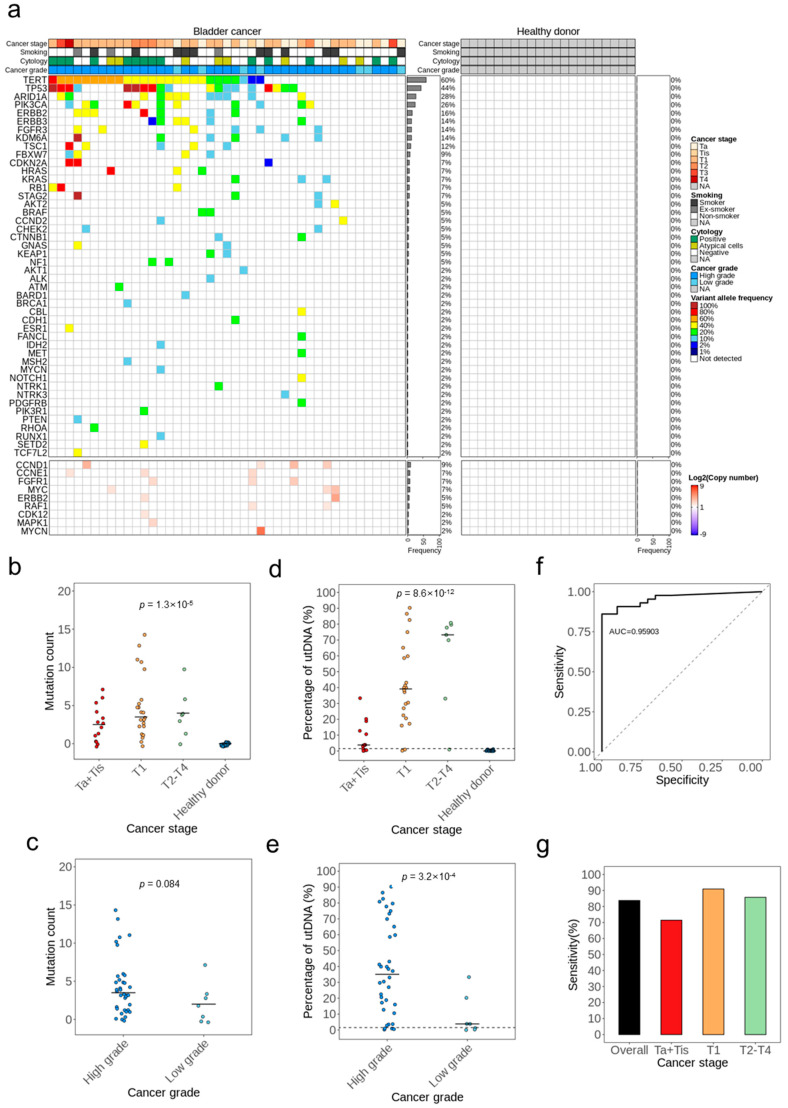
Results of utDNA detection without tumor samples. (**a**) Landscape of the genes with SNVs, INDELs, and CNVs identified by utDNA detection without tumor samples. Color-coded VAFs correspond to the mutations with the highest VAF in each gene. The grey bars on the right side denote the fraction of cases with the mutated/amplified gene. (**b**,**c**) Mutation counts (SNVs, INDELs) across disease-stage subgroups (**b**) or cancer-grade subgroups (**c**). (**d**,**e**) Proportions of utDNA in total urine DNA across disease-stage subgroups (**d**) or cancer-grade subgroups (**e**). The proportion of utDNA corresponds to the maximum VAF value per patient. (**f**) The receiver operating curve (ROC) of uAL100 for the detection of BC according to VAF filter levels (case: BC, control: healthy control). (**g**) The sensitivity of BC detection by uAL100 without tumor samples in different disease-stage subgroups. *p* values calculated by *t*-test for comparisons between two groups and ANOVA for comparisons among multiple groups.

**Figure 4 cancers-15-02868-f004:**
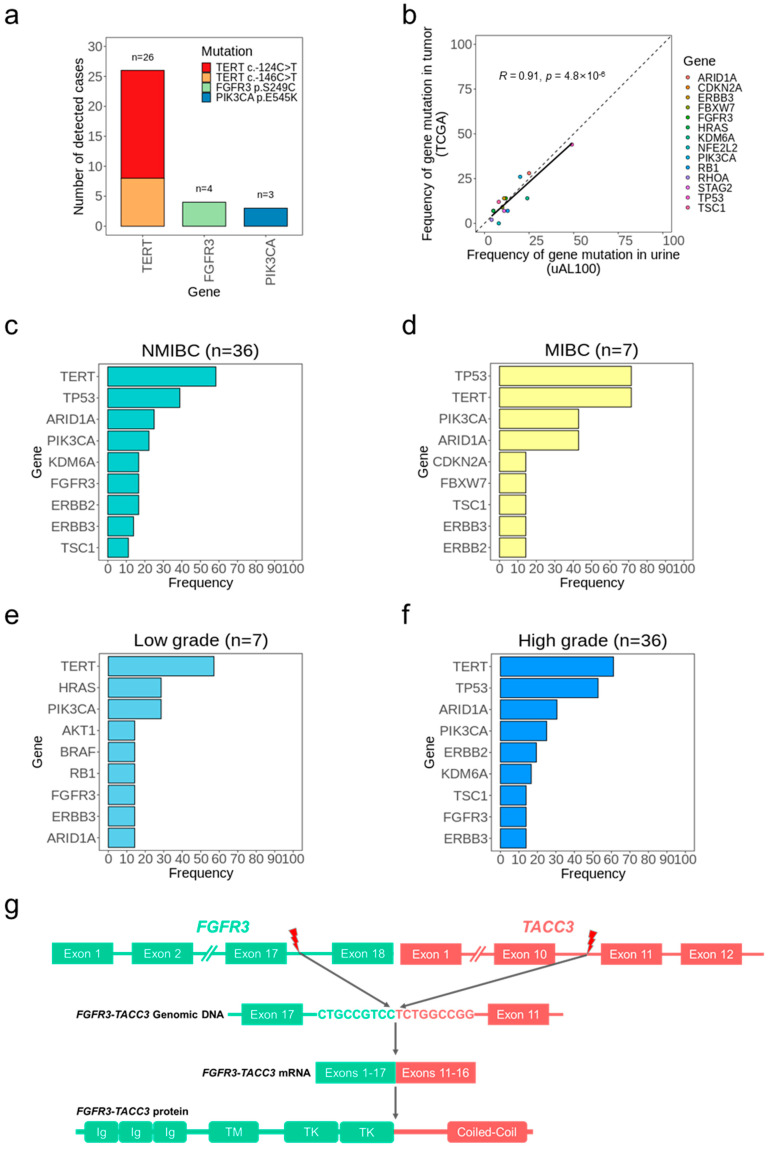
Genotypes of BC determined by utDNA detection without tumor samples. (**a**) SNVs detected in more than 5% of the patients with BC. (**b**) Comparisons of mutated gene frequency between tumor samples (TCGA) and urine samples (uAL100). The analyzed genes consisted of those that overlapped between the target genes of uAL100 and the significantly mutated genes identified in TCGA. The correlation coefficient (R) corresponds to the Pearson correlation coefficient. (**c**–**f**) Top nine most frequently mutated genes (SNVs/INDELs) per subgroup; (**c**) Non-muscle-invasive bladder cancer (NMIBC), (**d**) Muscle-invasive bladder cancer (MIBC), (**e**) High-grade bladder cancer, (**f**) Low-grade bladder cancer. (**g**) Graphical description of the *FGFR3*-*TACC3* fusion gene detected in urine from patient BC038. Red lightning bolts depict breakpoints of two genes involved in the fusion.

**Table 1 cancers-15-02868-t001:** Patient characteristics. M = male, F = female, Labels in cytology: 0 = Negative, 1 = Positive, 2 = Atypical cells.

Disease Status	Total	M/F	Median Age	Cancer Stage				Cancer Grade		Cytology		
				Ta	Tis	T1	T2–T4	High Grade	Low Grade	0	1	2
Bladder Cancer	43	28/15	68	13	1	22	7	36	7	21	14	8
Non-Bladder Cancer	21	20/1	64	–	–	–	–	–	–	–	–	–

**Table 2 cancers-15-02868-t002:** Comparisons of sensitivity and specificity between urine-based BC detection methods utilizing high-depth sequencing without tumor information.

Methods	Ward et al. (2022) [11]	Dudley et al. (2019) [10]	uAL100
Sample Type	Urine cell-pellet DNA	Urine DNA	Urine DNA
Sensitivity	87.3%	84%	83.7%
Specificity	84.8%	96%	100%

## Data Availability

The data and source codes used in this study are available upon request to the corresponding author, Duhee Bang (duheebang@yonsei.ac.kr).

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
