# Peer review of "Accurate Detection of Urothelial Bladder Cancer Using Targeted Deep Sequencing of Urine DNA"

_cancers, 2023, doi:10.3390/cancers15102868_

Round 1

Reviewer 1 Report

This is well written manuscript on an important topic. I don't have any major concerns or comments but a few points require clarification: 

Introduction:

- in line 48, the author state that high VAFs are linked to false-positive. This seems counterintuitive. Can they briefly explain why this is the case?

- lines 54-63 and Figure 1a, are more suited for methodology and result section. I suggest the authors leave only the first and final sentence here and move the other content elsewhere. 

Methods:

- lines 103-104. I find the use of the term 'germline mutation' misleading. In my opinion the term 'polymorphism' would be clearer.

- line 120. I am not sure this is the best way to calculate mutational concordance as it provides two different results, one for the tumour samples and one for the urine sample. As concordance is a measure of how much the results of tumour and urine analysis agree with each other, I would have expected to be a single value. 

- lines 125-127. The authors states that the mutation with the highest VAF in each patient was used to calculate sensitivity and specificity. Can the authors provide more details about how this was calculated?

Results:

- lines 151-154. I don't follow how these results were obtained from the formula in methods (line 120). Can the authors show the calculations? Also they state 'concordance was 90.5% for the tumor samples and 82.6% for the urine samples (Figure 2b) but in figure 2b the value reported are different from those stated. 

- line 211. The mutation rate of some of these genes are much lower than expected based on mutation rates reported in tumour tissue (e.g. FGFR3). Can the authors comment on this in the discussion?

Discussion

- line 272. The authors state that their test perform better compared with others. Can they report and reference specificity and sensitivity for the other tests to support their claim?

- line 275. Can the author add a comment regarding the feasibility of this type of testing (costs, time etc) for routine analysis within the health system (e.g. to replace cystoscopy)?

Author Response

May 13, 2023

Kiran Yu

Assistant Editor, Cancers

Dear Dr. Yu,

We are grateful to you and the Reviewers for your valuable comments and suggestions pertaining to our manuscript. We have taken all these comments into account and are submitting a revised version of our paper.

A full point-by-point response to the Reviewer’s comments is included below. We believe that incorporating the helpful suggestions from the Reviewers has substantially improved our manuscript, and it is now suitable for publication in Cancers.

Thank you for considering our revised manuscript. We look forward to hearing from you.

Kwang Hyun Kim, MD, Ph.D.

Department of Urology,

Ewha Womans University Seoul Hospital, Seoul 07804, Republic of Korea

Tel: +82-2-6986-1685

and

Duhee Bang, Ph.D.

Professor of Chemistry,

Yonsei University, Seoul 03722, Republic of Korea

Tel: +82-10-3357-0611

Reviewer Comments: Point-by-point response:

We thank the Reviewers for their constructive comments on our manuscript and recommendation of publication with modifications.

Listed below are our responses to the Reviewers’ suggestions for improvement, and the suggestions are grouped according to the Reviewer:

Reviewer #1

Comment 1-1

- In line 44, the author state that high VAFs are linked to false-positive. This seems counterintuitive. Can they briefly explain why this is the case?

Authors’ response

We appreciate the Reviewer’s valuable suggestion. We admitted that the pointed statement is counterintuitive since the indicated sentence is vague. Actually, we wrote the indicated sentence to imply the following claim: The variant allele frequencies of called mutations are high in urine samples from healthy donors as well as in urine samples from bladder cancer patients, which leads to misclassification of healthy donors to bladder cancer patients. Therefore, what we really wanted to claim was that high VAFs of mutations can be observed in urine DNA from healthy donors after stringent mutation calling, and we have observed such cases in previous studies (reference #11, 14). However, since this line can cause confusion for readers, as the Reviewer experienced, we decided to rephrase the indicated line by not mentioning “high VAF”, as follows:

Before:

[Line 51] Therefore, targeted deep sequencing of urine DNA is a promising approach for clinical applications in BC, although accurate detection of urine tumor DNA (utDNA) is challenging due to the presence of false-positive signals at high VAFs [11-14].

             After:

[Line 51] Therefore, targeted deep sequencing of DNA in supernatant or cell pellet of urine is a promising approach for clinical applications in BC, although accurate detection of BC is challenging due to the presence of significant levels of false-positive signals (sensitivity: 83-87% and specificity: 85~96%) [10,11,14,15].

Comment 1-2

- Lines 54-63 and Figure 1a, are more suited for methodology and result section. I suggest the authors leave only the first and final sentence here and move the other content elsewhere.

Authors’ response

We appreciate the Reviewer’s valuable suggestion. We have revised the indicated paragraph by replacing the sentences containing results with sentences that highlight the implications and significance of our method in clinical situations, as follows:

Before:

[Line 64] In this study, we describe uAL100, a urine-based BC detection method that accurately detects utDNA without matched tumor information. To validate the accuracy of uAL100, we applied uAL100 to 43 urine samples and 7 matched tumor samples from patients with BC and 21 urine samples from healthy donors (Figure 1a). We compared the mutations in the paired tumor and urine samples and observed a high level of mutational concordance within matched pairs. Specifically, 82.6% of the mutations identified in the urine samples were also identified in the matched tumor samples. The sensitivity and specificity of uAL100 for BC detection among the 43 patients and 21 controls were 83.7% and 100%, respectively. We confirmed that the mutations identified by uAL100 are significant in BC tumorigenesis and progression and have the potential to serve as biomarkers for BC.

After:

[Line 64] In this study, we describe uAL100, a urine-based BC detection method that accurately detects utDNA without matched tumor information. When applied to urine samples from our cohort, uAL100 successfully suppressed technical errors, resulting in a specificity of 100% for BC detection with a sensitivity of 83.7%. We expect that uAL100 will more reliably detect early BC than other urine based methods, thereby significantly reducing unnecessary tests for patients with hematuria. Also, uAL100 is expected to be helpful in making clinical decisions for patients with BC, since uAL100 provides genotypes of BC including mutations that are significant in BC tumorigenesis and progression. Collectively, we an-ticipate that uAL100 has the potential to greatly improve the efficiency and efficacy of BC treatment.

Also, we moved Figure 1a to the end of the result section “3.1. Patient Characteristics and Study Design” where it was first mentioned. Additionally, the sentences containing results were removed as they overlap with the contents in the result sections.

Comment 1-3

- Lines 103-104. I find the use of the term 'germline mutation' misleading. In my opinion the term 'polymorphism' would be clearer.

Authors’ response

We appreciate the Reviewer’s valuable suggestion. We admitted that the term ‘single nucleotide polymorphism (SNP) databases’ seems clearer than the term ‘germline mutation databases’ and changed indicated lines as follows:

Before:

[Line 112] Germline databases were used at the primary variant calling step to filter out putative germline mutations.

             After:

[Line 112] Single-nucleotide polymorphism databases were used at the primary variant calling step to filter out putative germline mutations.

Since we intended to remove only germline mutations (not somatic mutations), we decided not to change the term ‘germline mutation’ to ‘SNP’ in the other parts of the manuscripts.

Comment 1-4

- Line 120. I am not sure this is the best way to calculate mutational concordance as it provides two different results, one for the tumour samples and one for the urine sample. As concordance is a measure of how much the results of tumour and urine analysis agree with each other, I would have expected to be a single value.

Authors’ response

We appreciate the Reviewer’s valuable suggestion. We considered that both two values (the ratio of urine mutations detected in tumors and vice versa) necessary for the mutational concordance study as each value indicates positive predictive value and sensitivity, respectively. Therefore, we still want to present both values. However, revised the indicated line by changing the term ‘the mutational concordance level’ as follows:

Before:

[Line 127] The annotated SNVs and INDELs of all seven patients were then merged, and the mutational concordance level (detection ratio) was calculated using the following formula:

             After:

[Line 127] The annotated SNVs and INDELs of all seven patients were then merged, and the ratio of mutations of one given sample type that detected in paired sample type (detection ratio) was calculated using the following formula:

Comment 1-5

- Lines 125-127. The authors states that the mutation with the highest VAF in each patient was used to calculate sensitivity and specificity. Can the authors provide more details about how this was calculated?

Authors’ response

We appreciate the Reviewer’s valuable suggestion. We used ‘the highest VAF’ in the indicated lines when we investigated cutoff levels of VAF that give best sensitivity and specificity after applying all filters, which is not proper to be used in the indicated lines. Therefore, we revised the explanation of sensitivity and specificity without mentioning of “the highest VAF” as follows:

             Before:

              [Line 135] The mutation with the highest VAF in each patient was selected for calculation of the sensitivity and specificity of BC detection, and the respective VAFs were used as the percentage of utDNA in the entire sample of urine DNA. The sensitivity was the proportion of BC cases detected by uAL100 among 43 cases tested.

             After:

              [Line 135] The mutation with the highest VAF in each patient was selected for calculation of the sensitivity and specificity of BC detection, and the respective VAFs were used as the percentage of utDNA in the entire sample of urine DNA. The sensitivity was the proportion of BC cases having BC driver mutations in their urine DNA among 43 cases tested. The specificity was the proportion of the 21 controls that were correctly identified as not having BC driver mutations in their urine DNA.

Comment 1-6

- lines 151-154. I don't follow how these results were obtained from the formula in methods (line 120). Can the authors show the calculations? Also they state 'concordance was 90.5% for the tumor samples and 82.6% for the urine samples (Figure 2b) but in figure 2b the value reported are different from those stated.

Authors’ response

We appreciate the Reviewer’s valuable suggestion. This suggestion provided us with a valuable opportunity to correct our serious mistake. When we wrote the manuscript, we mistakenly typed the number of mutations for each group (“both tumor and urine”, “tumor only”, urine only”) in lines 151-154. In detail, the number of mutations for shared and urine only are 37 and 9, respectively. However, we mistakenly typed them as 38 and 8, respectively. Fortunately, there were no errors in the relevant figures (Figure 2a-c), so we corrected the indicated sentences as follows:

             Before:

[Line 167] Of the 50 mutations, 38 were shared between paired tumor and urine samples, whereas 12 were detected only in tumor samples or urine samples (tumor only = 4, urine only = 8; Figure 2a; Figure S2). As a result, the mutational concordance was 90.5% for the tumor samples and 82.6% for the urine samples (Figure 2b)

             After:

[Line 167] Of the 50 mutations, 37 were shared between paired tumor and urine samples, whereas 13 were detected only in tumor samples or urine samples (tumor only = 4, urine only = 9; Figure 2a; Figure S2). As a result, 90.2% of tumor mutations were also detected in paired urine samples, and 80.4% of urine mutations were also detected in paired tumor samples (Figure 2b).

Also, though now the Reviewer can deduce the same values using the formula in methods (line129) with the corrected numbers, we provide the specific calculations for deducing the detection ratio values as follows:

We sincerely appreciate this comment, and at the same time, are deeply reflecting on our critical mistake. We believe this valuable experience will be helpful in self-revision of our future publications.

Comment 1-7

- line 211. The mutation rate of some of these genes are much lower than expected based on mutation rates reported in tumour tissue (e.g. FGFR3). Can the authors comment on this in the discussion?

Authors’ response

We appreciate the Reviewer’s valuable suggestion. To answer this question, we compared the mutation rates in urine samples of our study and the mutation rates in bladder cancer tumor from a previous TCGA study [Figure 1 in Nature 507, 315–322 (2014)] with genes included in both the target genes of uAL100 and significantly mutated genes of the TCGA study. Contrary to the Reviewer’s concerns, we observed a high correlation of mutation rates between urine samples (uAL100) and bladder cancer tumor samples. Consequently, we added a relevant figure (Figure 4b) and a sentence in the result section “3.4. Bladder Cancer Genotypes Determined Using uAL100” as follows:

Before:

[Line 232] Other frequently mutated genes were TP53, ARID1A, PIK3CA, ERBB2, ERBB3, FGFR3, KDM6A, and TSC1 (> 10% of cases) that were observed as BC driver genes in previous studies (9, 24).

             After:

[Line 232] Other frequently mutated genes were TP53, ARID1A, PIK3CA, ERBB2, ERBB3, FGFR3, KDM6A, and TSC1 (> 10% of cases) that were observed as BC driver genes in previous studies [10,25]. Also, when compared to the TCGA study that investigated BC tumors, the frequency of cases with mutations in each gene detected by uAL100 showed a high correlation with those detected in the BC tumors reported in the TCGA study (R = 0.91; Figure 4b) [27], which indicates BC genotyping using uAL100 significantly reflects tumor information.

             Figure 4b

(b) Comparisons of mutated gene frequency between tumor samples (TCGA) and urine samples (uAL100). The analyzed genes consisted of those that overlapped between the target genes of uAL100 and the significantly mutated genes identified in TCGA. The correlation coefficient (R) corresponds to the Pearson correlation coefficient.

In the meanwhile, we are cautious about these results as they differ from what the Reviewer mentioned. Therefore, to provide a more profound answer to this question, we would like to know which previous studies the reviewer referred to.

Comment 1-8

- line 272. The authors state that their test perform better compared with others. Can they report and reference specificity and sensitivity for the other tests to support their claim?

Authors’ response

We appreciate the Reviewer’s valuable suggestion. We added a table that compares the sensitivity and specificity of urine-based methods that utilize deep sequencing (Table 2) after the discussion section, as follows:

Also, we added citations for the mentioned urine-based methods to the indicated lines (reference #9, 12), as follows:

             Before:

[Line 300] Previous urine-based methods for BC detection showed significant levels of false-positive signals with similar or lower sensitivity compared with our method.

After:

[Line 300] Previous urine-based methods using deep sequencing for BC detection showed significant levels of false-positive signals with similar or lower sensitivity compared with our method (Table 2) [10,11].

Comment 1-9

- line 275. Can the author add a comment regarding the feasibility of this type of testing (costs, time etc) for routine analysis within the health system (e.g. to replace cystoscopy)?

Authors’ response

We appreciate the Reviewer’s valuable suggestion. To respond to this comment, we merged the first and second paragraphs in the discussion section in to one paragraph and added a new paragraph that discusses about this comment, as follows:

             Before:

[Line 294] In this study, we described uAL100 and validated its performance in BC diagnosis. There was high concordance between mutations detected by uAL100 in matched urine and tumor samples, although the number of matched pairs was only seven in our cohorts. This indicated that the mutations detected in urine by uAL100 were derived from BC tumors. We then showed that uAL100 had 83.7% sensitivity and 100% specificity in detecting BC without tumor samples, suggesting that our method has high accuracy regardless of the tumor stage. Previous urine-based methods for BC detection showed significant levels of false-positive signals with similar or lower sensitivity compared with our method. A genotyping analysis showed that the mutations identified by uAL100 were highly related to BC tumorigenesis.

To increase convenience and reduce cost, we designed uAL100 to detect tumor DNA in urine without additional analysis of PBMCs from patients. We showed that accurate utDNA detection was possible without PBMCs; however, several mutations were removed by the germline filter in 15 of the 36 patients that were diagnosed with BC, so it is possible that the germline filter can lead to imperfect clinical decision making based on the mutation profiles that it produces. Therefore, although the germline filter had little effect on the sensitivity of BC detection without tumor samples, information from PBMCs may still be needed to support optimal clinical decisions.

             After:

[Line 294] In this study, we described uAL100 and validated its performance in BC diagnosis. There was high concordance between mutations detected by uAL100 in matched urine and tumor samples, although the number of matched pairs was only seven in our cohorts. This indicated that the mutations detected in urine by uAL100 were derived from BC tumors. We then showed that uAL100 had 83.7% sensitivity and 100% specificity in detecting BC without tumor samples, suggesting that our method has high accuracy regardless of the tumor stage. Previous urine-based methods using deep sequencing for BC detection showed significant levels of false-positive signals with similar or lower sensitivity com-pared with our method (Table 2) [10,11]. A genotyping analysis showed that the mutations identified by uAL100 were highly related to BC tumorigenesis. Additionally, we showed that accurate utDNA detection was possible without PBMCs which enables entire non-invasiveness of uAL100. However, since several mutations were removed by the germline filter in some patients that were diagnosed with BC, so it is possible that the germline filter can lead to imperfect clinical decision making based on the mutation pro-files that it produces. Therefore, although the germline filter had little effect on the sensitivity of BC detection without tumor samples, information from PBMCs may still be needed to support optimal clinical decisions.

For the clinical utilization of liquid biopsy for bladder cancer diagnosis, factors such as cost-effectiveness, test standardization and diagnostic accuracy should be taken into consideration. In this study, we described the process of sample preparation, DNA extraction and genomic analysis for our test and validated the diagnostic accuracy of uAL100 for bladder cancer. During the validation of uAL100, we showed that uAL100 accurately detects utDNA without matched PBMCs, which enables an increase in convenience and a reduction in cost. Regarding cost-effectiveness, although sequencing cost is decreasing, the cost for sequencing may still be more expensive than conventional urine tests for bladder cancer. Therefore, we do not expect uAL100 to fully replace cystoscopy which is the standard procedure for the diagnosis of bladder cancer. However, uAL100 has high diagnostic accuracy, and we expect that uAL100 could identify patients who needs cystoscopy and reduce the number of unnecessary procedures in patients who may have undergone invasive procedures without our test

Reviewer #2

Comment 2-1

-Could you please indicate the coverage of the sequencing for each sample?

Authors’ response

We appreciate the Reviewer’s valuable suggestion. We presented the median sequencing coverage of each sample as Table S2 and added sentences describing the median sequencing coverage values in the method section “2.2. Analysis of DNA from Urine and Bladder Tumors”, as follows:

             Before:

[Line 96] The target-enriched DNA libraries were sequenced using the Illumina Novaseq 6000 plat-form (Illumina, USA) to create 150 bp paired-end reads.

             After:

[Line 96] The target-enriched DNA libraries were sequenced using the Illumina Novaseq 6000 plat-form (Illumina, USA) to create 150 bp paired-end reads, with median sequencing depths of 6094× and 7732× for targeted regions in urine and tumor samples, respectively (Table S2).

Comment 2-2

- I suggest not including at the end of the introduction the main results of your results but instead a broad significance of the importance of this study

Authors’ response

We appreciate the Reviewer’s valuable suggestion. We have revised the indicated paragraph by replacing the sentences containing results with sentences that highlight the implications and significance of our method in clinical situations, as follows:

Before:

[Line 64] In this study, we describe uAL100, a urine-based BC detection method that accurately detects utDNA without matched tumor information. To validate the accuracy of uAL100, we applied uAL100 to 43 urine samples and 7 matched tumor samples from patients with BC and 21 urine samples from healthy donors (Figure 1a). We compared the mutations in the paired tumor and urine samples and observed a high level of mutational concordance within matched pairs. Specifically, 82.6% of the mutations identified in the urine samples were also identified in the matched tumor samples. The sensitivity and specificity of uAL100 for BC detection among the 43 patients and 21 controls were 83.7% and 100%, respectively. We confirmed that the mutations identified by uAL100 are significant in BC tumorigenesis and progression and have the potential to serve as biomarkers for BC.

After:

[Line 64] In this study, we describe uAL100, a urine-based BC detection method that accurately detects utDNA without matched tumor information. When applied to urine samples from our cohort, uAL100 successfully suppressed technical errors, resulting in a specificity of 100% for BC detection with a sensitivity of 83.7%. We expect that uAL100 will more reliably detect early BC than other urine based methods, thereby significantly reducing unnecessary tests for patients with hematuria. Also, uAL100 is expected to be helpful in making clinical decisions for patients with BC, since uAL100 provides genotypes of BC including mutations that are significant in BC tumorigenesis and progression. Collectively, we an-ticipate that uAL100 has the potential to greatly improve the efficiency and efficacy of BC treatment.

Also, we moved Figure 1a to the end of the result section “3.1. Patient Characteristics and Study Design” where it was first mentioned. Additionally, the sentences containing results were removed as they overlap with the contents in the result sections.

Reviewer #3

Comment 3-1, 3-2

- Line 54-63: This paragraph should be rewritten and focus on the objective of the work. Elements that are not part of the discussion are developed.

- Figure 1 should be removed and placed in the materials and methods and/or results part.

Authors’ response

We appreciate the Reviewer’s valuable suggestion. We have revised the indicated paragraph by replacing the sentences containing results with sentences that highlight the implications and significance of our method in clinical situations, as follows:

Before:

[Line 64] In this study, we describe uAL100, a urine-based BC detection method that accurately detects utDNA without matched tumor information. To validate the accuracy of uAL100, we applied uAL100 to 43 urine samples and 7 matched tumor samples from patients with BC and 21 urine samples from healthy donors (Figure 1a). We compared the mutations in the paired tumor and urine samples and observed a high level of mutational concordance within matched pairs. Specifically, 82.6% of the mutations identified in the urine samples were also identified in the matched tumor samples. The sensitivity and specificity of uAL100 for BC detection among the 43 patients and 21 controls were 83.7% and 100%, respectively. We confirmed that the mutations identified by uAL100 are significant in BC tumorigenesis and progression and have the potential to serve as biomarkers for BC.

After:

[Line 64] In this study, we describe uAL100, a urine-based BC detection method that accurately detects utDNA without matched tumor information. When applied to urine samples from our cohort, uAL100 successfully suppressed technical errors, resulting in a specificity of 100% for BC detection with a sensitivity of 83.7%. We expect that uAL100 will more reliably detect early BC than other urine based methods, thereby significantly reducing unnecessary tests for patients with hematuria. Also, uAL100 is expected to be helpful in making clinical decisions for patients with BC, since uAL100 provides genotypes of BC including mutations that are significant in BC tumorigenesis and progression. Collectively, we an-ticipate that uAL100 has the potential to greatly improve the efficiency and efficacy of BC treatment.

Also, we moved Figure 1a to the end of the result section “3.1. Patient Characteristics and Study Design” where it was first mentioned. Additionally, the sentences containing results were removed as they overlap with the contents in the result sections.

Comment 3-3

- How was the tumor material obtained? Was it obtained from paraffin-embedded material? Or was it obtained from fresh selection? If fresh material was used, was the presence of tumor confirmed? How was the selection carried out?

Authors’ response

We appreciate the Reviewer’s valuable suggestion. As stated in Material and Methods section, tumor sample was obtained at the time of surgery and stored as a fresh frozen sample. However, pathologic review for fresh frozen sample was not performed in this study.

Comment 3-4

- Why were only 7 paired cases studied? What were the clinicopathological characteristics of these seven patients. For example, pathological stage, histopathological grade, etc.

Authors’ response

We appreciate the Reviewer’s valuable suggestion. Sequencing of tumor DNA was performed only in the initial samples of this study, and 7 paired cases were not selected for this study. After sequencing of 7 tumor DNA for concordance analysis of tumor and urine DNA, we could determine the cutoff VAF for urine DNA for detecting bladder cancer with high sensitivity and specificity. As stated in the result section “3.2. Mutational Concordance in Paired Samples of Tumor and Urine DNA”, although tumor samples were analyzed only in 7 patients, most patients had non-muscle invasive bladder cancer. Therefore, we believe our method is reliable regardless of tumor stage. The patient characteristics of 7 patients are summarized in the table below and shown graphically in Figure 2a.

Comment 3-5

- Table 1. It is noteworthy that most of the cases included in the study were high-grade carcinomas, and that the cytological study was diagnostic in less than half of the cases. What classification system was used to assess the cytologies? Was the cytological material analyzed from a new sample or was the same material used for DNA extraction? Please detail.

Authors’ response

We appreciate the Reviewer’s valuable suggestion. Urine samples for cytological examination was obtained at the time of cystoscopy in clinics (different urine samples for DNA extraction) and the pathologist reported the results according to the Paris System [Acta Cytologica (2016) 60 (3): 185–197.]. If the result was reported as ‘suspicious for high-grade urothelial carcinoma’ or ‘high-grade urothelial carcinoma’, it was categorized as positive and in the case of negative for urothelial carcinoma or no adequate diagnosis possible, then it was categorized as negative. We briefly added a sentence stating about cytology in the method section “2.1. Patient Enrollment and Sample Collection” as follows:

             Before:

[Line 80] All patients provided informed consent for tissue banking and genetic testing.

             After:

[Line 80] All patients provided informed consent for tissue banking and genetic testing. Urine cytology was performed at the time of cystoscopy according to the Paris System [17].

Comment 3-6

- During the presentation of the results, reference is made to their discussion, including bibliographical references. This should be modified and clearly show the results of the study.

Authors’ response

We appreciate the Reviewer’s valuable suggestion. We suspect that the Reviewer raised this comment, especially with regards to the result sections “3.4. Bladder Cancer Genotypes Determined Using uAL100” or “3.5. Information from Peripheral Blood Mononuclear Cells in Urine-Based Bladder Cancer Detection”. Here, we would like to state the validity of using several citations in those result sections.

First, in the result section 3.4, we had to verify the significance of mutations identified by uAL100, and we determined that comparison with the results of previous studies were necessary. Since we analyzed small numbers of tumors (n=7), we believe that comparing our results with those of previous studies was the best way to validate our genotyping results.

For the result section 3.5, we wanted to claim that without matched PBMCs, accurate BC detection is possible using uAL100. However, we did not analyze matched PBMCs for direct validation of our hypothesis. Therefore, we had to indirectly validate our hypothesis, and for inferring processes, cited studies are needed. In summary due to the absence of matched PBMCs, we inferred conclusions that were largely dependent on results of previous studies we cited, which is one of the limitations of our study. While we mentioned this limitation in the end of the result section 3.5, we modified the relevant sentence to make it more clearer, as follows:

Before:

[Line 279] Based on these results, we suggest that PBMCs are not necessary for detecting BC using urine samples; however, further study is required with larger numbers of healthy controls.

After:

[Line 279] Based on these results, we suggest that PBMCs are not necessary for detecting BC using urine samples; however, further study is required with larger numbers of urine samples and matched PBMCs for precise validation of our hypothesis.

We hope that our presented validity can be positively reflected in reconsidering of the Reviewer’s opinions in comment 3-6.

Comment 3-7, 3-8

- The relevant findings of the study are not discussed, nor are the advantages and disadvantages taken into account.

- The discussion must be modified and the analysis supported with adequate bibliographical sources.

Authors’ response

We appreciate the Reviewer’s valuable suggestions. We determined to address both comment 3-7 and comment 3-8 simultaneously, as the Reviewer’s comment ‘The discussion must be modified’ contains both of these comments.

In comment 3-7, the Reviewer recommended to discuss about the advantages and disadvantages of our method. We considered that we have already presented advantages (enhanced accuracy and entirely non-invasiveness due to absence of PBMC) and disadvantages (loss of putative driver mutations due to absence of PBMC) in the first and second paragraphs of unrevised version of our manuscript. However, we also added the advantages of uAL100 in the middle of discussion section in terms of capturing genomic landscape of NMIBC for clinical decisions, since accurate genotyping of NMIBC is difficult due to tumor cell heterogeneity:

[Line 324] As in our cohort, NMIBC is the most common type of bladder cancer at the time of diagnosis. However, unlike similar clinical and histopathological characteristics of NMIBC tumors, their genomic landscapes are variable, resulting in largely different dis-ease progression and response to treatment [33,34]. Consequently, about 50-70% of NMIBC patients experience recurrence and progression to MIBC, necessitating continuous cystoscopy and diverse therapeutic interventions [35,36]. Therefore, accurate capture of the genomic landscape of NMIBC, including its tumor cell heterogeneity, is necessary for the complete cancer remission in early stages of the disease. However, the amount of released tumor-derived materials in bodily-fluids for NMIBC are small, and therefore reflecting integrative information of NMIBC is challenging. To overcome such challenges, we envision that uAL100, with its high accuracy, has the potential to capture genomic landscape of NMIBC containing information from subclonal or metastatic tumors at low levels. Thus, uAL100 is expected to be helpful in early clinical decisions and lifelong surveillance by preventing excessive cystoscopy and therapeutic intervention, which we need to validate in the future studies.

Also, as the Reviewer recommended, we added citations of previous methods when we compared our accuracy with them, as follows:

             Before:

[Line 300] Previous urine-based methods for BC detection showed significant levels of false-positive signals with similar or lower sensitivity compared with our method.

             After:

[Line 300] Previous urine-based methods using deep sequencing for BC detection showed significant levels of false-positive signals with similar or lower sensitivity compared with our method (Table 2) [10,11].

Additionally, we added a table that compared sensitivity and specificity of the methods (Table 2) after discussion section, as follows:

Finally, we added more limitations in terms of low numbers of tumor and urine sample pairs and absence of surveillance study of uAL100 to the final paragraph of the discussion section where we presented the limitations of our study, as follows:

             Before:

[Line 339] We found that uAL100 performed well in detecting BC without tumor or blood samples; however, our study has several limitations. First, our cohort was small, and further validation of our method with larger samples is needed. Second, since the analysis pipe-line of uAL100 was originally developed for plasma samples, the error suppression may not be sufficient to remove all errors in urine DNA. Therefore, although the current version of error suppression in uAL100 is reliable, an error suppression method specifically for urine samples is needed to improve the accuracy of uAL100. Finally, since our target capture panel is not solely composed of genes related to BC, the cost of our method is not fully optimized for BC detection. Furthermore, some known BC-related genes, such as PLEKHS1, are not contained in the target capture panel. Consequently, it might be possible to in-crease the sensitivity for BC detection while reducing the error rate and cost by adjusting the gene panel.

             After:

[Line 339] We found that uAL100 performed well in detecting BC without tumor or blood samples; however, our study has several limitations. First, our cohort was small, and further validation of our method with larger samples is needed. Specifically, more tumor and urine sample pairs are need to be analyzed to obtain statistical significance of mutational concordance study, since we only analyzed 7 sample pairs, which is a small number to provide statistical significance to the results. Second, since the analysis pipeline of uAL100 was originally developed for plasma samples, the error suppression may not be sufficient to re-move all errors in urine DNA. Therefore, although the current version of error suppression in uAL100 is reliable, an error suppression method specifically for urine samples is needed to improve the accuracy of uAL100. Third, since our target capture panel is not solely composed of genes related to BC, the cost of our method is not fully optimized for BC detection. Furthermore, some known BC-related genes, such as PLEKHS1, are not contained in the target capture panel. Consequently, it might be possible to increase the sensitivity for BC detection while reducing the error rate and cost by adjusting the gene panel. Finally, as we mentioned, surveillance of NMIBC patients is important due to their high recurrence rate. Therefore, we need to validate the surveillance ability of uAL100 to make our method more meaningful in real clinical situations.

With mentioned modifications, we believe that we satisfied the Reviewer’s recommendation ‘The discussion must be modified’.

Comment 3-9

- Finally, the references do not have the format required by the journal.

Authors’ response

We appreciate the Reviewer’s valuable suggestion. We changed the style of reference that satisfies criteria of Cancers.

Reviewer #4

Comment 4-1

- How many authors are sure about the accuracy of the screening, especially because urine-based method detection has significant levels of false-positive signals? During disease progression, individual disseminated tumor cells and consecutively metastases can acquire characteristics that do not match those of the corresponding primary tumors, and often are only hardly assessable for further evaluation.

Authors’ response

We appreciate the Reviewer’s valuable suggestion. We would like to clarify that all authors who reviewed the results of bladder cancer screening using urine DNA (including all correspondence and first authors) agreed on the accuracy of our method for BC detection, even though the number of samples we assayed was not large enough. As described in the manuscript, uAL100 was developed based on AL100, which was developed for blood-based cancer screening and AL100 showed significantly low levels of false positive signals in our previous study [Br J Cancer 127, 898–907 (2022)]. When we directly applied AL100 to urine DNA, we observed several significant false positive mutations in healthy control groups, as Reviewer indicated. To remove mutations in healthy control groups, we increased the cutoff level of variant allele frequency from 0.1-0.5% to 1.5% and utilized several oncogene databases, which were the main differences. As a result, we were able to remove false positive mutations in healthy control groups and, through mentioned developing processes, attain certainty in our results.

Regarding the latter statement of this comment, we agreed that obtaining metastatic tumor-derived materials is challenging, especially when metastasis occurred outside of urinary tract. However, we anticipate that metastatic tumor-derived materials can flow in urines and eventually be detected by sampling urine. However, this hypothesis will require further studies to be validated.

Comment 4-2

- Author should describe more about the Alpha Liquid technology if it is only for screening Circulating tumor DNA (ctDNA), how is this platform developed, and other details in the manuscript. This information will be useful for other scientists and clinicians.

Authors’ response

We appreciate the Reviewer’s valuable suggestion. The detailed description about the AL100 was presented in our previous study [Br J Cancer 127, 898–907 (2022)], and we referred this study in the introduction section of our manuscript. Also, another study that utilized blood-based AL100 is preparing for publication, we anticipate that other scientists and clinicians can more easily find and understand principle and properties of AL100.

Comment 4-3, 4-4, 4-6

- Does this technology have a patent, or anyone can use it?

- How much feasible is this technology for the common lab to use?

- Also author should mention the real clinical scenario with reflect on uAL100 technology and whether it has already been approved using blood-serum-based screening for cancer patients such as colorectal or others.

Authors’ response

We appreciate the Reviewer’s valuable suggestion. Here, we would like to reply to comments 4-3, 4-4, and 4-6 simultaneously, since those comments seems closely related. IMBdx, one of the affiliations of authors of this manuscript, is one of the main group (company) that developed the Alpha Liquid technology and has a commercial license for the blood-based AL100 technology. IMBdx is a distributor of the Alpha Liquid technology and provides it in several Asian countries as well as South Korea. Also, IMBdx is preparing to distribute the technology to western countries. As a result, we anticipate that both AL100 and uAL100 can be utilized wordwide, even in common labs.

Regarding the response to comment 4-6, we added a new paragraph in the discussion section, as follows:

[Line 311] For the clinical utilization of liquid biopsy for bladder cancer diagnosis, factors such as cost-effectiveness, test standardization and diagnostic accuracy should be taken into consideration. In this study, we described the process of sample preparation, DNA extraction and genomic analysis for our test and validated the diagnostic accuracy of uAL100 for bladder cancer. During the validation of uAL100, we showed that uAL100 accurately detects utDNA without matched PBMCs, which enables an increase in convenience and a reduction in cost. Regarding cost-effectiveness, although sequencing cost is decreasing, the cost for sequencing may still be more expensive than conventional urine tests for bladder cancer. Therefore, we do not expect uAL100 to fully replace cystoscopy which is the standard procedure for the diagnosis of bladder cancer. However, uAL100 has high diagnostic accuracy, and we expect that uAL100 could identify patients who needs cystoscopy and reduce the number of unnecessary procedures in patients who may have undergone invasive procedures without our test.

In the case of the approval of blood-serum-based cancer screening of AL100, AL100 is already utilized in 23 major cancer centers in South Korea, and we mentioned above, we are preparing approval and commercialization of AL100 in several other countries. However, we determined that these statements are not proper to mention in the manuscript and decided not to mention them.

Comment 4-5

- Indeed, screening BC patients using liquid biopsy samples such as urine is an effective effort. However, the mirror image of tumor cell heterogeneity and real-time reflection in the genomic landscape for non-muscle invasive cancer is difficult. The author’s viewpoint is important in this regard and may include in the conclusion section of the manuscript.

Authors’ response

We appreciate the Reviewer’s valuable suggestion. We agree that obtaining the entire genomic landscape of NMIBC is crucial for making proper clinical decisions and achieving consequent complete remission of the disease. Also, we agree that reflecting information from the whole clones of NMIBC tumors is challenging due to the severely low levels of subclonal-tumor-derived materials in urine. Nevertheless, we believe that still liquid biopsy has advantageous in obtaining integrative tumor information, particularly when considering tumor metastasis and ease of serial sampling. With its high accuracy, we expect that uAL100 has the potential to reflect entire tumor heterogeneity with further developments. Also, as we mentioned earlier, real-time acquisition of genomic landscapes may be obtainable with serial sampling of urine at different time points. We added a new paragraph in the discussion section to address this comment in more detail, as follows:

              [Line 324] As in our cohort, NMIBC is the most common type of bladder cancer at the time of diagnosis. However, unlike similar clinical and histopathological characteristics of NMIBC tumors, their genomic landscapes are variable, resulting in largely different dis-ease progression and response to treatment [33,34]. Consequently, about 50-70% of NMIBC patients experience recurrence and progression to MIBC, necessitating continuous cystoscopy and diverse therapeutic interventions [35,36]. Therefore, accurate capture of the genomic landscape of NMIBC, including its tumor cell heterogeneity, is necessary for the complete cancer remission in early stages of the disease. However, the amount of released tumor-derived materials in bodily-fluids for NMIBC are small, and therefore reflecting integrative information of NMIBC is challenging. To overcome such challenges, we envision that uAL100, with its high accuracy, has the potential to capture genomic landscape of NMIBC containing information from subclonal or metastatic tumors at low levels. Thus, uAL100 is expected to be helpful in early clinical decisions and lifelong surveillance by preventing excessive cystoscopy and therapeutic intervention, which we need to validate in the future studies.

Reviewer #5

Comment 5-1

- The introduction could be a little longer. In the introduction, I would suggest taking information from a recently published article within MDPI (e.g. 10.3390/ijms232113206 or a different analysis) reviewing FDA-approved tests along with many others including ncRNA and cfDNA, and state the actual poor sensitivities and specificities. – also, in this case, lines 37-38 with “… however, and …” is a poor choice of phrasing please rephrase.

Authors’ response

We appreciate the Reviewer’s valuable suggestion. In this comment, the Reviewer recommended revising the introduction section with several opinions: 1) lengthening the introduction section, 2) stating additional urine-based BC detection methods, 3) stating the actual sensitivity and specificity, and 4) rephrasing lines 37-38. We revised the introduction section by reflecting the Reviewer’s opinions with several additional changes as follows:

             Before:

[Page 1, Introduction] Hematuria is a prominent symptom in the early stages of bladder cancer (BC) and is present in 80–90% of patients with BC [1]. However, only 10–22% of patients with hematuria are eventually diagnosed with BC [2, 3]. Cystoscopy, the gold-standard method for detecting BC, is an invasive and costly procedure that carries risks of complication, and its accuracy is highly dependent on the skills of the operator. Therefore, there is a need for non-invasive tests to detect BC with reliable accuracy.

Several non-invasive urine-based tests have been developed to complement invasive cystoscopy for BC diagnosis (e.g. urine cytology, UroVysion, and NMP22) [4-6]. These tests have poor sensitivity, however, and do not provide genotyping information to aid clinical decisions [5, 6]. As an alternative, targeted deep sequencing of cell-free DNA (cfDNA) in blood or urine can detect genomic mutations at low allele frequencies, enabling early diagnosis of BC [7-10]. In patients with BC, DNA in urine has been reported to contain more mutations with higher variant allele frequencies (VAFs) than cfDNA in blood [10]. There-fore, targeted deep sequencing of urine DNA is a promising approach for clinical applications in BC, although accurate detection of urine tumor DNA (utDNA) is challenging due to the presence of false-positive signals at high VAFs [11-14].

In a previous study, we detected circulating tumor DNA in the blood of patients with colorectal cancer using our AlphaLiquid100 target capture panel and its paired analysis pipeline, which we refer to collectively as AL100 [15]. The analysis pipeline suppresses errors in deep sequencing data for 118 cancer-associated gene regions and detects sin-gle-nucleotide variants (SNVs), insertions/deletions (INDELs), copy-number variations (CNVs), and gene fusions with VAFs as low as 0.1%. The AlphaLiquid100 panel includes genes associated with BC, so we decided to utilize AL100 to accurately detect utDNA for the diagnosis of BC.

In this study, we describe uAL100, a urine-based BC detection method that accurately detects utDNA without matched tumor information. To validate the accuracy of uAL100, we applied uAL100 to 43 urine samples and 7 matched tumor samples from patients with BC and 21 urine samples from healthy donors (Figure 1a). We compared the mutations in the paired tumor and urine samples and observed a high level of mutational concordance within matched pairs. Specifically, 82.6% of the mutations identified in the urine samples were also identified in the matched tumor samples. The sensitivity and specificity of uAL100 for BC detection among the 43 patients and 21 controls were 83.7% and 100%, respectively. We confirmed that the mutations identified by uAL100 are significant in BC tumorigenesis and progression and have the potential to serve as biomarkers for BC.

             After:

[Page 1, Introduction] Hematuria is a prominent symptom in the early stages of bladder cancer (BC) and is present in 80–90% of patients with BC [1]. However, only 10–22% of patients with hematuria are eventually diagnosed with BC [2,3]. Cystoscopy, the gold-standard method for detecting BC, is an invasive and costly procedure that carries risks of complication, and its accuracy is highly dependent on the skills of the operator. Therefore, there is a need for non-invasive tests to detect BC with reliable accuracy.

Several non-invasive urine-based tests have been developed to complement invasive cystoscopy for BC diagnosis utilizing materials in urine (e.g. urine cytology, BTA, UroVysion, uCyt+, and NMP22) [4-7]. However, these tests have poor accuracy in detecting BC, with sensitivities between 56% and 83% and specificities between 64% and 88%. Additionally, these urine-based tests do not provide genotyping information to aid clinical decisions [5,6]. Therefore, there is a need for the development of alternative non-invasive methods with enhanced accuracy and genetic information of BC.

As alternative non-invasive methods for detecting BC, targeted deep sequencing of cell-free DNA (cfDNA) in blood or DNA in urine (supernatant or cell pellet) have been developed, as these methods can detect genomic mutations at low allele frequencies, enabling diagnosis and genotyping of BC in early stages of the disease [8-12]. Especially, deep sequencing of urine is more advantageous than blood in BC detection since urine contains relatively more enriched tumor-derived materials than blood due to a lower amount of leukocyte-derived materials in patients with BC [12,13]. Additionally, since urine can be sampled entirely non-invasively with practically unlimited sample volume, urine sampling is more advantageous than blood sampling. Therefore, targeted deep sequencing of DNA in supernatant or cell pellet of urine is a promising approach for clinical applications in BC, although accurate detection of BC is challenging due to the presence of significant levels of false-positive signals (sensitivity: 83-87% and specificity: 85~96%) [10,11,14,15].

In a previous study, we detected circulating tumor DNA in the blood of patients with colorectal cancer using our AlphaLiquid100 target capture panel and its paired analysis pipeline, which we refer to collectively as AL100 [16]. The analysis pipeline suppresses errors in deep sequencing data for 118 cancer-associated gene regions and detects single-nucleotide variants (SNVs), insertions/deletions (INDELs), copy-number variations (CNVs), and gene fusions with variant allele frequencies (VAFs) as low as 0.1%. The Al-phaLiquid100 panel includes genes associated with BC, so we decided to utilize AL100 to accurately detect urine tumor DNA (utDNA) in urine supernatant for the diagnosis of BC.

In this study, we describe uAL100, a urine-based BC detection method that accurately detects utDNA without matched tumor information. When applied to urine samples from our cohort, uAL100 successfully suppressed technical errors, resulting in a specificity of 100% for BC detection with a sensitivity of 83.7%. We expect that uAL100 will more reliably detect early BC than other urine based methods, thereby reducing unnecessary tests for patients with hematuria. Also, uAL100 is expected to be helpful in making clinical decisions for patients with BC since uAL100 provides genotypes of BC including mutations that are significant in BC tumorigenesis and progression. Collectively, we anticipate that uAL100 has the potential to greatly improve the efficiency and efficacy of BC detection and treatment.

We cited a journal that the Reviewer recommended as #Reference 7, and added two more methods, BTA and uCyto+. Also, we stated actual sensitivity and specificity levels of previous methods, and added the advantages of deep sequencing of urine than blood. Finally, we revised last paragraph by replacing results with implication and significance of uAL100.

Comment 5-2

- How cost-effective would your method of diagnosis be compared to today’s golden standard?

Authors’ response

If we consider the golden standard in terms of liquid biopsy methods, we would suggest the blood-based method of Guardant Health that is the most commercially advanced company in the field of liquid biopsy. When compared to Guardant Health’s method, the cost of our method is 1/5 (>$1,000). Therefore, we would like to claim that our method has cost competitiveness. However, we considered that cost comparison between blood-based method is not proper to be discussed in our study, we omitted such a comparison.

In the meanwhile, urine-based commercial non-invasive BC detection methods, such as the ImmunoCyt test of Scimedx, the nuclear matrix protein 22 (NMP22) immunoassay test of Matritech, and multi-target FISH (UroVysion), may be options for BC detection with lower costs than uAL100. However, as we mentioned in our manuscript, since the accuracy of these methods is poor and genotyping of bladder cancer is not available, we consider price comparison with these methods inappropriate.

Comment 5-3

- What was the VAF distribution in the study? Why did you choose this germline filter? Subjects under 40, 40-60, …, above 95 and were there any subjects outside of ranges?

Authors’ response

We appreciate the Reviewer’s valuable suggestion. For the removal of putative germline mutations, we removed mutations with VAFs between 40%-60% or >95%. Although theoretical VAFs of germline mutations are 50% or 100%, we broadened the ranges to sufficiently remove germline mutations since germline mutations showed VAFs between 40%-60% or >95% in real situations, when we analyzed PBMCs in our other studies which we have not published. As we mentioned in the result section “3.5. Information from Peripheral Blood Mononuclear Cells in Urine-Based Bladder Cancer Detection” and the discussion section, VAFs of several mutations were included in the 40%-60% or >95% range and excluded from the final result. Nevertheless, due to this germline filter, the sensitivity was not decreased while the specificity increased. Therefore, we concluded that this germline filter is effective in preventing healthy donors from being classified as bladder cancer patients while preserving sensitivity. However, more urine samples and paired PBMCs need to be analyzed to more profoundly test this germline filter, as discussed in the result section 3.5. In our cohort, we could not observe any subjects outside of the ranges, which was inferred from the absence of any detected mutations in healthy donors.

Comment 5-4

- Line 122-123 T-tests are used to compare or determine the differences between groups.

Authors’ response

We appreciate the Reviewer’s valuable suggestion. We corrected explanation of T-test as the reviewer commented, and changed indicated line, as follows:

             Before:

              [Line 132] Comparisons between two groups were performed using t-tests. ANOVA was used for comparisons of more than two groups.

             After:

[Line 132] T-test was used to test for differences between two groups, and ANOVA was used for more than two groups.

Comment 5-5

- Line 138-139 are there any differences in results for NMIBC and MIBC

Authors’ response

We appreciate the Reviewer’s valuable suggestion. There were differences in the frequency of mutated genes between NMIBC and MIBC, as shown in Figure 4b, although the number of samples from MIBC patients was relatively small compared to samples from NMIBC patients. We presented this result in the result section “3.4. Bladder Cancer Genotypes Determined Using uAL100” in our original manuscript. Also, the mutation counts and percentage of utDNA were higher in urines from MIBC patients than those from NMIBC patients, as shown in Figure 3b and d. However, in terms of sensitivity levels, NMIBC and MIBC were not significantly different (83.3% vs 85.7%). Since NMIBC and MIBC can be distinguished from cancer stages, we considered that presenting sensitivity levels according to tumor stage would sufficiently explain the sensitivity levels of NMIBC and MIBC. However, additional presentation of sensitivity levels in terms of histology would provide a broader perspective. Therefore, we added a relevant sentence in the result section “3.3. Bladder Cancer Detection without Tumor Samples”, as follows:

             Before:

[Line 199] In a subgroup analysis according to tumor stage, the sensitivity of detection in subgroups with stage Ta+Tis, T1, and T2–T4 disease was 71.4%, 90.9%, and 85.7%, respectively, with a specificity of 100% in each subgroup (Figure 3g).

After:

[Line 199] In a subgroup analysis according to tumor stage, the sensitivity of detection in subgroups with stage Ta+Tis, T1, and T2–T4 disease was 71.4%, 90.9%, and 85.7%, respectively, with a specificity of 100% in each subgroup (Figure 3g). When subgroups with stages that cor-respond to NMIBC were combined, the sensitivity of detection was 83.3%, which is similar to those of MIBC (T2-T4).

Comment 5-6

- Lines 156-157 specify the only one found mutation as this part gets distracting when reading.

Authors’ response

We appreciate the Reviewer’s valuable suggestion. We admitted that the indicated line could lead to confusion for readers, and we specified indicated mutation, as follows:

             Before:

[Line 172] Only one mutation was detected in the other patient, and that mutation was detected only in the urine DNA.

             After:

[Line 172] Only one mutation, TP53 M237I, was detected in the other patient, and that mutation was detected only in the urine DNA.

Comment 5-7

- Line 159-161 please rephrase for clarity.

Authors’ response

We appreciate the Reviewer’s valuable suggestion. We admitted that the indicated line can occur confusions for readers, and we changed those line clearly, as follows:

             Before:

              [Line 175] Among the mutations detected in the urine samples, those that were shared with matched tumor samples had higher VAFs than those that were not shared (median VAF = 17.18% vs. 2.06%; p = 0.025; Figure 2c).

After:

[Line 175] Within urine samples, the mutations that were shared with matched tumor samples had a higher median VAF than those that were not shared with matched tumor samples (median VAF = 17.18% vs. 2.06%; p = 0.025; Figure 2c).

Comment 5-8

- Line 163 comparison with 7 samples, ‘6 of 7’ positive is still a low statistical amount to state that these are truly positive. This is a considerable limitation that should be stated in the limitations.

Authors’ response

We appreciate the Reviewer’s valuable suggestion. We agreed with the Reviewer’s opinion and recognized that the low number of available paired samples is one of the limitations of our study when we wrote this manuscript. However, while we were focusing on the performance and limitations of uAL100 in terms of BC detection, we missed stating the limitation of our mutational concordance study. Therefore, we added a statement about the pointed limitation in the result section “3.2. Mutational Concordance in Paired Samples of Tumor and Urine DNA” after the indicated line, as follows:

             Before:

[Line 180] We concluded that most of the mutations detected in the urine samples by uAL100 were true positives, suggesting that uAL100 accurately detects utDNA in urine.

             After:

              [Line 180] We concluded that most of the mutations detected in the urine samples by uAL100 were true positives, suggesting that uAL100 accurately detects utDNA in urine. Nevertheless, to obtain statistical significance of high mutational concordance between paired samples additional sample pairs should be assayed, which is one of our future tasks.

and in discussion section as follows:

             Before:

              [Line 340] First, our cohort was small, and further validation of our method with larger samples is needed.

             After:

[Line 340] First, our cohort was small, and further validation of our method with larger samples is needed. Specifically, more tumor and urine sample pairs are need to be analyzed to obtain statistical significance of mutational concordance study, since we only analyzed 7 sample pairs, which is a small number to provide statistical significance to the results.

Comment 5-9

- Line 271 - Previous urine-based methods – citations.

Authors’ response

We appreciate the Reviewer’s valuable suggestion. We added citations that correspond to previous urine-based methods in indicated lines (reference #10, 11) as follows:

             Before:

[Line 300] Previous urine-based methods for BC detection showed significant levels of false-positive signals with similar or lower sensitivity compared with our method.

After:

[Line 300] Previous urine-based methods using deep sequencing for BC detection showed significant levels of false-positive signals with similar or lower sensitivity compared with our method (Table 2) [10,11].

Additionally, we added a table that compared sensitivity and specificity of the methods (Table 2) after the discussion section as follows:

Comment 5-10

- Please revise limitations.

Authors’ response

We appreciate the Reviewer’s valuable suggestion. We revised the limitations of our method in the discussion section as follows:

             Before:

              [Line 339] We found that uAL100 performed well in detecting BC without tumor or blood samples; however, our study has several limitations. First, our cohort was small, and further validation of our method with larger samples is needed. Second, since the analysis pipe-line of uAL100 was originally developed for plasma samples, the error suppression may not be sufficient to remove all errors in urine DNA. Therefore, although the current version of error suppression in uAL100 is reliable, an error suppression method specifically for urine samples is needed to improve the accuracy of uAL100. Finally, since our target capture panel is not solely composed of genes related to BC, the cost of our method is not fully optimized for BC detection. Furthermore, some known BC-related genes, such as PLEKHS1, are not contained in the target capture panel. Consequently, it might be possible to in-crease the sensitivity for BC detection while reducing the error rate and cost by adjusting the gene panel.

             After:

[Line 339] We found that uAL100 performed well in detecting BC without tumor or blood samples; however, our study has several limitations. First, our cohort was small, and further validation of our method with larger samples is needed. Specifically, more tumor and urine sample pairs are need to be analyzed to obtain statistical significance of mutational concordance study, since we only analyzed 7 sample pairs, which is a small number to provide statistical significance to the results. Second, since the analysis pipeline of uAL100 was originally developed for plasma samples, the error suppression may not be sufficient to remove all errors in urine DNA. Therefore, although the current version of error suppression in uAL100 is reliable, an error suppression method specifically for urine samples is needed to improve the accuracy of uAL100. Third, since our target capture panel is not solely composed of genes related to BC, the cost of our method is not fully optimized for BC detection. Furthermore, some known BC-related genes, such as PLEKHS1, are not contained in the target capture panel. Consequently, it might be possible to in-crease the sensitivity for BC detection while reducing the error rate and cost by adjusting the gene panel. Finally, as we mentioned, surveillance of NMIBC patients is important due to their high recurrence rate. Therefore, we need to validate the surveillance ability of uAL100 to make our method more meaningful in real clinical situations.

Reviewer 2 Report

The authors present an interesting study of the detection of somatic mutation in urine samples from patients with bladder cancer using targeted deep sequencing. The manuscript is well-written and the statistical analysis is appropriate. I have only a couple of questions/comments that should be addressed.

Could you please indicate the coverage of the sequencing for each sample?

I suggest not including at the end of the introduction the main results of your results but instead a broad significance of the importance of this study

Author Response

May 13, 2023

Kiran Yu

Assistant Editor, Cancers

Dear Dr. Yu,

We are grateful to you and the Reviewers for your valuable comments and suggestions pertaining to our manuscript. We have taken all these comments into account and are submitting a revised version of our paper.

A full point-by-point response to the Reviewer’s comments is included below. We believe that incorporating the helpful suggestions from the Reviewers has substantially improved our manuscript, and it is now suitable for publication in Cancers.

Thank you for considering our revised manuscript. We look forward to hearing from you.

Kwang Hyun Kim, MD, Ph.D.

Department of Urology,

Ewha Womans University Seoul Hospital, Seoul 07804, Republic of Korea

Tel: +82-2-6986-1685

and

Duhee Bang, Ph.D.

Professor of Chemistry,

Yonsei University, Seoul 03722, Republic of Korea

Tel: +82-10-3357-0611

Reviewer Comments: Point-by-point response:

We thank the Reviewers for their constructive comments on our manuscript and recommendation of publication with modifications.

Listed below are our responses to the Reviewers’ suggestions for improvement, and the suggestions are grouped according to the Reviewer:

Reviewer #1

Comment 1-1

- In line 44, the author state that high VAFs are linked to false-positive. This seems counterintuitive. Can they briefly explain why this is the case?

Authors’ response

We appreciate the Reviewer’s valuable suggestion. We admitted that the pointed statement is counterintuitive since the indicated sentence is vague. Actually, we wrote the indicated sentence to imply the following claim: The variant allele frequencies of called mutations are high in urine samples from healthy donors as well as in urine samples from bladder cancer patients, which leads to misclassification of healthy donors to bladder cancer patients. Therefore, what we really wanted to claim was that high VAFs of mutations can be observed in urine DNA from healthy donors after stringent mutation calling, and we have observed such cases in previous studies (reference #11, 14). However, since this line can cause confusion for readers, as the Reviewer experienced, we decided to rephrase the indicated line by not mentioning “high VAF”, as follows:

Before:

[Line 51] Therefore, targeted deep sequencing of urine DNA is a promising approach for clinical applications in BC, although accurate detection of urine tumor DNA (utDNA) is challenging due to the presence of false-positive signals at high VAFs [11-14].

             After:

[Line 51] Therefore, targeted deep sequencing of DNA in supernatant or cell pellet of urine is a promising approach for clinical applications in BC, although accurate detection of BC is challenging due to the presence of significant levels of false-positive signals (sensitivity: 83-87% and specificity: 85~96%) [10,11,14,15].

Comment 1-2

- Lines 54-63 and Figure 1a, are more suited for methodology and result section. I suggest the authors leave only the first and final sentence here and move the other content elsewhere.

Authors’ response

We appreciate the Reviewer’s valuable suggestion. We have revised the indicated paragraph by replacing the sentences containing results with sentences that highlight the implications and significance of our method in clinical situations, as follows:

Before:

[Line 64] In this study, we describe uAL100, a urine-based BC detection method that accurately detects utDNA without matched tumor information. To validate the accuracy of uAL100, we applied uAL100 to 43 urine samples and 7 matched tumor samples from patients with BC and 21 urine samples from healthy donors (Figure 1a). We compared the mutations in the paired tumor and urine samples and observed a high level of mutational concordance within matched pairs. Specifically, 82.6% of the mutations identified in the urine samples were also identified in the matched tumor samples. The sensitivity and specificity of uAL100 for BC detection among the 43 patients and 21 controls were 83.7% and 100%, respectively. We confirmed that the mutations identified by uAL100 are significant in BC tumorigenesis and progression and have the potential to serve as biomarkers for BC.

After:

[Line 64] In this study, we describe uAL100, a urine-based BC detection method that accurately detects utDNA without matched tumor information. When applied to urine samples from our cohort, uAL100 successfully suppressed technical errors, resulting in a specificity of 100% for BC detection with a sensitivity of 83.7%. We expect that uAL100 will more reliably detect early BC than other urine based methods, thereby significantly reducing unnecessary tests for patients with hematuria. Also, uAL100 is expected to be helpful in making clinical decisions for patients with BC, since uAL100 provides genotypes of BC including mutations that are significant in BC tumorigenesis and progression. Collectively, we an-ticipate that uAL100 has the potential to greatly improve the efficiency and efficacy of BC treatment.

Also, we moved Figure 1a to the end of the result section “3.1. Patient Characteristics and Study Design” where it was first mentioned. Additionally, the sentences containing results were removed as they overlap with the contents in the result sections.

Comment 1-3

- Lines 103-104. I find the use of the term 'germline mutation' misleading. In my opinion the term 'polymorphism' would be clearer.

Authors’ response

We appreciate the Reviewer’s valuable suggestion. We admitted that the term ‘single nucleotide polymorphism (SNP) databases’ seems clearer than the term ‘germline mutation databases’ and changed indicated lines as follows:

Before:

[Line 112] Germline databases were used at the primary variant calling step to filter out putative germline mutations.

             After:

[Line 112] Single-nucleotide polymorphism databases were used at the primary variant calling step to filter out putative germline mutations.

Since we intended to remove only germline mutations (not somatic mutations), we decided not to change the term ‘germline mutation’ to ‘SNP’ in the other parts of the manuscripts.

Comment 1-4

- Line 120. I am not sure this is the best way to calculate mutational concordance as it provides two different results, one for the tumour samples and one for the urine sample. As concordance is a measure of how much the results of tumour and urine analysis agree with each other, I would have expected to be a single value.

Authors’ response

We appreciate the Reviewer’s valuable suggestion. We considered that both two values (the ratio of urine mutations detected in tumors and vice versa) necessary for the mutational concordance study as each value indicates positive predictive value and sensitivity, respectively. Therefore, we still want to present both values. However, revised the indicated line by changing the term ‘the mutational concordance level’ as follows:

Before:

[Line 127] The annotated SNVs and INDELs of all seven patients were then merged, and the mutational concordance level (detection ratio) was calculated using the following formula:

             After:

[Line 127] The annotated SNVs and INDELs of all seven patients were then merged, and the ratio of mutations of one given sample type that detected in paired sample type (detection ratio) was calculated using the following formula:

Comment 1-5

- Lines 125-127. The authors states that the mutation with the highest VAF in each patient was used to calculate sensitivity and specificity. Can the authors provide more details about how this was calculated?

Authors’ response

We appreciate the Reviewer’s valuable suggestion. We used ‘the highest VAF’ in the indicated lines when we investigated cutoff levels of VAF that give best sensitivity and specificity after applying all filters, which is not proper to be used in the indicated lines. Therefore, we revised the explanation of sensitivity and specificity without mentioning of “the highest VAF” as follows:

             Before:

              [Line 135] The mutation with the highest VAF in each patient was selected for calculation of the sensitivity and specificity of BC detection, and the respective VAFs were used as the percentage of utDNA in the entire sample of urine DNA. The sensitivity was the proportion of BC cases detected by uAL100 among 43 cases tested.

             After:

              [Line 135] The mutation with the highest VAF in each patient was selected for calculation of the sensitivity and specificity of BC detection, and the respective VAFs were used as the percentage of utDNA in the entire sample of urine DNA. The sensitivity was the proportion of BC cases having BC driver mutations in their urine DNA among 43 cases tested. The specificity was the proportion of the 21 controls that were correctly identified as not having BC driver mutations in their urine DNA.

Comment 1-6

- lines 151-154. I don't follow how these results were obtained from the formula in methods (line 120). Can the authors show the calculations? Also they state 'concordance was 90.5% for the tumor samples and 82.6% for the urine samples (Figure 2b) but in figure 2b the value reported are different from those stated.

Authors’ response

We appreciate the Reviewer’s valuable suggestion. This suggestion provided us with a valuable opportunity to correct our serious mistake. When we wrote the manuscript, we mistakenly typed the number of mutations for each group (“both tumor and urine”, “tumor only”, urine only”) in lines 151-154. In detail, the number of mutations for shared and urine only are 37 and 9, respectively. However, we mistakenly typed them as 38 and 8, respectively. Fortunately, there were no errors in the relevant figures (Figure 2a-c), so we corrected the indicated sentences as follows:

             Before:

[Line 167] Of the 50 mutations, 38 were shared between paired tumor and urine samples, whereas 12 were detected only in tumor samples or urine samples (tumor only = 4, urine only = 8; Figure 2a; Figure S2). As a result, the mutational concordance was 90.5% for the tumor samples and 82.6% for the urine samples (Figure 2b)

             After:

[Line 167] Of the 50 mutations, 37 were shared between paired tumor and urine samples, whereas 13 were detected only in tumor samples or urine samples (tumor only = 4, urine only = 9; Figure 2a; Figure S2). As a result, 90.2% of tumor mutations were also detected in paired urine samples, and 80.4% of urine mutations were also detected in paired tumor samples (Figure 2b).

Also, though now the Reviewer can deduce the same values using the formula in methods (line129) with the corrected numbers, we provide the specific calculations for deducing the detection ratio values as follows:

We sincerely appreciate this comment, and at the same time, are deeply reflecting on our critical mistake. We believe this valuable experience will be helpful in self-revision of our future publications.

Comment 1-7

- line 211. The mutation rate of some of these genes are much lower than expected based on mutation rates reported in tumour tissue (e.g. FGFR3). Can the authors comment on this in the discussion?

Authors’ response

We appreciate the Reviewer’s valuable suggestion. To answer this question, we compared the mutation rates in urine samples of our study and the mutation rates in bladder cancer tumor from a previous TCGA study [Figure 1 in Nature 507, 315–322 (2014)] with genes included in both the target genes of uAL100 and significantly mutated genes of the TCGA study. Contrary to the Reviewer’s concerns, we observed a high correlation of mutation rates between urine samples (uAL100) and bladder cancer tumor samples. Consequently, we added a relevant figure (Figure 4b) and a sentence in the result section “3.4. Bladder Cancer Genotypes Determined Using uAL100” as follows:

Before:

[Line 232] Other frequently mutated genes were TP53, ARID1A, PIK3CA, ERBB2, ERBB3, FGFR3, KDM6A, and TSC1 (> 10% of cases) that were observed as BC driver genes in previous studies (9, 24).

             After:

[Line 232] Other frequently mutated genes were TP53, ARID1A, PIK3CA, ERBB2, ERBB3, FGFR3, KDM6A, and TSC1 (> 10% of cases) that were observed as BC driver genes in previous studies [10,25]. Also, when compared to the TCGA study that investigated BC tumors, the frequency of cases with mutations in each gene detected by uAL100 showed a high correlation with those detected in the BC tumors reported in the TCGA study (R = 0.91; Figure 4b) [27], which indicates BC genotyping using uAL100 significantly reflects tumor information.

             Figure 4b

(b) Comparisons of mutated gene frequency between tumor samples (TCGA) and urine samples (uAL100). The analyzed genes consisted of those that overlapped between the target genes of uAL100 and the significantly mutated genes identified in TCGA. The correlation coefficient (R) corresponds to the Pearson correlation coefficient.

In the meanwhile, we are cautious about these results as they differ from what the Reviewer mentioned. Therefore, to provide a more profound answer to this question, we would like to know which previous studies the reviewer referred to.

Comment 1-8

- line 272. The authors state that their test perform better compared with others. Can they report and reference specificity and sensitivity for the other tests to support their claim?

Authors’ response

We appreciate the Reviewer’s valuable suggestion. We added a table that compares the sensitivity and specificity of urine-based methods that utilize deep sequencing (Table 2) after the discussion section, as follows:

Also, we added citations for the mentioned urine-based methods to the indicated lines (reference #9, 12), as follows:

             Before:

[Line 300] Previous urine-based methods for BC detection showed significant levels of false-positive signals with similar or lower sensitivity compared with our method.

After:

[Line 300] Previous urine-based methods using deep sequencing for BC detection showed significant levels of false-positive signals with similar or lower sensitivity compared with our method (Table 2) [10,11].

Comment 1-9

- line 275. Can the author add a comment regarding the feasibility of this type of testing (costs, time etc) for routine analysis within the health system (e.g. to replace cystoscopy)?

Authors’ response

We appreciate the Reviewer’s valuable suggestion. To respond to this comment, we merged the first and second paragraphs in the discussion section in to one paragraph and added a new paragraph that discusses about this comment, as follows:

             Before:

[Line 294] In this study, we described uAL100 and validated its performance in BC diagnosis. There was high concordance between mutations detected by uAL100 in matched urine and tumor samples, although the number of matched pairs was only seven in our cohorts. This indicated that the mutations detected in urine by uAL100 were derived from BC tumors. We then showed that uAL100 had 83.7% sensitivity and 100% specificity in detecting BC without tumor samples, suggesting that our method has high accuracy regardless of the tumor stage. Previous urine-based methods for BC detection showed significant levels of false-positive signals with similar or lower sensitivity compared with our method. A genotyping analysis showed that the mutations identified by uAL100 were highly related to BC tumorigenesis.

To increase convenience and reduce cost, we designed uAL100 to detect tumor DNA in urine without additional analysis of PBMCs from patients. We showed that accurate utDNA detection was possible without PBMCs; however, several mutations were removed by the germline filter in 15 of the 36 patients that were diagnosed with BC, so it is possible that the germline filter can lead to imperfect clinical decision making based on the mutation profiles that it produces. Therefore, although the germline filter had little effect on the sensitivity of BC detection without tumor samples, information from PBMCs may still be needed to support optimal clinical decisions.

             After:

[Line 294] In this study, we described uAL100 and validated its performance in BC diagnosis. There was high concordance between mutations detected by uAL100 in matched urine and tumor samples, although the number of matched pairs was only seven in our cohorts. This indicated that the mutations detected in urine by uAL100 were derived from BC tumors. We then showed that uAL100 had 83.7% sensitivity and 100% specificity in detecting BC without tumor samples, suggesting that our method has high accuracy regardless of the tumor stage. Previous urine-based methods using deep sequencing for BC detection showed significant levels of false-positive signals with similar or lower sensitivity com-pared with our method (Table 2) [10,11]. A genotyping analysis showed that the mutations identified by uAL100 were highly related to BC tumorigenesis. Additionally, we showed that accurate utDNA detection was possible without PBMCs which enables entire non-invasiveness of uAL100. However, since several mutations were removed by the germline filter in some patients that were diagnosed with BC, so it is possible that the germline filter can lead to imperfect clinical decision making based on the mutation pro-files that it produces. Therefore, although the germline filter had little effect on the sensitivity of BC detection without tumor samples, information from PBMCs may still be needed to support optimal clinical decisions.

For the clinical utilization of liquid biopsy for bladder cancer diagnosis, factors such as cost-effectiveness, test standardization and diagnostic accuracy should be taken into consideration. In this study, we described the process of sample preparation, DNA extraction and genomic analysis for our test and validated the diagnostic accuracy of uAL100 for bladder cancer. During the validation of uAL100, we showed that uAL100 accurately detects utDNA without matched PBMCs, which enables an increase in convenience and a reduction in cost. Regarding cost-effectiveness, although sequencing cost is decreasing, the cost for sequencing may still be more expensive than conventional urine tests for bladder cancer. Therefore, we do not expect uAL100 to fully replace cystoscopy which is the standard procedure for the diagnosis of bladder cancer. However, uAL100 has high diagnostic accuracy, and we expect that uAL100 could identify patients who needs cystoscopy and reduce the number of unnecessary procedures in patients who may have undergone invasive procedures without our test

Reviewer #2

Comment 2-1

-Could you please indicate the coverage of the sequencing for each sample?

Authors’ response

We appreciate the Reviewer’s valuable suggestion. We presented the median sequencing coverage of each sample as Table S2 and added sentences describing the median sequencing coverage values in the method section “2.2. Analysis of DNA from Urine and Bladder Tumors”, as follows:

             Before:

[Line 96] The target-enriched DNA libraries were sequenced using the Illumina Novaseq 6000 plat-form (Illumina, USA) to create 150 bp paired-end reads.

             After:

[Line 96] The target-enriched DNA libraries were sequenced using the Illumina Novaseq 6000 plat-form (Illumina, USA) to create 150 bp paired-end reads, with median sequencing depths of 6094× and 7732× for targeted regions in urine and tumor samples, respectively (Table S2).

Comment 2-2

- I suggest not including at the end of the introduction the main results of your results but instead a broad significance of the importance of this study

Authors’ response

We appreciate the Reviewer’s valuable suggestion. We have revised the indicated paragraph by replacing the sentences containing results with sentences that highlight the implications and significance of our method in clinical situations, as follows:

Before:

[Line 64] In this study, we describe uAL100, a urine-based BC detection method that accurately detects utDNA without matched tumor information. To validate the accuracy of uAL100, we applied uAL100 to 43 urine samples and 7 matched tumor samples from patients with BC and 21 urine samples from healthy donors (Figure 1a). We compared the mutations in the paired tumor and urine samples and observed a high level of mutational concordance within matched pairs. Specifically, 82.6% of the mutations identified in the urine samples were also identified in the matched tumor samples. The sensitivity and specificity of uAL100 for BC detection among the 43 patients and 21 controls were 83.7% and 100%, respectively. We confirmed that the mutations identified by uAL100 are significant in BC tumorigenesis and progression and have the potential to serve as biomarkers for BC.

After:

[Line 64] In this study, we describe uAL100, a urine-based BC detection method that accurately detects utDNA without matched tumor information. When applied to urine samples from our cohort, uAL100 successfully suppressed technical errors, resulting in a specificity of 100% for BC detection with a sensitivity of 83.7%. We expect that uAL100 will more reliably detect early BC than other urine based methods, thereby significantly reducing unnecessary tests for patients with hematuria. Also, uAL100 is expected to be helpful in making clinical decisions for patients with BC, since uAL100 provides genotypes of BC including mutations that are significant in BC tumorigenesis and progression. Collectively, we an-ticipate that uAL100 has the potential to greatly improve the efficiency and efficacy of BC treatment.

Also, we moved Figure 1a to the end of the result section “3.1. Patient Characteristics and Study Design” where it was first mentioned. Additionally, the sentences containing results were removed as they overlap with the contents in the result sections.

Reviewer #3

Comment 3-1, 3-2

- Line 54-63: This paragraph should be rewritten and focus on the objective of the work. Elements that are not part of the discussion are developed.

- Figure 1 should be removed and placed in the materials and methods and/or results part.

Authors’ response

We appreciate the Reviewer’s valuable suggestion. We have revised the indicated paragraph by replacing the sentences containing results with sentences that highlight the implications and significance of our method in clinical situations, as follows:

Before:

[Line 64] In this study, we describe uAL100, a urine-based BC detection method that accurately detects utDNA without matched tumor information. To validate the accuracy of uAL100, we applied uAL100 to 43 urine samples and 7 matched tumor samples from patients with BC and 21 urine samples from healthy donors (Figure 1a). We compared the mutations in the paired tumor and urine samples and observed a high level of mutational concordance within matched pairs. Specifically, 82.6% of the mutations identified in the urine samples were also identified in the matched tumor samples. The sensitivity and specificity of uAL100 for BC detection among the 43 patients and 21 controls were 83.7% and 100%, respectively. We confirmed that the mutations identified by uAL100 are significant in BC tumorigenesis and progression and have the potential to serve as biomarkers for BC.

After:

[Line 64] In this study, we describe uAL100, a urine-based BC detection method that accurately detects utDNA without matched tumor information. When applied to urine samples from our cohort, uAL100 successfully suppressed technical errors, resulting in a specificity of 100% for BC detection with a sensitivity of 83.7%. We expect that uAL100 will more reliably detect early BC than other urine based methods, thereby significantly reducing unnecessary tests for patients with hematuria. Also, uAL100 is expected to be helpful in making clinical decisions for patients with BC, since uAL100 provides genotypes of BC including mutations that are significant in BC tumorigenesis and progression. Collectively, we an-ticipate that uAL100 has the potential to greatly improve the efficiency and efficacy of BC treatment.

Also, we moved Figure 1a to the end of the result section “3.1. Patient Characteristics and Study Design” where it was first mentioned. Additionally, the sentences containing results were removed as they overlap with the contents in the result sections.

Comment 3-3

- How was the tumor material obtained? Was it obtained from paraffin-embedded material? Or was it obtained from fresh selection? If fresh material was used, was the presence of tumor confirmed? How was the selection carried out?

Authors’ response

We appreciate the Reviewer’s valuable suggestion. As stated in Material and Methods section, tumor sample was obtained at the time of surgery and stored as a fresh frozen sample. However, pathologic review for fresh frozen sample was not performed in this study.

Comment 3-4

- Why were only 7 paired cases studied? What were the clinicopathological characteristics of these seven patients. For example, pathological stage, histopathological grade, etc.

Authors’ response

We appreciate the Reviewer’s valuable suggestion. Sequencing of tumor DNA was performed only in the initial samples of this study, and 7 paired cases were not selected for this study. After sequencing of 7 tumor DNA for concordance analysis of tumor and urine DNA, we could determine the cutoff VAF for urine DNA for detecting bladder cancer with high sensitivity and specificity. As stated in the result section “3.2. Mutational Concordance in Paired Samples of Tumor and Urine DNA”, although tumor samples were analyzed only in 7 patients, most patients had non-muscle invasive bladder cancer. Therefore, we believe our method is reliable regardless of tumor stage. The patient characteristics of 7 patients are summarized in the table below and shown graphically in Figure 2a.

Comment 3-5

- Table 1. It is noteworthy that most of the cases included in the study were high-grade carcinomas, and that the cytological study was diagnostic in less than half of the cases. What classification system was used to assess the cytologies? Was the cytological material analyzed from a new sample or was the same material used for DNA extraction? Please detail.

Authors’ response

We appreciate the Reviewer’s valuable suggestion. Urine samples for cytological examination was obtained at the time of cystoscopy in clinics (different urine samples for DNA extraction) and the pathologist reported the results according to the Paris System [Acta Cytologica (2016) 60 (3): 185–197.]. If the result was reported as ‘suspicious for high-grade urothelial carcinoma’ or ‘high-grade urothelial carcinoma’, it was categorized as positive and in the case of negative for urothelial carcinoma or no adequate diagnosis possible, then it was categorized as negative. We briefly added a sentence stating about cytology in the method section “2.1. Patient Enrollment and Sample Collection” as follows:

             Before:

[Line 80] All patients provided informed consent for tissue banking and genetic testing.

             After:

[Line 80] All patients provided informed consent for tissue banking and genetic testing. Urine cytology was performed at the time of cystoscopy according to the Paris System [17].

Comment 3-6

- During the presentation of the results, reference is made to their discussion, including bibliographical references. This should be modified and clearly show the results of the study.

Authors’ response

We appreciate the Reviewer’s valuable suggestion. We suspect that the Reviewer raised this comment, especially with regards to the result sections “3.4. Bladder Cancer Genotypes Determined Using uAL100” or “3.5. Information from Peripheral Blood Mononuclear Cells in Urine-Based Bladder Cancer Detection”. Here, we would like to state the validity of using several citations in those result sections.

First, in the result section 3.4, we had to verify the significance of mutations identified by uAL100, and we determined that comparison with the results of previous studies were necessary. Since we analyzed small numbers of tumors (n=7), we believe that comparing our results with those of previous studies was the best way to validate our genotyping results.

For the result section 3.5, we wanted to claim that without matched PBMCs, accurate BC detection is possible using uAL100. However, we did not analyze matched PBMCs for direct validation of our hypothesis. Therefore, we had to indirectly validate our hypothesis, and for inferring processes, cited studies are needed. In summary due to the absence of matched PBMCs, we inferred conclusions that were largely dependent on results of previous studies we cited, which is one of the limitations of our study. While we mentioned this limitation in the end of the result section 3.5, we modified the relevant sentence to make it more clearer, as follows:

Before:

[Line 279] Based on these results, we suggest that PBMCs are not necessary for detecting BC using urine samples; however, further study is required with larger numbers of healthy controls.

After:

[Line 279] Based on these results, we suggest that PBMCs are not necessary for detecting BC using urine samples; however, further study is required with larger numbers of urine samples and matched PBMCs for precise validation of our hypothesis.

We hope that our presented validity can be positively reflected in reconsidering of the Reviewer’s opinions in comment 3-6.

Comment 3-7, 3-8

- The relevant findings of the study are not discussed, nor are the advantages and disadvantages taken into account.

- The discussion must be modified and the analysis supported with adequate bibliographical sources.

Authors’ response

We appreciate the Reviewer’s valuable suggestions. We determined to address both comment 3-7 and comment 3-8 simultaneously, as the Reviewer’s comment ‘The discussion must be modified’ contains both of these comments.

In comment 3-7, the Reviewer recommended to discuss about the advantages and disadvantages of our method. We considered that we have already presented advantages (enhanced accuracy and entirely non-invasiveness due to absence of PBMC) and disadvantages (loss of putative driver mutations due to absence of PBMC) in the first and second paragraphs of unrevised version of our manuscript. However, we also added the advantages of uAL100 in the middle of discussion section in terms of capturing genomic landscape of NMIBC for clinical decisions, since accurate genotyping of NMIBC is difficult due to tumor cell heterogeneity:

[Line 324] As in our cohort, NMIBC is the most common type of bladder cancer at the time of diagnosis. However, unlike similar clinical and histopathological characteristics of NMIBC tumors, their genomic landscapes are variable, resulting in largely different dis-ease progression and response to treatment [33,34]. Consequently, about 50-70% of NMIBC patients experience recurrence and progression to MIBC, necessitating continuous cystoscopy and diverse therapeutic interventions [35,36]. Therefore, accurate capture of the genomic landscape of NMIBC, including its tumor cell heterogeneity, is necessary for the complete cancer remission in early stages of the disease. However, the amount of released tumor-derived materials in bodily-fluids for NMIBC are small, and therefore reflecting integrative information of NMIBC is challenging. To overcome such challenges, we envision that uAL100, with its high accuracy, has the potential to capture genomic landscape of NMIBC containing information from subclonal or metastatic tumors at low levels. Thus, uAL100 is expected to be helpful in early clinical decisions and lifelong surveillance by preventing excessive cystoscopy and therapeutic intervention, which we need to validate in the future studies.

Also, as the Reviewer recommended, we added citations of previous methods when we compared our accuracy with them, as follows:

             Before:

[Line 300] Previous urine-based methods for BC detection showed significant levels of false-positive signals with similar or lower sensitivity compared with our method.

             After:

[Line 300] Previous urine-based methods using deep sequencing for BC detection showed significant levels of false-positive signals with similar or lower sensitivity compared with our method (Table 2) [10,11].

Additionally, we added a table that compared sensitivity and specificity of the methods (Table 2) after discussion section, as follows:

Finally, we added more limitations in terms of low numbers of tumor and urine sample pairs and absence of surveillance study of uAL100 to the final paragraph of the discussion section where we presented the limitations of our study, as follows:

             Before:

[Line 339] We found that uAL100 performed well in detecting BC without tumor or blood samples; however, our study has several limitations. First, our cohort was small, and further validation of our method with larger samples is needed. Second, since the analysis pipe-line of uAL100 was originally developed for plasma samples, the error suppression may not be sufficient to remove all errors in urine DNA. Therefore, although the current version of error suppression in uAL100 is reliable, an error suppression method specifically for urine samples is needed to improve the accuracy of uAL100. Finally, since our target capture panel is not solely composed of genes related to BC, the cost of our method is not fully optimized for BC detection. Furthermore, some known BC-related genes, such as PLEKHS1, are not contained in the target capture panel. Consequently, it might be possible to in-crease the sensitivity for BC detection while reducing the error rate and cost by adjusting the gene panel.

             After:

[Line 339] We found that uAL100 performed well in detecting BC without tumor or blood samples; however, our study has several limitations. First, our cohort was small, and further validation of our method with larger samples is needed. Specifically, more tumor and urine sample pairs are need to be analyzed to obtain statistical significance of mutational concordance study, since we only analyzed 7 sample pairs, which is a small number to provide statistical significance to the results. Second, since the analysis pipeline of uAL100 was originally developed for plasma samples, the error suppression may not be sufficient to re-move all errors in urine DNA. Therefore, although the current version of error suppression in uAL100 is reliable, an error suppression method specifically for urine samples is needed to improve the accuracy of uAL100. Third, since our target capture panel is not solely composed of genes related to BC, the cost of our method is not fully optimized for BC detection. Furthermore, some known BC-related genes, such as PLEKHS1, are not contained in the target capture panel. Consequently, it might be possible to increase the sensitivity for BC detection while reducing the error rate and cost by adjusting the gene panel. Finally, as we mentioned, surveillance of NMIBC patients is important due to their high recurrence rate. Therefore, we need to validate the surveillance ability of uAL100 to make our method more meaningful in real clinical situations.

With mentioned modifications, we believe that we satisfied the Reviewer’s recommendation ‘The discussion must be modified’.

Comment 3-9

- Finally, the references do not have the format required by the journal.

Authors’ response

We appreciate the Reviewer’s valuable suggestion. We changed the style of reference that satisfies criteria of Cancers.

Reviewer #4

Comment 4-1

- How many authors are sure about the accuracy of the screening, especially because urine-based method detection has significant levels of false-positive signals? During disease progression, individual disseminated tumor cells and consecutively metastases can acquire characteristics that do not match those of the corresponding primary tumors, and often are only hardly assessable for further evaluation.

Authors’ response

We appreciate the Reviewer’s valuable suggestion. We would like to clarify that all authors who reviewed the results of bladder cancer screening using urine DNA (including all correspondence and first authors) agreed on the accuracy of our method for BC detection, even though the number of samples we assayed was not large enough. As described in the manuscript, uAL100 was developed based on AL100, which was developed for blood-based cancer screening and AL100 showed significantly low levels of false positive signals in our previous study [Br J Cancer 127, 898–907 (2022)]. When we directly applied AL100 to urine DNA, we observed several significant false positive mutations in healthy control groups, as Reviewer indicated. To remove mutations in healthy control groups, we increased the cutoff level of variant allele frequency from 0.1-0.5% to 1.5% and utilized several oncogene databases, which were the main differences. As a result, we were able to remove false positive mutations in healthy control groups and, through mentioned developing processes, attain certainty in our results.

Regarding the latter statement of this comment, we agreed that obtaining metastatic tumor-derived materials is challenging, especially when metastasis occurred outside of urinary tract. However, we anticipate that metastatic tumor-derived materials can flow in urines and eventually be detected by sampling urine. However, this hypothesis will require further studies to be validated.

Comment 4-2

- Author should describe more about the Alpha Liquid technology if it is only for screening Circulating tumor DNA (ctDNA), how is this platform developed, and other details in the manuscript. This information will be useful for other scientists and clinicians.

Authors’ response

We appreciate the Reviewer’s valuable suggestion. The detailed description about the AL100 was presented in our previous study [Br J Cancer 127, 898–907 (2022)], and we referred this study in the introduction section of our manuscript. Also, another study that utilized blood-based AL100 is preparing for publication, we anticipate that other scientists and clinicians can more easily find and understand principle and properties of AL100.

Comment 4-3, 4-4, 4-6

- Does this technology have a patent, or anyone can use it?

- How much feasible is this technology for the common lab to use?

- Also author should mention the real clinical scenario with reflect on uAL100 technology and whether it has already been approved using blood-serum-based screening for cancer patients such as colorectal or others.

Authors’ response

We appreciate the Reviewer’s valuable suggestion. Here, we would like to reply to comments 4-3, 4-4, and 4-6 simultaneously, since those comments seems closely related. IMBdx, one of the affiliations of authors of this manuscript, is one of the main group (company) that developed the Alpha Liquid technology and has a commercial license for the blood-based AL100 technology. IMBdx is a distributor of the Alpha Liquid technology and provides it in several Asian countries as well as South Korea. Also, IMBdx is preparing to distribute the technology to western countries. As a result, we anticipate that both AL100 and uAL100 can be utilized wordwide, even in common labs.

Regarding the response to comment 4-6, we added a new paragraph in the discussion section, as follows:

[Line 311] For the clinical utilization of liquid biopsy for bladder cancer diagnosis, factors such as cost-effectiveness, test standardization and diagnostic accuracy should be taken into consideration. In this study, we described the process of sample preparation, DNA extraction and genomic analysis for our test and validated the diagnostic accuracy of uAL100 for bladder cancer. During the validation of uAL100, we showed that uAL100 accurately detects utDNA without matched PBMCs, which enables an increase in convenience and a reduction in cost. Regarding cost-effectiveness, although sequencing cost is decreasing, the cost for sequencing may still be more expensive than conventional urine tests for bladder cancer. Therefore, we do not expect uAL100 to fully replace cystoscopy which is the standard procedure for the diagnosis of bladder cancer. However, uAL100 has high diagnostic accuracy, and we expect that uAL100 could identify patients who needs cystoscopy and reduce the number of unnecessary procedures in patients who may have undergone invasive procedures without our test.

In the case of the approval of blood-serum-based cancer screening of AL100, AL100 is already utilized in 23 major cancer centers in South Korea, and we mentioned above, we are preparing approval and commercialization of AL100 in several other countries. However, we determined that these statements are not proper to mention in the manuscript and decided not to mention them.

Comment 4-5

- Indeed, screening BC patients using liquid biopsy samples such as urine is an effective effort. However, the mirror image of tumor cell heterogeneity and real-time reflection in the genomic landscape for non-muscle invasive cancer is difficult. The author’s viewpoint is important in this regard and may include in the conclusion section of the manuscript.

Authors’ response

We appreciate the Reviewer’s valuable suggestion. We agree that obtaining the entire genomic landscape of NMIBC is crucial for making proper clinical decisions and achieving consequent complete remission of the disease. Also, we agree that reflecting information from the whole clones of NMIBC tumors is challenging due to the severely low levels of subclonal-tumor-derived materials in urine. Nevertheless, we believe that still liquid biopsy has advantageous in obtaining integrative tumor information, particularly when considering tumor metastasis and ease of serial sampling. With its high accuracy, we expect that uAL100 has the potential to reflect entire tumor heterogeneity with further developments. Also, as we mentioned earlier, real-time acquisition of genomic landscapes may be obtainable with serial sampling of urine at different time points. We added a new paragraph in the discussion section to address this comment in more detail, as follows:

              [Line 324] As in our cohort, NMIBC is the most common type of bladder cancer at the time of diagnosis. However, unlike similar clinical and histopathological characteristics of NMIBC tumors, their genomic landscapes are variable, resulting in largely different dis-ease progression and response to treatment [33,34]. Consequently, about 50-70% of NMIBC patients experience recurrence and progression to MIBC, necessitating continuous cystoscopy and diverse therapeutic interventions [35,36]. Therefore, accurate capture of the genomic landscape of NMIBC, including its tumor cell heterogeneity, is necessary for the complete cancer remission in early stages of the disease. However, the amount of released tumor-derived materials in bodily-fluids for NMIBC are small, and therefore reflecting integrative information of NMIBC is challenging. To overcome such challenges, we envision that uAL100, with its high accuracy, has the potential to capture genomic landscape of NMIBC containing information from subclonal or metastatic tumors at low levels. Thus, uAL100 is expected to be helpful in early clinical decisions and lifelong surveillance by preventing excessive cystoscopy and therapeutic intervention, which we need to validate in the future studies.

Reviewer #5

Comment 5-1

- The introduction could be a little longer. In the introduction, I would suggest taking information from a recently published article within MDPI (e.g. 10.3390/ijms232113206 or a different analysis) reviewing FDA-approved tests along with many others including ncRNA and cfDNA, and state the actual poor sensitivities and specificities. – also, in this case, lines 37-38 with “… however, and …” is a poor choice of phrasing please rephrase.

Authors’ response

We appreciate the Reviewer’s valuable suggestion. In this comment, the Reviewer recommended revising the introduction section with several opinions: 1) lengthening the introduction section, 2) stating additional urine-based BC detection methods, 3) stating the actual sensitivity and specificity, and 4) rephrasing lines 37-38. We revised the introduction section by reflecting the Reviewer’s opinions with several additional changes as follows:

             Before:

[Page 1, Introduction] Hematuria is a prominent symptom in the early stages of bladder cancer (BC) and is present in 80–90% of patients with BC [1]. However, only 10–22% of patients with hematuria are eventually diagnosed with BC [2, 3]. Cystoscopy, the gold-standard method for detecting BC, is an invasive and costly procedure that carries risks of complication, and its accuracy is highly dependent on the skills of the operator. Therefore, there is a need for non-invasive tests to detect BC with reliable accuracy.

Several non-invasive urine-based tests have been developed to complement invasive cystoscopy for BC diagnosis (e.g. urine cytology, UroVysion, and NMP22) [4-6]. These tests have poor sensitivity, however, and do not provide genotyping information to aid clinical decisions [5, 6]. As an alternative, targeted deep sequencing of cell-free DNA (cfDNA) in blood or urine can detect genomic mutations at low allele frequencies, enabling early diagnosis of BC [7-10]. In patients with BC, DNA in urine has been reported to contain more mutations with higher variant allele frequencies (VAFs) than cfDNA in blood [10]. There-fore, targeted deep sequencing of urine DNA is a promising approach for clinical applications in BC, although accurate detection of urine tumor DNA (utDNA) is challenging due to the presence of false-positive signals at high VAFs [11-14].

In a previous study, we detected circulating tumor DNA in the blood of patients with colorectal cancer using our AlphaLiquid100 target capture panel and its paired analysis pipeline, which we refer to collectively as AL100 [15]. The analysis pipeline suppresses errors in deep sequencing data for 118 cancer-associated gene regions and detects sin-gle-nucleotide variants (SNVs), insertions/deletions (INDELs), copy-number variations (CNVs), and gene fusions with VAFs as low as 0.1%. The AlphaLiquid100 panel includes genes associated with BC, so we decided to utilize AL100 to accurately detect utDNA for the diagnosis of BC.

In this study, we describe uAL100, a urine-based BC detection method that accurately detects utDNA without matched tumor information. To validate the accuracy of uAL100, we applied uAL100 to 43 urine samples and 7 matched tumor samples from patients with BC and 21 urine samples from healthy donors (Figure 1a). We compared the mutations in the paired tumor and urine samples and observed a high level of mutational concordance within matched pairs. Specifically, 82.6% of the mutations identified in the urine samples were also identified in the matched tumor samples. The sensitivity and specificity of uAL100 for BC detection among the 43 patients and 21 controls were 83.7% and 100%, respectively. We confirmed that the mutations identified by uAL100 are significant in BC tumorigenesis and progression and have the potential to serve as biomarkers for BC.

             After:

[Page 1, Introduction] Hematuria is a prominent symptom in the early stages of bladder cancer (BC) and is present in 80–90% of patients with BC [1]. However, only 10–22% of patients with hematuria are eventually diagnosed with BC [2,3]. Cystoscopy, the gold-standard method for detecting BC, is an invasive and costly procedure that carries risks of complication, and its accuracy is highly dependent on the skills of the operator. Therefore, there is a need for non-invasive tests to detect BC with reliable accuracy.

Several non-invasive urine-based tests have been developed to complement invasive cystoscopy for BC diagnosis utilizing materials in urine (e.g. urine cytology, BTA, UroVysion, uCyt+, and NMP22) [4-7]. However, these tests have poor accuracy in detecting BC, with sensitivities between 56% and 83% and specificities between 64% and 88%. Additionally, these urine-based tests do not provide genotyping information to aid clinical decisions [5,6]. Therefore, there is a need for the development of alternative non-invasive methods with enhanced accuracy and genetic information of BC.

As alternative non-invasive methods for detecting BC, targeted deep sequencing of cell-free DNA (cfDNA) in blood or DNA in urine (supernatant or cell pellet) have been developed, as these methods can detect genomic mutations at low allele frequencies, enabling diagnosis and genotyping of BC in early stages of the disease [8-12]. Especially, deep sequencing of urine is more advantageous than blood in BC detection since urine contains relatively more enriched tumor-derived materials than blood due to a lower amount of leukocyte-derived materials in patients with BC [12,13]. Additionally, since urine can be sampled entirely non-invasively with practically unlimited sample volume, urine sampling is more advantageous than blood sampling. Therefore, targeted deep sequencing of DNA in supernatant or cell pellet of urine is a promising approach for clinical applications in BC, although accurate detection of BC is challenging due to the presence of significant levels of false-positive signals (sensitivity: 83-87% and specificity: 85~96%) [10,11,14,15].

In a previous study, we detected circulating tumor DNA in the blood of patients with colorectal cancer using our AlphaLiquid100 target capture panel and its paired analysis pipeline, which we refer to collectively as AL100 [16]. The analysis pipeline suppresses errors in deep sequencing data for 118 cancer-associated gene regions and detects single-nucleotide variants (SNVs), insertions/deletions (INDELs), copy-number variations (CNVs), and gene fusions with variant allele frequencies (VAFs) as low as 0.1%. The Al-phaLiquid100 panel includes genes associated with BC, so we decided to utilize AL100 to accurately detect urine tumor DNA (utDNA) in urine supernatant for the diagnosis of BC.

In this study, we describe uAL100, a urine-based BC detection method that accurately detects utDNA without matched tumor information. When applied to urine samples from our cohort, uAL100 successfully suppressed technical errors, resulting in a specificity of 100% for BC detection with a sensitivity of 83.7%. We expect that uAL100 will more reliably detect early BC than other urine based methods, thereby reducing unnecessary tests for patients with hematuria. Also, uAL100 is expected to be helpful in making clinical decisions for patients with BC since uAL100 provides genotypes of BC including mutations that are significant in BC tumorigenesis and progression. Collectively, we anticipate that uAL100 has the potential to greatly improve the efficiency and efficacy of BC detection and treatment.

We cited a journal that the Reviewer recommended as #Reference 7, and added two more methods, BTA and uCyto+. Also, we stated actual sensitivity and specificity levels of previous methods, and added the advantages of deep sequencing of urine than blood. Finally, we revised last paragraph by replacing results with implication and significance of uAL100.

Comment 5-2

- How cost-effective would your method of diagnosis be compared to today’s golden standard?

Authors’ response

If we consider the golden standard in terms of liquid biopsy methods, we would suggest the blood-based method of Guardant Health that is the most commercially advanced company in the field of liquid biopsy. When compared to Guardant Health’s method, the cost of our method is 1/5 (>$1,000). Therefore, we would like to claim that our method has cost competitiveness. However, we considered that cost comparison between blood-based method is not proper to be discussed in our study, we omitted such a comparison.

In the meanwhile, urine-based commercial non-invasive BC detection methods, such as the ImmunoCyt test of Scimedx, the nuclear matrix protein 22 (NMP22) immunoassay test of Matritech, and multi-target FISH (UroVysion), may be options for BC detection with lower costs than uAL100. However, as we mentioned in our manuscript, since the accuracy of these methods is poor and genotyping of bladder cancer is not available, we consider price comparison with these methods inappropriate.

Comment 5-3

- What was the VAF distribution in the study? Why did you choose this germline filter? Subjects under 40, 40-60, …, above 95 and were there any subjects outside of ranges?

Authors’ response

We appreciate the Reviewer’s valuable suggestion. For the removal of putative germline mutations, we removed mutations with VAFs between 40%-60% or >95%. Although theoretical VAFs of germline mutations are 50% or 100%, we broadened the ranges to sufficiently remove germline mutations since germline mutations showed VAFs between 40%-60% or >95% in real situations, when we analyzed PBMCs in our other studies which we have not published. As we mentioned in the result section “3.5. Information from Peripheral Blood Mononuclear Cells in Urine-Based Bladder Cancer Detection” and the discussion section, VAFs of several mutations were included in the 40%-60% or >95% range and excluded from the final result. Nevertheless, due to this germline filter, the sensitivity was not decreased while the specificity increased. Therefore, we concluded that this germline filter is effective in preventing healthy donors from being classified as bladder cancer patients while preserving sensitivity. However, more urine samples and paired PBMCs need to be analyzed to more profoundly test this germline filter, as discussed in the result section 3.5. In our cohort, we could not observe any subjects outside of the ranges, which was inferred from the absence of any detected mutations in healthy donors.

Comment 5-4

- Line 122-123 T-tests are used to compare or determine the differences between groups.

Authors’ response

We appreciate the Reviewer’s valuable suggestion. We corrected explanation of T-test as the reviewer commented, and changed indicated line, as follows:

             Before:

              [Line 132] Comparisons between two groups were performed using t-tests. ANOVA was used for comparisons of more than two groups.

             After:

[Line 132] T-test was used to test for differences between two groups, and ANOVA was used for more than two groups.

Comment 5-5

- Line 138-139 are there any differences in results for NMIBC and MIBC

Authors’ response

We appreciate the Reviewer’s valuable suggestion. There were differences in the frequency of mutated genes between NMIBC and MIBC, as shown in Figure 4b, although the number of samples from MIBC patients was relatively small compared to samples from NMIBC patients. We presented this result in the result section “3.4. Bladder Cancer Genotypes Determined Using uAL100” in our original manuscript. Also, the mutation counts and percentage of utDNA were higher in urines from MIBC patients than those from NMIBC patients, as shown in Figure 3b and d. However, in terms of sensitivity levels, NMIBC and MIBC were not significantly different (83.3% vs 85.7%). Since NMIBC and MIBC can be distinguished from cancer stages, we considered that presenting sensitivity levels according to tumor stage would sufficiently explain the sensitivity levels of NMIBC and MIBC. However, additional presentation of sensitivity levels in terms of histology would provide a broader perspective. Therefore, we added a relevant sentence in the result section “3.3. Bladder Cancer Detection without Tumor Samples”, as follows:

             Before:

[Line 199] In a subgroup analysis according to tumor stage, the sensitivity of detection in subgroups with stage Ta+Tis, T1, and T2–T4 disease was 71.4%, 90.9%, and 85.7%, respectively, with a specificity of 100% in each subgroup (Figure 3g).

After:

[Line 199] In a subgroup analysis according to tumor stage, the sensitivity of detection in subgroups with stage Ta+Tis, T1, and T2–T4 disease was 71.4%, 90.9%, and 85.7%, respectively, with a specificity of 100% in each subgroup (Figure 3g). When subgroups with stages that cor-respond to NMIBC were combined, the sensitivity of detection was 83.3%, which is similar to those of MIBC (T2-T4).

Comment 5-6

- Lines 156-157 specify the only one found mutation as this part gets distracting when reading.

Authors’ response

We appreciate the Reviewer’s valuable suggestion. We admitted that the indicated line could lead to confusion for readers, and we specified indicated mutation, as follows:

             Before:

[Line 172] Only one mutation was detected in the other patient, and that mutation was detected only in the urine DNA.

             After:

[Line 172] Only one mutation, TP53 M237I, was detected in the other patient, and that mutation was detected only in the urine DNA.

Comment 5-7

- Line 159-161 please rephrase for clarity.

Authors’ response

We appreciate the Reviewer’s valuable suggestion. We admitted that the indicated line can occur confusions for readers, and we changed those line clearly, as follows:

             Before:

              [Line 175] Among the mutations detected in the urine samples, those that were shared with matched tumor samples had higher VAFs than those that were not shared (median VAF = 17.18% vs. 2.06%; p = 0.025; Figure 2c).

After:

[Line 175] Within urine samples, the mutations that were shared with matched tumor samples had a higher median VAF than those that were not shared with matched tumor samples (median VAF = 17.18% vs. 2.06%; p = 0.025; Figure 2c).

Comment 5-8

- Line 163 comparison with 7 samples, ‘6 of 7’ positive is still a low statistical amount to state that these are truly positive. This is a considerable limitation that should be stated in the limitations.

Authors’ response

We appreciate the Reviewer’s valuable suggestion. We agreed with the Reviewer’s opinion and recognized that the low number of available paired samples is one of the limitations of our study when we wrote this manuscript. However, while we were focusing on the performance and limitations of uAL100 in terms of BC detection, we missed stating the limitation of our mutational concordance study. Therefore, we added a statement about the pointed limitation in the result section “3.2. Mutational Concordance in Paired Samples of Tumor and Urine DNA” after the indicated line, as follows:

             Before:

[Line 180] We concluded that most of the mutations detected in the urine samples by uAL100 were true positives, suggesting that uAL100 accurately detects utDNA in urine.

             After:

              [Line 180] We concluded that most of the mutations detected in the urine samples by uAL100 were true positives, suggesting that uAL100 accurately detects utDNA in urine. Nevertheless, to obtain statistical significance of high mutational concordance between paired samples additional sample pairs should be assayed, which is one of our future tasks.

and in discussion section as follows:

             Before:

              [Line 340] First, our cohort was small, and further validation of our method with larger samples is needed.

             After:

[Line 340] First, our cohort was small, and further validation of our method with larger samples is needed. Specifically, more tumor and urine sample pairs are need to be analyzed to obtain statistical significance of mutational concordance study, since we only analyzed 7 sample pairs, which is a small number to provide statistical significance to the results.

Comment 5-9

- Line 271 - Previous urine-based methods – citations.

Authors’ response

We appreciate the Reviewer’s valuable suggestion. We added citations that correspond to previous urine-based methods in indicated lines (reference #10, 11) as follows:

             Before:

[Line 300] Previous urine-based methods for BC detection showed significant levels of false-positive signals with similar or lower sensitivity compared with our method.

After:

[Line 300] Previous urine-based methods using deep sequencing for BC detection showed significant levels of false-positive signals with similar or lower sensitivity compared with our method (Table 2) [10,11].

Additionally, we added a table that compared sensitivity and specificity of the methods (Table 2) after the discussion section as follows:

Comment 5-10

- Please revise limitations.

Authors’ response

We appreciate the Reviewer’s valuable suggestion. We revised the limitations of our method in the discussion section as follows:

             Before:

              [Line 339] We found that uAL100 performed well in detecting BC without tumor or blood samples; however, our study has several limitations. First, our cohort was small, and further validation of our method with larger samples is needed. Second, since the analysis pipe-line of uAL100 was originally developed for plasma samples, the error suppression may not be sufficient to remove all errors in urine DNA. Therefore, although the current version of error suppression in uAL100 is reliable, an error suppression method specifically for urine samples is needed to improve the accuracy of uAL100. Finally, since our target capture panel is not solely composed of genes related to BC, the cost of our method is not fully optimized for BC detection. Furthermore, some known BC-related genes, such as PLEKHS1, are not contained in the target capture panel. Consequently, it might be possible to in-crease the sensitivity for BC detection while reducing the error rate and cost by adjusting the gene panel.

             After:

[Line 339] We found that uAL100 performed well in detecting BC without tumor or blood samples; however, our study has several limitations. First, our cohort was small, and further validation of our method with larger samples is needed. Specifically, more tumor and urine sample pairs are need to be analyzed to obtain statistical significance of mutational concordance study, since we only analyzed 7 sample pairs, which is a small number to provide statistical significance to the results. Second, since the analysis pipeline of uAL100 was originally developed for plasma samples, the error suppression may not be sufficient to remove all errors in urine DNA. Therefore, although the current version of error suppression in uAL100 is reliable, an error suppression method specifically for urine samples is needed to improve the accuracy of uAL100. Third, since our target capture panel is not solely composed of genes related to BC, the cost of our method is not fully optimized for BC detection. Furthermore, some known BC-related genes, such as PLEKHS1, are not contained in the target capture panel. Consequently, it might be possible to in-crease the sensitivity for BC detection while reducing the error rate and cost by adjusting the gene panel. Finally, as we mentioned, surveillance of NMIBC patients is important due to their high recurrence rate. Therefore, we need to validate the surveillance ability of uAL100 to make our method more meaningful in real clinical situations 

Reviewer 3 Report

Dear Editor,

I have reviewed the article by Lee D, et al., titled: "Accurate Detection of Urothelial Bladder Cancer Using Deep Targeted Urine DNA Sequencing." In this research study, the authors analyze patients with urinary bladder carcinoma and healthy controls, using the AlphaLiquid100 target capture panel to diagnose cancer. The study was based on 43 patients with carcinoma of the urinary bladder and 21 controls with benign pathology, considered as healthy controls. For the study, urine samples were analyzed and in seven patients they were correlated with tumor samples. The overall sensitivity was 83.7% with a specificity of 100% for the diagnosis of carcinoma. In general, the objective of the work can be interesting in urinary bladder carcinoma. However, the work requires extensive modifications before its possible publication. Here are my comments:

Point 1: Line 54-63: This paragraph should be rewritten and focus on the objective of the work. Elements that are not part of the discussion are developed

Point 2: Figure 1 should be removed and placed in the materials and methods and/or results part.

Point 3: Materials and methods:

How was the tumor material obtained? Was it obtained from paraffin-embedded material? Or was it obtained from fresh selection? If fresh material was used, was the presence of tumor confirmed? How was the selection carried out?.

Point 4: Materials and methods:

Why were only 7 paired cases studied? What were the clinicopathological characteristics of these seven patients. For example, pathological stage, histopathological grade, etc.

Point 5: Results:

 Table 1. It is noteworthy that most of the cases included in the study were high-grade carcinomas, and that the cytological study was diagnostic in less than half of the cases. What classification system was used to assess the cytologies?. Was the cytological material analyzed from a new sample or was the same material used for DNA extraction? Please detail.

Point 5: Results:

During the presentation of the results, reference is made to their discussion, including bibliographical references. This should be modified and clearly show the results of the study.

Point 6: Discussion:

The relevant findings of the study are not discussed, nor are the advantages and disadvantages taken into account.

Point 7: Discussion:

The discussion must be modified and the analysis supported with adequate bibliographical sources.

Finally, the references do not have the format required by the journal.

Author Response

May 13, 2023

Kiran Yu

Assistant Editor, Cancers

Dear Dr. Yu,

We are grateful to you and the Reviewers for your valuable comments and suggestions pertaining to our manuscript. We have taken all these comments into account and are submitting a revised version of our paper.

A full point-by-point response to the Reviewer’s comments is included below. We believe that incorporating the helpful suggestions from the Reviewers has substantially improved our manuscript, and it is now suitable for publication in Cancers.

Thank you for considering our revised manuscript. We look forward to hearing from you.

Kwang Hyun Kim, MD, Ph.D.

Department of Urology,

Ewha Womans University Seoul Hospital, Seoul 07804, Republic of Korea

Tel: +82-2-6986-1685

and

Duhee Bang, Ph.D.

Professor of Chemistry,

Yonsei University, Seoul 03722, Republic of Korea

Tel: +82-10-3357-0611

Reviewer Comments: Point-by-point response:

We thank the Reviewers for their constructive comments on our manuscript and recommendation of publication with modifications.

Listed below are our responses to the Reviewers’ suggestions for improvement, and the suggestions are grouped according to the Reviewer:

Reviewer #1

Comment 1-1

- In line 44, the author state that high VAFs are linked to false-positive. This seems counterintuitive. Can they briefly explain why this is the case?

Authors’ response

We appreciate the Reviewer’s valuable suggestion. We admitted that the pointed statement is counterintuitive since the indicated sentence is vague. Actually, we wrote the indicated sentence to imply the following claim: The variant allele frequencies of called mutations are high in urine samples from healthy donors as well as in urine samples from bladder cancer patients, which leads to misclassification of healthy donors to bladder cancer patients. Therefore, what we really wanted to claim was that high VAFs of mutations can be observed in urine DNA from healthy donors after stringent mutation calling, and we have observed such cases in previous studies (reference #11, 14). However, since this line can cause confusion for readers, as the Reviewer experienced, we decided to rephrase the indicated line by not mentioning “high VAF”, as follows:

Before:

[Line 51] Therefore, targeted deep sequencing of urine DNA is a promising approach for clinical applications in BC, although accurate detection of urine tumor DNA (utDNA) is challenging due to the presence of false-positive signals at high VAFs [11-14].

             After:

[Line 51] Therefore, targeted deep sequencing of DNA in supernatant or cell pellet of urine is a promising approach for clinical applications in BC, although accurate detection of BC is challenging due to the presence of significant levels of false-positive signals (sensitivity: 83-87% and specificity: 85~96%) [10,11,14,15].

Comment 1-2

- Lines 54-63 and Figure 1a, are more suited for methodology and result section. I suggest the authors leave only the first and final sentence here and move the other content elsewhere.

Authors’ response

We appreciate the Reviewer’s valuable suggestion. We have revised the indicated paragraph by replacing the sentences containing results with sentences that highlight the implications and significance of our method in clinical situations, as follows:

Before:

[Line 64] In this study, we describe uAL100, a urine-based BC detection method that accurately detects utDNA without matched tumor information. To validate the accuracy of uAL100, we applied uAL100 to 43 urine samples and 7 matched tumor samples from patients with BC and 21 urine samples from healthy donors (Figure 1a). We compared the mutations in the paired tumor and urine samples and observed a high level of mutational concordance within matched pairs. Specifically, 82.6% of the mutations identified in the urine samples were also identified in the matched tumor samples. The sensitivity and specificity of uAL100 for BC detection among the 43 patients and 21 controls were 83.7% and 100%, respectively. We confirmed that the mutations identified by uAL100 are significant in BC tumorigenesis and progression and have the potential to serve as biomarkers for BC.

After:

[Line 64] In this study, we describe uAL100, a urine-based BC detection method that accurately detects utDNA without matched tumor information. When applied to urine samples from our cohort, uAL100 successfully suppressed technical errors, resulting in a specificity of 100% for BC detection with a sensitivity of 83.7%. We expect that uAL100 will more reliably detect early BC than other urine based methods, thereby significantly reducing unnecessary tests for patients with hematuria. Also, uAL100 is expected to be helpful in making clinical decisions for patients with BC, since uAL100 provides genotypes of BC including mutations that are significant in BC tumorigenesis and progression. Collectively, we an-ticipate that uAL100 has the potential to greatly improve the efficiency and efficacy of BC treatment.

Also, we moved Figure 1a to the end of the result section “3.1. Patient Characteristics and Study Design” where it was first mentioned. Additionally, the sentences containing results were removed as they overlap with the contents in the result sections.

Comment 1-3

- Lines 103-104. I find the use of the term 'germline mutation' misleading. In my opinion the term 'polymorphism' would be clearer.

Authors’ response

We appreciate the Reviewer’s valuable suggestion. We admitted that the term ‘single nucleotide polymorphism (SNP) databases’ seems clearer than the term ‘germline mutation databases’ and changed indicated lines as follows:

Before:

[Line 112] Germline databases were used at the primary variant calling step to filter out putative germline mutations.

             After:

[Line 112] Single-nucleotide polymorphism databases were used at the primary variant calling step to filter out putative germline mutations.

Since we intended to remove only germline mutations (not somatic mutations), we decided not to change the term ‘germline mutation’ to ‘SNP’ in the other parts of the manuscripts.

Comment 1-4

- Line 120. I am not sure this is the best way to calculate mutational concordance as it provides two different results, one for the tumour samples and one for the urine sample. As concordance is a measure of how much the results of tumour and urine analysis agree with each other, I would have expected to be a single value.

Authors’ response

We appreciate the Reviewer’s valuable suggestion. We considered that both two values (the ratio of urine mutations detected in tumors and vice versa) necessary for the mutational concordance study as each value indicates positive predictive value and sensitivity, respectively. Therefore, we still want to present both values. However, revised the indicated line by changing the term ‘the mutational concordance level’ as follows:

Before:

[Line 127] The annotated SNVs and INDELs of all seven patients were then merged, and the mutational concordance level (detection ratio) was calculated using the following formula:

             After:

[Line 127] The annotated SNVs and INDELs of all seven patients were then merged, and the ratio of mutations of one given sample type that detected in paired sample type (detection ratio) was calculated using the following formula:

Comment 1-5

- Lines 125-127. The authors states that the mutation with the highest VAF in each patient was used to calculate sensitivity and specificity. Can the authors provide more details about how this was calculated?

Authors’ response

We appreciate the Reviewer’s valuable suggestion. We used ‘the highest VAF’ in the indicated lines when we investigated cutoff levels of VAF that give best sensitivity and specificity after applying all filters, which is not proper to be used in the indicated lines. Therefore, we revised the explanation of sensitivity and specificity without mentioning of “the highest VAF” as follows:

             Before:

              [Line 135] The mutation with the highest VAF in each patient was selected for calculation of the sensitivity and specificity of BC detection, and the respective VAFs were used as the percentage of utDNA in the entire sample of urine DNA. The sensitivity was the proportion of BC cases detected by uAL100 among 43 cases tested.

             After:

              [Line 135] The mutation with the highest VAF in each patient was selected for calculation of the sensitivity and specificity of BC detection, and the respective VAFs were used as the percentage of utDNA in the entire sample of urine DNA. The sensitivity was the proportion of BC cases having BC driver mutations in their urine DNA among 43 cases tested. The specificity was the proportion of the 21 controls that were correctly identified as not having BC driver mutations in their urine DNA.

Comment 1-6

- lines 151-154. I don't follow how these results were obtained from the formula in methods (line 120). Can the authors show the calculations? Also they state 'concordance was 90.5% for the tumor samples and 82.6% for the urine samples (Figure 2b) but in figure 2b the value reported are different from those stated.

Authors’ response

We appreciate the Reviewer’s valuable suggestion. This suggestion provided us with a valuable opportunity to correct our serious mistake. When we wrote the manuscript, we mistakenly typed the number of mutations for each group (“both tumor and urine”, “tumor only”, urine only”) in lines 151-154. In detail, the number of mutations for shared and urine only are 37 and 9, respectively. However, we mistakenly typed them as 38 and 8, respectively. Fortunately, there were no errors in the relevant figures (Figure 2a-c), so we corrected the indicated sentences as follows:

             Before:

[Line 167] Of the 50 mutations, 38 were shared between paired tumor and urine samples, whereas 12 were detected only in tumor samples or urine samples (tumor only = 4, urine only = 8; Figure 2a; Figure S2). As a result, the mutational concordance was 90.5% for the tumor samples and 82.6% for the urine samples (Figure 2b)

             After:

[Line 167] Of the 50 mutations, 37 were shared between paired tumor and urine samples, whereas 13 were detected only in tumor samples or urine samples (tumor only = 4, urine only = 9; Figure 2a; Figure S2). As a result, 90.2% of tumor mutations were also detected in paired urine samples, and 80.4% of urine mutations were also detected in paired tumor samples (Figure 2b).

Also, though now the Reviewer can deduce the same values using the formula in methods (line129) with the corrected numbers, we provide the specific calculations for deducing the detection ratio values as follows:

We sincerely appreciate this comment, and at the same time, are deeply reflecting on our critical mistake. We believe this valuable experience will be helpful in self-revision of our future publications.

Comment 1-7

- line 211. The mutation rate of some of these genes are much lower than expected based on mutation rates reported in tumour tissue (e.g. FGFR3). Can the authors comment on this in the discussion?

Authors’ response

We appreciate the Reviewer’s valuable suggestion. To answer this question, we compared the mutation rates in urine samples of our study and the mutation rates in bladder cancer tumor from a previous TCGA study [Figure 1 in Nature 507, 315–322 (2014)] with genes included in both the target genes of uAL100 and significantly mutated genes of the TCGA study. Contrary to the Reviewer’s concerns, we observed a high correlation of mutation rates between urine samples (uAL100) and bladder cancer tumor samples. Consequently, we added a relevant figure (Figure 4b) and a sentence in the result section “3.4. Bladder Cancer Genotypes Determined Using uAL100” as follows:

Before:

[Line 232] Other frequently mutated genes were TP53, ARID1A, PIK3CA, ERBB2, ERBB3, FGFR3, KDM6A, and TSC1 (> 10% of cases) that were observed as BC driver genes in previous studies (9, 24).

             After:

[Line 232] Other frequently mutated genes were TP53, ARID1A, PIK3CA, ERBB2, ERBB3, FGFR3, KDM6A, and TSC1 (> 10% of cases) that were observed as BC driver genes in previous studies [10,25]. Also, when compared to the TCGA study that investigated BC tumors, the frequency of cases with mutations in each gene detected by uAL100 showed a high correlation with those detected in the BC tumors reported in the TCGA study (R = 0.91; Figure 4b) [27], which indicates BC genotyping using uAL100 significantly reflects tumor information.

             Figure 4b

(b) Comparisons of mutated gene frequency between tumor samples (TCGA) and urine samples (uAL100). The analyzed genes consisted of those that overlapped between the target genes of uAL100 and the significantly mutated genes identified in TCGA. The correlation coefficient (R) corresponds to the Pearson correlation coefficient.

In the meanwhile, we are cautious about these results as they differ from what the Reviewer mentioned. Therefore, to provide a more profound answer to this question, we would like to know which previous studies the reviewer referred to.

Comment 1-8

- line 272. The authors state that their test perform better compared with others. Can they report and reference specificity and sensitivity for the other tests to support their claim?

Authors’ response

We appreciate the Reviewer’s valuable suggestion. We added a table that compares the sensitivity and specificity of urine-based methods that utilize deep sequencing (Table 2) after the discussion section, as follows:

Also, we added citations for the mentioned urine-based methods to the indicated lines (reference #9, 12), as follows:

             Before:

[Line 300] Previous urine-based methods for BC detection showed significant levels of false-positive signals with similar or lower sensitivity compared with our method.

After:

[Line 300] Previous urine-based methods using deep sequencing for BC detection showed significant levels of false-positive signals with similar or lower sensitivity compared with our method (Table 2) [10,11].

Comment 1-9

- line 275. Can the author add a comment regarding the feasibility of this type of testing (costs, time etc) for routine analysis within the health system (e.g. to replace cystoscopy)?

Authors’ response

We appreciate the Reviewer’s valuable suggestion. To respond to this comment, we merged the first and second paragraphs in the discussion section in to one paragraph and added a new paragraph that discusses about this comment, as follows:

             Before:

[Line 294] In this study, we described uAL100 and validated its performance in BC diagnosis. There was high concordance between mutations detected by uAL100 in matched urine and tumor samples, although the number of matched pairs was only seven in our cohorts. This indicated that the mutations detected in urine by uAL100 were derived from BC tumors. We then showed that uAL100 had 83.7% sensitivity and 100% specificity in detecting BC without tumor samples, suggesting that our method has high accuracy regardless of the tumor stage. Previous urine-based methods for BC detection showed significant levels of false-positive signals with similar or lower sensitivity compared with our method. A genotyping analysis showed that the mutations identified by uAL100 were highly related to BC tumorigenesis.

To increase convenience and reduce cost, we designed uAL100 to detect tumor DNA in urine without additional analysis of PBMCs from patients. We showed that accurate utDNA detection was possible without PBMCs; however, several mutations were removed by the germline filter in 15 of the 36 patients that were diagnosed with BC, so it is possible that the germline filter can lead to imperfect clinical decision making based on the mutation profiles that it produces. Therefore, although the germline filter had little effect on the sensitivity of BC detection without tumor samples, information from PBMCs may still be needed to support optimal clinical decisions.

             After:

[Line 294] In this study, we described uAL100 and validated its performance in BC diagnosis. There was high concordance between mutations detected by uAL100 in matched urine and tumor samples, although the number of matched pairs was only seven in our cohorts. This indicated that the mutations detected in urine by uAL100 were derived from BC tumors. We then showed that uAL100 had 83.7% sensitivity and 100% specificity in detecting BC without tumor samples, suggesting that our method has high accuracy regardless of the tumor stage. Previous urine-based methods using deep sequencing for BC detection showed significant levels of false-positive signals with similar or lower sensitivity com-pared with our method (Table 2) [10,11]. A genotyping analysis showed that the mutations identified by uAL100 were highly related to BC tumorigenesis. Additionally, we showed that accurate utDNA detection was possible without PBMCs which enables entire non-invasiveness of uAL100. However, since several mutations were removed by the germline filter in some patients that were diagnosed with BC, so it is possible that the germline filter can lead to imperfect clinical decision making based on the mutation pro-files that it produces. Therefore, although the germline filter had little effect on the sensitivity of BC detection without tumor samples, information from PBMCs may still be needed to support optimal clinical decisions.

For the clinical utilization of liquid biopsy for bladder cancer diagnosis, factors such as cost-effectiveness, test standardization and diagnostic accuracy should be taken into consideration. In this study, we described the process of sample preparation, DNA extraction and genomic analysis for our test and validated the diagnostic accuracy of uAL100 for bladder cancer. During the validation of uAL100, we showed that uAL100 accurately detects utDNA without matched PBMCs, which enables an increase in convenience and a reduction in cost. Regarding cost-effectiveness, although sequencing cost is decreasing, the cost for sequencing may still be more expensive than conventional urine tests for bladder cancer. Therefore, we do not expect uAL100 to fully replace cystoscopy which is the standard procedure for the diagnosis of bladder cancer. However, uAL100 has high diagnostic accuracy, and we expect that uAL100 could identify patients who needs cystoscopy and reduce the number of unnecessary procedures in patients who may have undergone invasive procedures without our test.

Reviewer #2

Comment 2-1

-Could you please indicate the coverage of the sequencing for each sample?

Authors’ response

We appreciate the Reviewer’s valuable suggestion. We presented the median sequencing coverage of each sample as Table S2 and added sentences describing the median sequencing coverage values in the method section “2.2. Analysis of DNA from Urine and Bladder Tumors”, as follows:

             Before:

[Line 96] The target-enriched DNA libraries were sequenced using the Illumina Novaseq 6000 plat-form (Illumina, USA) to create 150 bp paired-end reads.

             After:

[Line 96] The target-enriched DNA libraries were sequenced using the Illumina Novaseq 6000 plat-form (Illumina, USA) to create 150 bp paired-end reads, with median sequencing depths of 6094× and 7732× for targeted regions in urine and tumor samples, respectively (Table S2).

Comment 2-2

- I suggest not including at the end of the introduction the main results of your results but instead a broad significance of the importance of this study

Authors’ response

We appreciate the Reviewer’s valuable suggestion. We have revised the indicated paragraph by replacing the sentences containing results with sentences that highlight the implications and significance of our method in clinical situations, as follows:

Before:

[Line 64] In this study, we describe uAL100, a urine-based BC detection method that accurately detects utDNA without matched tumor information. To validate the accuracy of uAL100, we applied uAL100 to 43 urine samples and 7 matched tumor samples from patients with BC and 21 urine samples from healthy donors (Figure 1a). We compared the mutations in the paired tumor and urine samples and observed a high level of mutational concordance within matched pairs. Specifically, 82.6% of the mutations identified in the urine samples were also identified in the matched tumor samples. The sensitivity and specificity of uAL100 for BC detection among the 43 patients and 21 controls were 83.7% and 100%, respectively. We confirmed that the mutations identified by uAL100 are significant in BC tumorigenesis and progression and have the potential to serve as biomarkers for BC.

After:

[Line 64] In this study, we describe uAL100, a urine-based BC detection method that accurately detects utDNA without matched tumor information. When applied to urine samples from our cohort, uAL100 successfully suppressed technical errors, resulting in a specificity of 100% for BC detection with a sensitivity of 83.7%. We expect that uAL100 will more reliably detect early BC than other urine based methods, thereby significantly reducing unnecessary tests for patients with hematuria. Also, uAL100 is expected to be helpful in making clinical decisions for patients with BC, since uAL100 provides genotypes of BC including mutations that are significant in BC tumorigenesis and progression. Collectively, we an-ticipate that uAL100 has the potential to greatly improve the efficiency and efficacy of BC treatment.

Also, we moved Figure 1a to the end of the result section “3.1. Patient Characteristics and Study Design” where it was first mentioned. Additionally, the sentences containing results were removed as they overlap with the contents in the result sections.

Reviewer #3

Comment 3-1, 3-2

- Line 54-63: This paragraph should be rewritten and focus on the objective of the work. Elements that are not part of the discussion are developed.

- Figure 1 should be removed and placed in the materials and methods and/or results part.

Authors’ response

We appreciate the Reviewer’s valuable suggestion. We have revised the indicated paragraph by replacing the sentences containing results with sentences that highlight the implications and significance of our method in clinical situations, as follows:

Before:

[Line 64] In this study, we describe uAL100, a urine-based BC detection method that accurately detects utDNA without matched tumor information. To validate the accuracy of uAL100, we applied uAL100 to 43 urine samples and 7 matched tumor samples from patients with BC and 21 urine samples from healthy donors (Figure 1a). We compared the mutations in the paired tumor and urine samples and observed a high level of mutational concordance within matched pairs. Specifically, 82.6% of the mutations identified in the urine samples were also identified in the matched tumor samples. The sensitivity and specificity of uAL100 for BC detection among the 43 patients and 21 controls were 83.7% and 100%, respectively. We confirmed that the mutations identified by uAL100 are significant in BC tumorigenesis and progression and have the potential to serve as biomarkers for BC.

After:

[Line 64] In this study, we describe uAL100, a urine-based BC detection method that accurately detects utDNA without matched tumor information. When applied to urine samples from our cohort, uAL100 successfully suppressed technical errors, resulting in a specificity of 100% for BC detection with a sensitivity of 83.7%. We expect that uAL100 will more reliably detect early BC than other urine based methods, thereby significantly reducing unnecessary tests for patients with hematuria. Also, uAL100 is expected to be helpful in making clinical decisions for patients with BC, since uAL100 provides genotypes of BC including mutations that are significant in BC tumorigenesis and progression. Collectively, we an-ticipate that uAL100 has the potential to greatly improve the efficiency and efficacy of BC treatment.

Also, we moved Figure 1a to the end of the result section “3.1. Patient Characteristics and Study Design” where it was first mentioned. Additionally, the sentences containing results were removed as they overlap with the contents in the result sections.

Comment 3-3

- How was the tumor material obtained? Was it obtained from paraffin-embedded material? Or was it obtained from fresh selection? If fresh material was used, was the presence of tumor confirmed? How was the selection carried out?

Authors’ response

We appreciate the Reviewer’s valuable suggestion. As stated in Material and Methods section, tumor sample was obtained at the time of surgery and stored as a fresh frozen sample. However, pathologic review for fresh frozen sample was not performed in this study.

Comment 3-4

- Why were only 7 paired cases studied? What were the clinicopathological characteristics of these seven patients. For example, pathological stage, histopathological grade, etc.

Authors’ response

We appreciate the Reviewer’s valuable suggestion. Sequencing of tumor DNA was performed only in the initial samples of this study, and 7 paired cases were not selected for this study. After sequencing of 7 tumor DNA for concordance analysis of tumor and urine DNA, we could determine the cutoff VAF for urine DNA for detecting bladder cancer with high sensitivity and specificity. As stated in the result section “3.2. Mutational Concordance in Paired Samples of Tumor and Urine DNA”, although tumor samples were analyzed only in 7 patients, most patients had non-muscle invasive bladder cancer. Therefore, we believe our method is reliable regardless of tumor stage. The patient characteristics of 7 patients are summarized in the table below and shown graphically in Figure 2a.

Comment 3-5

- Table 1. It is noteworthy that most of the cases included in the study were high-grade carcinomas, and that the cytological study was diagnostic in less than half of the cases. What classification system was used to assess the cytologies? Was the cytological material analyzed from a new sample or was the same material used for DNA extraction? Please detail.

Authors’ response

We appreciate the Reviewer’s valuable suggestion. Urine samples for cytological examination was obtained at the time of cystoscopy in clinics (different urine samples for DNA extraction) and the pathologist reported the results according to the Paris System [Acta Cytologica (2016) 60 (3): 185–197.]. If the result was reported as ‘suspicious for high-grade urothelial carcinoma’ or ‘high-grade urothelial carcinoma’, it was categorized as positive and in the case of negative for urothelial carcinoma or no adequate diagnosis possible, then it was categorized as negative. We briefly added a sentence stating about cytology in the method section “2.1. Patient Enrollment and Sample Collection” as follows:

             Before:

[Line 80] All patients provided informed consent for tissue banking and genetic testing.

             After:

[Line 80] All patients provided informed consent for tissue banking and genetic testing. Urine cytology was performed at the time of cystoscopy according to the Paris System [17].

Comment 3-6

- During the presentation of the results, reference is made to their discussion, including bibliographical references. This should be modified and clearly show the results of the study.

Authors’ response

We appreciate the Reviewer’s valuable suggestion. We suspect that the Reviewer raised this comment, especially with regards to the result sections “3.4. Bladder Cancer Genotypes Determined Using uAL100” or “3.5. Information from Peripheral Blood Mononuclear Cells in Urine-Based Bladder Cancer Detection”. Here, we would like to state the validity of using several citations in those result sections.

First, in the result section 3.4, we had to verify the significance of mutations identified by uAL100, and we determined that comparison with the results of previous studies were necessary. Since we analyzed small numbers of tumors (n=7), we believe that comparing our results with those of previous studies was the best way to validate our genotyping results.

For the result section 3.5, we wanted to claim that without matched PBMCs, accurate BC detection is possible using uAL100. However, we did not analyze matched PBMCs for direct validation of our hypothesis. Therefore, we had to indirectly validate our hypothesis, and for inferring processes, cited studies are needed. In summary due to the absence of matched PBMCs, we inferred conclusions that were largely dependent on results of previous studies we cited, which is one of the limitations of our study. While we mentioned this limitation in the end of the result section 3.5, we modified the relevant sentence to make it more clearer, as follows:

Before:

[Line 279] Based on these results, we suggest that PBMCs are not necessary for detecting BC using urine samples; however, further study is required with larger numbers of healthy controls.

After:

[Line 279] Based on these results, we suggest that PBMCs are not necessary for detecting BC using urine samples; however, further study is required with larger numbers of urine samples and matched PBMCs for precise validation of our hypothesis.

We hope that our presented validity can be positively reflected in reconsidering of the Reviewer’s opinions in comment 3-6.

Comment 3-7, 3-8

- The relevant findings of the study are not discussed, nor are the advantages and disadvantages taken into account.

- The discussion must be modified and the analysis supported with adequate bibliographical sources.

Authors’ response

We appreciate the Reviewer’s valuable suggestions. We determined to address both comment 3-7 and comment 3-8 simultaneously, as the Reviewer’s comment ‘The discussion must be modified’ contains both of these comments.

In comment 3-7, the Reviewer recommended to discuss about the advantages and disadvantages of our method. We considered that we have already presented advantages (enhanced accuracy and entirely non-invasiveness due to absence of PBMC) and disadvantages (loss of putative driver mutations due to absence of PBMC) in the first and second paragraphs of unrevised version of our manuscript. However, we also added the advantages of uAL100 in the middle of discussion section in terms of capturing genomic landscape of NMIBC for clinical decisions, since accurate genotyping of NMIBC is difficult due to tumor cell heterogeneity:

[Line 324] As in our cohort, NMIBC is the most common type of bladder cancer at the time of diagnosis. However, unlike similar clinical and histopathological characteristics of NMIBC tumors, their genomic landscapes are variable, resulting in largely different dis-ease progression and response to treatment [33,34]. Consequently, about 50-70% of NMIBC patients experience recurrence and progression to MIBC, necessitating continuous cystoscopy and diverse therapeutic interventions [35,36]. Therefore, accurate capture of the genomic landscape of NMIBC, including its tumor cell heterogeneity, is necessary for the complete cancer remission in early stages of the disease. However, the amount of released tumor-derived materials in bodily-fluids for NMIBC are small, and therefore reflecting integrative information of NMIBC is challenging. To overcome such challenges, we envision that uAL100, with its high accuracy, has the potential to capture genomic landscape of NMIBC containing information from subclonal or metastatic tumors at low levels. Thus, uAL100 is expected to be helpful in early clinical decisions and lifelong surveillance by preventing excessive cystoscopy and therapeutic intervention, which we need to validate in the future studies.

Also, as the Reviewer recommended, we added citations of previous methods when we compared our accuracy with them, as follows:

             Before:

[Line 300] Previous urine-based methods for BC detection showed significant levels of false-positive signals with similar or lower sensitivity compared with our method.

             After:

[Line 300] Previous urine-based methods using deep sequencing for BC detection showed significant levels of false-positive signals with similar or lower sensitivity compared with our method (Table 2) [10,11].

Additionally, we added a table that compared sensitivity and specificity of the methods (Table 2) after discussion section, as follows:

Finally, we added more limitations in terms of low numbers of tumor and urine sample pairs and absence of surveillance study of uAL100 to the final paragraph of the discussion section where we presented the limitations of our study, as follows:

             Before:

[Line 339] We found that uAL100 performed well in detecting BC without tumor or blood samples; however, our study has several limitations. First, our cohort was small, and further validation of our method with larger samples is needed. Second, since the analysis pipe-line of uAL100 was originally developed for plasma samples, the error suppression may not be sufficient to remove all errors in urine DNA. Therefore, although the current version of error suppression in uAL100 is reliable, an error suppression method specifically for urine samples is needed to improve the accuracy of uAL100. Finally, since our target capture panel is not solely composed of genes related to BC, the cost of our method is not fully optimized for BC detection. Furthermore, some known BC-related genes, such as PLEKHS1, are not contained in the target capture panel. Consequently, it might be possible to in-crease the sensitivity for BC detection while reducing the error rate and cost by adjusting the gene panel.

             After:

[Line 339] We found that uAL100 performed well in detecting BC without tumor or blood samples; however, our study has several limitations. First, our cohort was small, and further validation of our method with larger samples is needed. Specifically, more tumor and urine sample pairs are need to be analyzed to obtain statistical significance of mutational concordance study, since we only analyzed 7 sample pairs, which is a small number to provide statistical significance to the results. Second, since the analysis pipeline of uAL100 was originally developed for plasma samples, the error suppression may not be sufficient to re-move all errors in urine DNA. Therefore, although the current version of error suppression in uAL100 is reliable, an error suppression method specifically for urine samples is needed to improve the accuracy of uAL100. Third, since our target capture panel is not solely composed of genes related to BC, the cost of our method is not fully optimized for BC detection. Furthermore, some known BC-related genes, such as PLEKHS1, are not contained in the target capture panel. Consequently, it might be possible to increase the sensitivity for BC detection while reducing the error rate and cost by adjusting the gene panel. Finally, as we mentioned, surveillance of NMIBC patients is important due to their high recurrence rate. Therefore, we need to validate the surveillance ability of uAL100 to make our method more meaningful in real clinical situations.

With mentioned modifications, we believe that we satisfied the Reviewer’s recommendation ‘The discussion must be modified’.

Comment 3-9

- Finally, the references do not have the format required by the journal.

Authors’ response

We appreciate the Reviewer’s valuable suggestion. We changed the style of reference that satisfies criteria of Cancers.

Reviewer #4

Comment 4-1

- How many authors are sure about the accuracy of the screening, especially because urine-based method detection has significant levels of false-positive signals? During disease progression, individual disseminated tumor cells and consecutively metastases can acquire characteristics that do not match those of the corresponding primary tumors, and often are only hardly assessable for further evaluation.

Authors’ response

We appreciate the Reviewer’s valuable suggestion. We would like to clarify that all authors who reviewed the results of bladder cancer screening using urine DNA (including all correspondence and first authors) agreed on the accuracy of our method for BC detection, even though the number of samples we assayed was not large enough. As described in the manuscript, uAL100 was developed based on AL100, which was developed for blood-based cancer screening and AL100 showed significantly low levels of false positive signals in our previous study [Br J Cancer 127, 898–907 (2022)]. When we directly applied AL100 to urine DNA, we observed several significant false positive mutations in healthy control groups, as Reviewer indicated. To remove mutations in healthy control groups, we increased the cutoff level of variant allele frequency from 0.1-0.5% to 1.5% and utilized several oncogene databases, which were the main differences. As a result, we were able to remove false positive mutations in healthy control groups and, through mentioned developing processes, attain certainty in our results.

Regarding the latter statement of this comment, we agreed that obtaining metastatic tumor-derived materials is challenging, especially when metastasis occurred outside of urinary tract. However, we anticipate that metastatic tumor-derived materials can flow in urines and eventually be detected by sampling urine. However, this hypothesis will require further studies to be validated.

Comment 4-2

- Author should describe more about the Alpha Liquid technology if it is only for screening Circulating tumor DNA (ctDNA), how is this platform developed, and other details in the manuscript. This information will be useful for other scientists and clinicians.

Authors’ response

We appreciate the Reviewer’s valuable suggestion. The detailed description about the AL100 was presented in our previous study [Br J Cancer 127, 898–907 (2022)], and we referred this study in the introduction section of our manuscript. Also, another study that utilized blood-based AL100 is preparing for publication, we anticipate that other scientists and clinicians can more easily find and understand principle and properties of AL100.

Comment 4-3, 4-4, 4-6

- Does this technology have a patent, or anyone can use it?

- How much feasible is this technology for the common lab to use?

- Also author should mention the real clinical scenario with reflect on uAL100 technology and whether it has already been approved using blood-serum-based screening for cancer patients such as colorectal or others.

Authors’ response

We appreciate the Reviewer’s valuable suggestion. Here, we would like to reply to comments 4-3, 4-4, and 4-6 simultaneously, since those comments seems closely related. IMBdx, one of the affiliations of authors of this manuscript, is one of the main group (company) that developed the Alpha Liquid technology and has a commercial license for the blood-based AL100 technology. IMBdx is a distributor of the Alpha Liquid technology and provides it in several Asian countries as well as South Korea. Also, IMBdx is preparing to distribute the technology to western countries. As a result, we anticipate that both AL100 and uAL100 can be utilized wordwide, even in common labs.

Regarding the response to comment 4-6, we added a new paragraph in the discussion section, as follows:

[Line 311] For the clinical utilization of liquid biopsy for bladder cancer diagnosis, factors such as cost-effectiveness, test standardization and diagnostic accuracy should be taken into consideration. In this study, we described the process of sample preparation, DNA extraction and genomic analysis for our test and validated the diagnostic accuracy of uAL100 for bladder cancer. During the validation of uAL100, we showed that uAL100 accurately detects utDNA without matched PBMCs, which enables an increase in convenience and a reduction in cost. Regarding cost-effectiveness, although sequencing cost is decreasing, the cost for sequencing may still be more expensive than conventional urine tests for bladder cancer. Therefore, we do not expect uAL100 to fully replace cystoscopy which is the standard procedure for the diagnosis of bladder cancer. However, uAL100 has high diagnostic accuracy, and we expect that uAL100 could identify patients who needs cystoscopy and reduce the number of unnecessary procedures in patients who may have undergone invasive procedures without our test.

In the case of the approval of blood-serum-based cancer screening of AL100, AL100 is already utilized in 23 major cancer centers in South Korea, and we mentioned above, we are preparing approval and commercialization of AL100 in several other countries. However, we determined that these statements are not proper to mention in the manuscript and decided not to mention them.

Comment 4-5

- Indeed, screening BC patients using liquid biopsy samples such as urine is an effective effort. However, the mirror image of tumor cell heterogeneity and real-time reflection in the genomic landscape for non-muscle invasive cancer is difficult. The author’s viewpoint is important in this regard and may include in the conclusion section of the manuscript.

Authors’ response

We appreciate the Reviewer’s valuable suggestion. We agree that obtaining the entire genomic landscape of NMIBC is crucial for making proper clinical decisions and achieving consequent complete remission of the disease. Also, we agree that reflecting information from the whole clones of NMIBC tumors is challenging due to the severely low levels of subclonal-tumor-derived materials in urine. Nevertheless, we believe that still liquid biopsy has advantageous in obtaining integrative tumor information, particularly when considering tumor metastasis and ease of serial sampling. With its high accuracy, we expect that uAL100 has the potential to reflect entire tumor heterogeneity with further developments. Also, as we mentioned earlier, real-time acquisition of genomic landscapes may be obtainable with serial sampling of urine at different time points. We added a new paragraph in the discussion section to address this comment in more detail, as follows:

              [Line 324] As in our cohort, NMIBC is the most common type of bladder cancer at the time of diagnosis. However, unlike similar clinical and histopathological characteristics of NMIBC tumors, their genomic landscapes are variable, resulting in largely different dis-ease progression and response to treatment [33,34]. Consequently, about 50-70% of NMIBC patients experience recurrence and progression to MIBC, necessitating continuous cystoscopy and diverse therapeutic interventions [35,36]. Therefore, accurate capture of the genomic landscape of NMIBC, including its tumor cell heterogeneity, is necessary for the complete cancer remission in early stages of the disease. However, the amount of released tumor-derived materials in bodily-fluids for NMIBC are small, and therefore reflecting integrative information of NMIBC is challenging. To overcome such challenges, we envision that uAL100, with its high accuracy, has the potential to capture genomic landscape of NMIBC containing information from subclonal or metastatic tumors at low levels. Thus, uAL100 is expected to be helpful in early clinical decisions and lifelong surveillance by preventing excessive cystoscopy and therapeutic intervention, which we need to validate in the future studies.

Reviewer #5

Comment 5-1

- The introduction could be a little longer. In the introduction, I would suggest taking information from a recently published article within MDPI (e.g. 10.3390/ijms232113206 or a different analysis) reviewing FDA-approved tests along with many others including ncRNA and cfDNA, and state the actual poor sensitivities and specificities. – also, in this case, lines 37-38 with “… however, and …” is a poor choice of phrasing please rephrase.

Authors’ response

We appreciate the Reviewer’s valuable suggestion. In this comment, the Reviewer recommended revising the introduction section with several opinions: 1) lengthening the introduction section, 2) stating additional urine-based BC detection methods, 3) stating the actual sensitivity and specificity, and 4) rephrasing lines 37-38. We revised the introduction section by reflecting the Reviewer’s opinions with several additional changes as follows:

             Before:

[Page 1, Introduction] Hematuria is a prominent symptom in the early stages of bladder cancer (BC) and is present in 80–90% of patients with BC [1]. However, only 10–22% of patients with hematuria are eventually diagnosed with BC [2, 3]. Cystoscopy, the gold-standard method for detecting BC, is an invasive and costly procedure that carries risks of complication, and its accuracy is highly dependent on the skills of the operator. Therefore, there is a need for non-invasive tests to detect BC with reliable accuracy.

Several non-invasive urine-based tests have been developed to complement invasive cystoscopy for BC diagnosis (e.g. urine cytology, UroVysion, and NMP22) [4-6]. These tests have poor sensitivity, however, and do not provide genotyping information to aid clinical decisions [5, 6]. As an alternative, targeted deep sequencing of cell-free DNA (cfDNA) in blood or urine can detect genomic mutations at low allele frequencies, enabling early diagnosis of BC [7-10]. In patients with BC, DNA in urine has been reported to contain more mutations with higher variant allele frequencies (VAFs) than cfDNA in blood [10]. There-fore, targeted deep sequencing of urine DNA is a promising approach for clinical applications in BC, although accurate detection of urine tumor DNA (utDNA) is challenging due to the presence of false-positive signals at high VAFs [11-14].

In a previous study, we detected circulating tumor DNA in the blood of patients with colorectal cancer using our AlphaLiquid100 target capture panel and its paired analysis pipeline, which we refer to collectively as AL100 [15]. The analysis pipeline suppresses errors in deep sequencing data for 118 cancer-associated gene regions and detects sin-gle-nucleotide variants (SNVs), insertions/deletions (INDELs), copy-number variations (CNVs), and gene fusions with VAFs as low as 0.1%. The AlphaLiquid100 panel includes genes associated with BC, so we decided to utilize AL100 to accurately detect utDNA for the diagnosis of BC.

In this study, we describe uAL100, a urine-based BC detection method that accurately detects utDNA without matched tumor information. To validate the accuracy of uAL100, we applied uAL100 to 43 urine samples and 7 matched tumor samples from patients with BC and 21 urine samples from healthy donors (Figure 1a). We compared the mutations in the paired tumor and urine samples and observed a high level of mutational concordance within matched pairs. Specifically, 82.6% of the mutations identified in the urine samples were also identified in the matched tumor samples. The sensitivity and specificity of uAL100 for BC detection among the 43 patients and 21 controls were 83.7% and 100%, respectively. We confirmed that the mutations identified by uAL100 are significant in BC tumorigenesis and progression and have the potential to serve as biomarkers for BC.

             After:

[Page 1, Introduction] Hematuria is a prominent symptom in the early stages of bladder cancer (BC) and is present in 80–90% of patients with BC [1]. However, only 10–22% of patients with hematuria are eventually diagnosed with BC [2,3]. Cystoscopy, the gold-standard method for detecting BC, is an invasive and costly procedure that carries risks of complication, and its accuracy is highly dependent on the skills of the operator. Therefore, there is a need for non-invasive tests to detect BC with reliable accuracy.

Several non-invasive urine-based tests have been developed to complement invasive cystoscopy for BC diagnosis utilizing materials in urine (e.g. urine cytology, BTA, UroVysion, uCyt+, and NMP22) [4-7]. However, these tests have poor accuracy in detecting BC, with sensitivities between 56% and 83% and specificities between 64% and 88%. Additionally, these urine-based tests do not provide genotyping information to aid clinical decisions [5,6]. Therefore, there is a need for the development of alternative non-invasive methods with enhanced accuracy and genetic information of BC.

As alternative non-invasive methods for detecting BC, targeted deep sequencing of cell-free DNA (cfDNA) in blood or DNA in urine (supernatant or cell pellet) have been developed, as these methods can detect genomic mutations at low allele frequencies, enabling diagnosis and genotyping of BC in early stages of the disease [8-12]. Especially, deep sequencing of urine is more advantageous than blood in BC detection since urine contains relatively more enriched tumor-derived materials than blood due to a lower amount of leukocyte-derived materials in patients with BC [12,13]. Additionally, since urine can be sampled entirely non-invasively with practically unlimited sample volume, urine sampling is more advantageous than blood sampling. Therefore, targeted deep sequencing of DNA in supernatant or cell pellet of urine is a promising approach for clinical applications in BC, although accurate detection of BC is challenging due to the presence of significant levels of false-positive signals (sensitivity: 83-87% and specificity: 85~96%) [10,11,14,15].

In a previous study, we detected circulating tumor DNA in the blood of patients with colorectal cancer using our AlphaLiquid100 target capture panel and its paired analysis pipeline, which we refer to collectively as AL100 [16]. The analysis pipeline suppresses errors in deep sequencing data for 118 cancer-associated gene regions and detects single-nucleotide variants (SNVs), insertions/deletions (INDELs), copy-number variations (CNVs), and gene fusions with variant allele frequencies (VAFs) as low as 0.1%. The Al-phaLiquid100 panel includes genes associated with BC, so we decided to utilize AL100 to accurately detect urine tumor DNA (utDNA) in urine supernatant for the diagnosis of BC.

In this study, we describe uAL100, a urine-based BC detection method that accurately detects utDNA without matched tumor information. When applied to urine samples from our cohort, uAL100 successfully suppressed technical errors, resulting in a specificity of 100% for BC detection with a sensitivity of 83.7%. We expect that uAL100 will more reliably detect early BC than other urine based methods, thereby reducing unnecessary tests for patients with hematuria. Also, uAL100 is expected to be helpful in making clinical decisions for patients with BC since uAL100 provides genotypes of BC including mutations that are significant in BC tumorigenesis and progression. Collectively, we anticipate that uAL100 has the potential to greatly improve the efficiency and efficacy of BC detection and treatment.

We cited a journal that the Reviewer recommended as #Reference 7, and added two more methods, BTA and uCyto+. Also, we stated actual sensitivity and specificity levels of previous methods, and added the advantages of deep sequencing of urine than blood. Finally, we revised last paragraph by replacing results with implication and significance of uAL100.

Comment 5-2

- How cost-effective would your method of diagnosis be compared to today’s golden standard?

Authors’ response

If we consider the golden standard in terms of liquid biopsy methods, we would suggest the blood-based method of Guardant Health that is the most commercially advanced company in the field of liquid biopsy. When compared to Guardant Health’s method, the cost of our method is 1/5 (>$1,000). Therefore, we would like to claim that our method has cost competitiveness. However, we considered that cost comparison between blood-based method is not proper to be discussed in our study, we omitted such a comparison.

In the meanwhile, urine-based commercial non-invasive BC detection methods, such as the ImmunoCyt test of Scimedx, the nuclear matrix protein 22 (NMP22) immunoassay test of Matritech, and multi-target FISH (UroVysion), may be options for BC detection with lower costs than uAL100. However, as we mentioned in our manuscript, since the accuracy of these methods is poor and genotyping of bladder cancer is not available, we consider price comparison with these methods inappropriate.

Comment 5-3

- What was the VAF distribution in the study? Why did you choose this germline filter? Subjects under 40, 40-60, …, above 95 and were there any subjects outside of ranges?

Authors’ response

We appreciate the Reviewer’s valuable suggestion. For the removal of putative germline mutations, we removed mutations with VAFs between 40%-60% or >95%. Although theoretical VAFs of germline mutations are 50% or 100%, we broadened the ranges to sufficiently remove germline mutations since germline mutations showed VAFs between 40%-60% or >95% in real situations, when we analyzed PBMCs in our other studies which we have not published. As we mentioned in the result section “3.5. Information from Peripheral Blood Mononuclear Cells in Urine-Based Bladder Cancer Detection” and the discussion section, VAFs of several mutations were included in the 40%-60% or >95% range and excluded from the final result. Nevertheless, due to this germline filter, the sensitivity was not decreased while the specificity increased. Therefore, we concluded that this germline filter is effective in preventing healthy donors from being classified as bladder cancer patients while preserving sensitivity. However, more urine samples and paired PBMCs need to be analyzed to more profoundly test this germline filter, as discussed in the result section 3.5. In our cohort, we could not observe any subjects outside of the ranges, which was inferred from the absence of any detected mutations in healthy donors.

Comment 5-4

- Line 122-123 T-tests are used to compare or determine the differences between groups.

Authors’ response

We appreciate the Reviewer’s valuable suggestion. We corrected explanation of T-test as the reviewer commented, and changed indicated line, as follows:

             Before:

              [Line 132] Comparisons between two groups were performed using t-tests. ANOVA was used for comparisons of more than two groups.

             After:

[Line 132] T-test was used to test for differences between two groups, and ANOVA was used for more than two groups.

Comment 5-5

- Line 138-139 are there any differences in results for NMIBC and MIBC

Authors’ response

We appreciate the Reviewer’s valuable suggestion. There were differences in the frequency of mutated genes between NMIBC and MIBC, as shown in Figure 4b, although the number of samples from MIBC patients was relatively small compared to samples from NMIBC patients. We presented this result in the result section “3.4. Bladder Cancer Genotypes Determined Using uAL100” in our original manuscript. Also, the mutation counts and percentage of utDNA were higher in urines from MIBC patients than those from NMIBC patients, as shown in Figure 3b and d. However, in terms of sensitivity levels, NMIBC and MIBC were not significantly different (83.3% vs 85.7%). Since NMIBC and MIBC can be distinguished from cancer stages, we considered that presenting sensitivity levels according to tumor stage would sufficiently explain the sensitivity levels of NMIBC and MIBC. However, additional presentation of sensitivity levels in terms of histology would provide a broader perspective. Therefore, we added a relevant sentence in the result section “3.3. Bladder Cancer Detection without Tumor Samples”, as follows:

             Before:

[Line 199] In a subgroup analysis according to tumor stage, the sensitivity of detection in subgroups with stage Ta+Tis, T1, and T2–T4 disease was 71.4%, 90.9%, and 85.7%, respectively, with a specificity of 100% in each subgroup (Figure 3g).

After:

[Line 199] In a subgroup analysis according to tumor stage, the sensitivity of detection in subgroups with stage Ta+Tis, T1, and T2–T4 disease was 71.4%, 90.9%, and 85.7%, respectively, with a specificity of 100% in each subgroup (Figure 3g). When subgroups with stages that cor-respond to NMIBC were combined, the sensitivity of detection was 83.3%, which is similar to those of MIBC (T2-T4).

Comment 5-6

- Lines 156-157 specify the only one found mutation as this part gets distracting when reading.

Authors’ response

We appreciate the Reviewer’s valuable suggestion. We admitted that the indicated line could lead to confusion for readers, and we specified indicated mutation, as follows:

             Before:

[Line 172] Only one mutation was detected in the other patient, and that mutation was detected only in the urine DNA.

             After:

[Line 172] Only one mutation, TP53 M237I, was detected in the other patient, and that mutation was detected only in the urine DNA.

Comment 5-7

- Line 159-161 please rephrase for clarity.

Authors’ response

We appreciate the Reviewer’s valuable suggestion. We admitted that the indicated line can occur confusions for readers, and we changed those line clearly, as follows:

             Before:

              [Line 175] Among the mutations detected in the urine samples, those that were shared with matched tumor samples had higher VAFs than those that were not shared (median VAF = 17.18% vs. 2.06%; p = 0.025; Figure 2c).

After:

[Line 175] Within urine samples, the mutations that were shared with matched tumor samples had a higher median VAF than those that were not shared with matched tumor samples (median VAF = 17.18% vs. 2.06%; p = 0.025; Figure 2c).

Comment 5-8

- Line 163 comparison with 7 samples, ‘6 of 7’ positive is still a low statistical amount to state that these are truly positive. This is a considerable limitation that should be stated in the limitations.

Authors’ response

We appreciate the Reviewer’s valuable suggestion. We agreed with the Reviewer’s opinion and recognized that the low number of available paired samples is one of the limitations of our study when we wrote this manuscript. However, while we were focusing on the performance and limitations of uAL100 in terms of BC detection, we missed stating the limitation of our mutational concordance study. Therefore, we added a statement about the pointed limitation in the result section “3.2. Mutational Concordance in Paired Samples of Tumor and Urine DNA” after the indicated line, as follows:

             Before:

[Line 180] We concluded that most of the mutations detected in the urine samples by uAL100 were true positives, suggesting that uAL100 accurately detects utDNA in urine.

             After:

              [Line 180] We concluded that most of the mutations detected in the urine samples by uAL100 were true positives, suggesting that uAL100 accurately detects utDNA in urine. Nevertheless, to obtain statistical significance of high mutational concordance between paired samples additional sample pairs should be assayed, which is one of our future tasks.

and in discussion section as follows:

             Before:

              [Line 340] First, our cohort was small, and further validation of our method with larger samples is needed.

             After:

[Line 340] First, our cohort was small, and further validation of our method with larger samples is needed. Specifically, more tumor and urine sample pairs are need to be analyzed to obtain statistical significance of mutational concordance study, since we only analyzed 7 sample pairs, which is a small number to provide statistical significance to the results.

Comment 5-9

- Line 271 - Previous urine-based methods – citations.

Authors’ response

We appreciate the Reviewer’s valuable suggestion. We added citations that correspond to previous urine-based methods in indicated lines (reference #10, 11) as follows:

             Before:

[Line 300] Previous urine-based methods for BC detection showed significant levels of false-positive signals with similar or lower sensitivity compared with our method.

After:

[Line 300] Previous urine-based methods using deep sequencing for BC detection showed significant levels of false-positive signals with similar or lower sensitivity compared with our method (Table 2) [10,11].

Additionally, we added a table that compared sensitivity and specificity of the methods (Table 2) after the discussion section as follows:

Comment 5-10

- Please revise limitations.

Authors’ response

We appreciate the Reviewer’s valuable suggestion. We revised the limitations of our method in the discussion section as follows:

             Before:

              [Line 339] We found that uAL100 performed well in detecting BC without tumor or blood samples; however, our study has several limitations. First, our cohort was small, and further validation of our method with larger samples is needed. Second, since the analysis pipe-line of uAL100 was originally developed for plasma samples, the error suppression may not be sufficient to remove all errors in urine DNA. Therefore, although the current version of error suppression in uAL100 is reliable, an error suppression method specifically for urine samples is needed to improve the accuracy of uAL100. Finally, since our target capture panel is not solely composed of genes related to BC, the cost of our method is not fully optimized for BC detection. Furthermore, some known BC-related genes, such as PLEKHS1, are not contained in the target capture panel. Consequently, it might be possible to in-crease the sensitivity for BC detection while reducing the error rate and cost by adjusting the gene panel.

             After:

[Line 339] We found that uAL100 performed well in detecting BC without tumor or blood samples; however, our study has several limitations. First, our cohort was small, and further validation of our method with larger samples is needed. Specifically, more tumor and urine sample pairs are need to be analyzed to obtain statistical significance of mutational concordance study, since we only analyzed 7 sample pairs, which is a small number to provide statistical significance to the results. Second, since the analysis pipeline of uAL100 was originally developed for plasma samples, the error suppression may not be sufficient to remove all errors in urine DNA. Therefore, although the current version of error suppression in uAL100 is reliable, an error suppression method specifically for urine samples is needed to improve the accuracy of uAL100. Third, since our target capture panel is not solely composed of genes related to BC, the cost of our method is not fully optimized for BC detection. Furthermore, some known BC-related genes, such as PLEKHS1, are not contained in the target capture panel. Consequently, it might be possible to in-crease the sensitivity for BC detection while reducing the error rate and cost by adjusting the gene panel. Finally, as we mentioned, surveillance of NMIBC patients is important due to their high recurrence rate. Therefore, we need to validate the surveillance ability of uAL100 to make our method more meaningful in real clinical situations.

Reviewer 4 Report

The author Lee et al., in the article “Accurate detection of urothelial bladder cancer using target 2 ed deep sequencing of urine DNA” author developed uAL100 to diagnose non-muscle invasive bladder cancer by identifying tumor DNA mutation in urine samples using deep sequencing. The author claims that using the mentioned screening method reduces cystoscopy tests for patients with hematuria.

1.    How many authors are sure about the accuracy of the screening, especially because urine-based method detection has significant levels of false-positive signals? During disease progression, individual disseminated tumor cells and consecutively metastases can acquire characteristics that do not match those of the corresponding primary tumors, and often are only hardly assessable for further evaluation.

2.    Author should describe more about the Alpha Liquid technology if it is only for screening Circulating tumor DNA (ctDNA), how is this platform developed, and other details in the manuscript. This information will be useful for other scientists and clinicians.

3.    Does this technology have a patent, or anyone can use it?

4.     How much feasible is this technology for the common lab to use?

5.    Indeed, screening BC patients using liquid biopsy samples such as urine is an effective effort. However, the mirror image of tumor cell heterogeneity and real-time reflection in the genomic landscape for non-muscle invasive cancer is difficult. The author’s viewpoint is important in this regard and may include in the conclusion section of the manuscript.

6.    Also author should mention the real clinical scenario with reflect on uAL100 technology and whether it has already been approved using blood-serum-based screening for cancer patients such as colorectal or others.

Author Response

(The authors gave the same response as above.)

Reviewer 5 Report

I would like to congratulate the authors on a quality piece of research.

The introduction could be a little longer

-          In the introduction, I would suggest taking information from a recently published article within MDPI (e.g. 10.3390/ijms232113206 or a different analysis) reviewing FDA-approved tests along with many others including ncRNA and cfDNA, and state the actual poor sensitivities and specificities. – also, in this case, lines 37-38 with “… however, and …” is a poor choice of phrasing please rephrase.

How cost-effective would your method of diagnosis be compared to today’s golden standard?

What was the VAF distribution in the study? Why did you choose this germline filter? Subjects under 40, 40-60, …, above 95 and were there any subjects outside of ranges?

Line 122-123 T-tests are used to compare or determine the differences between groups.

Line 138-139 are there any differences in results for NMIBC and MIBC

Lines 156-157 specify the only one found mutation as this part gets distracting when reading

Line 159-161 please rephrase for clarity

Line 163 comparison with 7 samples, 6 of 7 positive is still a low statistical amount to state that these are truly positive. This is a considerable limitation that should be stated in the limitations

Line 271 - Previous urine-based methods – citations

Please revise limitations

Author Response

May 13, 2023

Kiran Yu

Assistant Editor, Cancers

Dear Dr. Yu,

We are grateful to you and the Reviewers for your valuable comments and suggestions pertaining to our manuscript. We have taken all these comments into account and are submitting a revised version of our paper.

A full point-by-point response to the Reviewer’s comments is included below. We believe that incorporating the helpful suggestions from the Reviewers has substantially improved our manuscript, and it is now suitable for publication in Cancers.

Thank you for considering our revised manuscript. We look forward to hearing from you.

Kwang Hyun Kim, MD, Ph.D.

Department of Urology,

Ewha Womans University Seoul Hospital, Seoul 07804, Republic of Korea

Tel: +82-2-6986-1685

and

Duhee Bang, Ph.D.

Professor of Chemistry,

Yonsei University, Seoul 03722, Republic of Korea

Tel: +82-10-3357-0611

Reviewer Comments: Point-by-point response:

We thank the Reviewers for their constructive comments on our manuscript and recommendation of publication with modifications.

Listed below are our responses to the Reviewers’ suggestions for improvement, and the suggestions are grouped according to the Reviewer:

Reviewer #1

Comment 1-1

- In line 44, the author state that high VAFs are linked to false-positive. This seems counterintuitive. Can they briefly explain why this is the case?

Authors’ response

We appreciate the Reviewer’s valuable suggestion. We admitted that the pointed statement is counterintuitive since the indicated sentence is vague. Actually, we wrote the indicated sentence to imply the following claim: The variant allele frequencies of called mutations are high in urine samples from healthy donors as well as in urine samples from bladder cancer patients, which leads to misclassification of healthy donors to bladder cancer patients. Therefore, what we really wanted to claim was that high VAFs of mutations can be observed in urine DNA from healthy donors after stringent mutation calling, and we have observed such cases in previous studies (reference #11, 14). However, since this line can cause confusion for readers, as the Reviewer experienced, we decided to rephrase the indicated line by not mentioning “high VAF”, as follows:

Before:

[Line 51] Therefore, targeted deep sequencing of urine DNA is a promising approach for clinical applications in BC, although accurate detection of urine tumor DNA (utDNA) is challenging due to the presence of false-positive signals at high VAFs [11-14].

             After:

[Line 51] Therefore, targeted deep sequencing of DNA in supernatant or cell pellet of urine is a promising approach for clinical applications in BC, although accurate detection of BC is challenging due to the presence of significant levels of false-positive signals (sensitivity: 83-87% and specificity: 85~96%) [10,11,14,15].

Comment 1-2

- Lines 54-63 and Figure 1a, are more suited for methodology and result section. I suggest the authors leave only the first and final sentence here and move the other content elsewhere.

Authors’ response

We appreciate the Reviewer’s valuable suggestion. We have revised the indicated paragraph by replacing the sentences containing results with sentences that highlight the implications and significance of our method in clinical situations, as follows:

Before:

[Line 64] In this study, we describe uAL100, a urine-based BC detection method that accurately detects utDNA without matched tumor information. To validate the accuracy of uAL100, we applied uAL100 to 43 urine samples and 7 matched tumor samples from patients with BC and 21 urine samples from healthy donors (Figure 1a). We compared the mutations in the paired tumor and urine samples and observed a high level of mutational concordance within matched pairs. Specifically, 82.6% of the mutations identified in the urine samples were also identified in the matched tumor samples. The sensitivity and specificity of uAL100 for BC detection among the 43 patients and 21 controls were 83.7% and 100%, respectively. We confirmed that the mutations identified by uAL100 are significant in BC tumorigenesis and progression and have the potential to serve as biomarkers for BC.

After:

[Line 64] In this study, we describe uAL100, a urine-based BC detection method that accurately detects utDNA without matched tumor information. When applied to urine samples from our cohort, uAL100 successfully suppressed technical errors, resulting in a specificity of 100% for BC detection with a sensitivity of 83.7%. We expect that uAL100 will more reliably detect early BC than other urine based methods, thereby significantly reducing unnecessary tests for patients with hematuria. Also, uAL100 is expected to be helpful in making clinical decisions for patients with BC, since uAL100 provides genotypes of BC including mutations that are significant in BC tumorigenesis and progression. Collectively, we an-ticipate that uAL100 has the potential to greatly improve the efficiency and efficacy of BC treatment.

Also, we moved Figure 1a to the end of the result section “3.1. Patient Characteristics and Study Design” where it was first mentioned. Additionally, the sentences containing results were removed as they overlap with the contents in the result sections.

Comment 1-3

- Lines 103-104. I find the use of the term 'germline mutation' misleading. In my opinion the term 'polymorphism' would be clearer.

Authors’ response

We appreciate the Reviewer’s valuable suggestion. We admitted that the term ‘single nucleotide polymorphism (SNP) databases’ seems clearer than the term ‘germline mutation databases’ and changed indicated lines as follows:

Before:

[Line 112] Germline databases were used at the primary variant calling step to filter out putative germline mutations.

             After:

[Line 112] Single-nucleotide polymorphism databases were used at the primary variant calling step to filter out putative germline mutations.

Since we intended to remove only germline mutations (not somatic mutations), we decided not to change the term ‘germline mutation’ to ‘SNP’ in the other parts of the manuscripts.

Comment 1-4

- Line 120. I am not sure this is the best way to calculate mutational concordance as it provides two different results, one for the tumour samples and one for the urine sample. As concordance is a measure of how much the results of tumour and urine analysis agree with each other, I would have expected to be a single value.

Authors’ response

We appreciate the Reviewer’s valuable suggestion. We considered that both two values (the ratio of urine mutations detected in tumors and vice versa) necessary for the mutational concordance study as each value indicates positive predictive value and sensitivity, respectively. Therefore, we still want to present both values. However, revised the indicated line by changing the term ‘the mutational concordance level’ as follows:

Before:

[Line 127] The annotated SNVs and INDELs of all seven patients were then merged, and the mutational concordance level (detection ratio) was calculated using the following formula:

             After:

[Line 127] The annotated SNVs and INDELs of all seven patients were then merged, and the ratio of mutations of one given sample type that detected in paired sample type (detection ratio) was calculated using the following formula:

Comment 1-5

- Lines 125-127. The authors states that the mutation with the highest VAF in each patient was used to calculate sensitivity and specificity. Can the authors provide more details about how this was calculated?

Authors’ response

We appreciate the Reviewer’s valuable suggestion. We used ‘the highest VAF’ in the indicated lines when we investigated cutoff levels of VAF that give best sensitivity and specificity after applying all filters, which is not proper to be used in the indicated lines. Therefore, we revised the explanation of sensitivity and specificity without mentioning of “the highest VAF” as follows:

             Before:

              [Line 135] The mutation with the highest VAF in each patient was selected for calculation of the sensitivity and specificity of BC detection, and the respective VAFs were used as the percentage of utDNA in the entire sample of urine DNA. The sensitivity was the proportion of BC cases detected by uAL100 among 43 cases tested.

             After:

              [Line 135] The mutation with the highest VAF in each patient was selected for calculation of the sensitivity and specificity of BC detection, and the respective VAFs were used as the percentage of utDNA in the entire sample of urine DNA. The sensitivity was the proportion of BC cases having BC driver mutations in their urine DNA among 43 cases tested. The specificity was the proportion of the 21 controls that were correctly identified as not having BC driver mutations in their urine DNA.

Comment 1-6

- lines 151-154. I don't follow how these results were obtained from the formula in methods (line 120). Can the authors show the calculations? Also they state 'concordance was 90.5% for the tumor samples and 82.6% for the urine samples (Figure 2b) but in figure 2b the value reported are different from those stated.

Authors’ response

We appreciate the Reviewer’s valuable suggestion. This suggestion provided us with a valuable opportunity to correct our serious mistake. When we wrote the manuscript, we mistakenly typed the number of mutations for each group (“both tumor and urine”, “tumor only”, urine only”) in lines 151-154. In detail, the number of mutations for shared and urine only are 37 and 9, respectively. However, we mistakenly typed them as 38 and 8, respectively. Fortunately, there were no errors in the relevant figures (Figure 2a-c), so we corrected the indicated sentences as follows:

             Before:

[Line 167] Of the 50 mutations, 38 were shared between paired tumor and urine samples, whereas 12 were detected only in tumor samples or urine samples (tumor only = 4, urine only = 8; Figure 2a; Figure S2). As a result, the mutational concordance was 90.5% for the tumor samples and 82.6% for the urine samples (Figure 2b)

             After:

[Line 167] Of the 50 mutations, 37 were shared between paired tumor and urine samples, whereas 13 were detected only in tumor samples or urine samples (tumor only = 4, urine only = 9; Figure 2a; Figure S2). As a result, 90.2% of tumor mutations were also detected in paired urine samples, and 80.4% of urine mutations were also detected in paired tumor samples (Figure 2b).

Also, though now the Reviewer can deduce the same values using the formula in methods (line129) with the corrected numbers, we provide the specific calculations for deducing the detection ratio values as follows:

We sincerely appreciate this comment, and at the same time, are deeply reflecting on our critical mistake. We believe this valuable experience will be helpful in self-revision of our future publications.

Comment 1-7

- line 211. The mutation rate of some of these genes are much lower than expected based on mutation rates reported in tumour tissue (e.g. FGFR3). Can the authors comment on this in the discussion?

Authors’ response

We appreciate the Reviewer’s valuable suggestion. To answer this question, we compared the mutation rates in urine samples of our study and the mutation rates in bladder cancer tumor from a previous TCGA study [Figure 1 in Nature 507, 315–322 (2014)] with genes included in both the target genes of uAL100 and significantly mutated genes of the TCGA study. Contrary to the Reviewer’s concerns, we observed a high correlation of mutation rates between urine samples (uAL100) and bladder cancer tumor samples. Consequently, we added a relevant figure (Figure 4b) and a sentence in the result section “3.4. Bladder Cancer Genotypes Determined Using uAL100” as follows:

Before:

[Line 232] Other frequently mutated genes were TP53, ARID1A, PIK3CA, ERBB2, ERBB3, FGFR3, KDM6A, and TSC1 (> 10% of cases) that were observed as BC driver genes in previous studies (9, 24).

             After:

[Line 232] Other frequently mutated genes were TP53, ARID1A, PIK3CA, ERBB2, ERBB3, FGFR3, KDM6A, and TSC1 (> 10% of cases) that were observed as BC driver genes in previous studies [10,25]. Also, when compared to the TCGA study that investigated BC tumors, the frequency of cases with mutations in each gene detected by uAL100 showed a high correlation with those detected in the BC tumors reported in the TCGA study (R = 0.91; Figure 4b) [27], which indicates BC genotyping using uAL100 significantly reflects tumor information.

             Figure 4b

(b) Comparisons of mutated gene frequency between tumor samples (TCGA) and urine samples (uAL100). The analyzed genes consisted of those that overlapped between the target genes of uAL100 and the significantly mutated genes identified in TCGA. The correlation coefficient (R) corresponds to the Pearson correlation coefficient.

In the meanwhile, we are cautious about these results as they differ from what the Reviewer mentioned. Therefore, to provide a more profound answer to this question, we would like to know which previous studies the reviewer referred to.

Comment 1-8

- line 272. The authors state that their test perform better compared with others. Can they report and reference specificity and sensitivity for the other tests to support their claim?

Authors’ response

We appreciate the Reviewer’s valuable suggestion. We added a table that compares the sensitivity and specificity of urine-based methods that utilize deep sequencing (Table 2) after the discussion section, as follows:

Also, we added citations for the mentioned urine-based methods to the indicated lines (reference #9, 12), as follows:

             Before:

[Line 300] Previous urine-based methods for BC detection showed significant levels of false-positive signals with similar or lower sensitivity compared with our method.

After:

[Line 300] Previous urine-based methods using deep sequencing for BC detection showed significant levels of false-positive signals with similar or lower sensitivity compared with our method (Table 2) [10,11].

Comment 1-9

- line 275. Can the author add a comment regarding the feasibility of this type of testing (costs, time etc) for routine analysis within the health system (e.g. to replace cystoscopy)?

Authors’ response

We appreciate the Reviewer’s valuable suggestion. To respond to this comment, we merged the first and second paragraphs in the discussion section in to one paragraph and added a new paragraph that discusses about this comment, as follows:

             Before:

[Line 294] In this study, we described uAL100 and validated its performance in BC diagnosis. There was high concordance between mutations detected by uAL100 in matched urine and tumor samples, although the number of matched pairs was only seven in our cohorts. This indicated that the mutations detected in urine by uAL100 were derived from BC tumors. We then showed that uAL100 had 83.7% sensitivity and 100% specificity in detecting BC without tumor samples, suggesting that our method has high accuracy regardless of the tumor stage. Previous urine-based methods for BC detection showed significant levels of false-positive signals with similar or lower sensitivity compared with our method. A genotyping analysis showed that the mutations identified by uAL100 were highly related to BC tumorigenesis.

To increase convenience and reduce cost, we designed uAL100 to detect tumor DNA in urine without additional analysis of PBMCs from patients. We showed that accurate utDNA detection was possible without PBMCs; however, several mutations were removed by the germline filter in 15 of the 36 patients that were diagnosed with BC, so it is possible that the germline filter can lead to imperfect clinical decision making based on the mutation profiles that it produces. Therefore, although the germline filter had little effect on the sensitivity of BC detection without tumor samples, information from PBMCs may still be needed to support optimal clinical decisions.

             After:

[Line 294] In this study, we described uAL100 and validated its performance in BC diagnosis. There was high concordance between mutations detected by uAL100 in matched urine and tumor samples, although the number of matched pairs was only seven in our cohorts. This indicated that the mutations detected in urine by uAL100 were derived from BC tumors. We then showed that uAL100 had 83.7% sensitivity and 100% specificity in detecting BC without tumor samples, suggesting that our method has high accuracy regardless of the tumor stage. Previous urine-based methods using deep sequencing for BC detection showed significant levels of false-positive signals with similar or lower sensitivity com-pared with our method (Table 2) [10,11]. A genotyping analysis showed that the mutations identified by uAL100 were highly related to BC tumorigenesis. Additionally, we showed that accurate utDNA detection was possible without PBMCs which enables entire non-invasiveness of uAL100. However, since several mutations were removed by the germline filter in some patients that were diagnosed with BC, so it is possible that the germline filter can lead to imperfect clinical decision making based on the mutation pro-files that it produces. Therefore, although the germline filter had little effect on the sensitivity of BC detection without tumor samples, information from PBMCs may still be needed to support optimal clinical decisions.

For the clinical utilization of liquid biopsy for bladder cancer diagnosis, factors such as cost-effectiveness, test standardization and diagnostic accuracy should be taken into consideration. In this study, we described the process of sample preparation, DNA extraction and genomic analysis for our test and validated the diagnostic accuracy of uAL100 for bladder cancer. During the validation of uAL100, we showed that uAL100 accurately detects utDNA without matched PBMCs, which enables an increase in convenience and a reduction in cost. Regarding cost-effectiveness, although sequencing cost is decreasing, the cost for sequencing may still be more expensive than conventional urine tests for bladder cancer. Therefore, we do not expect uAL100 to fully replace cystoscopy which is the standard procedure for the diagnosis of bladder cancer. However, uAL100 has high diagnostic accuracy, and we expect that uAL100 could identify patients who needs cystoscopy and reduce the number of unnecessary procedures in patients who may have undergone invasive procedures without our test.

Reviewer #2

Comment 2-1

-Could you please indicate the coverage of the sequencing for each sample?

Authors’ response

We appreciate the Reviewer’s valuable suggestion. We presented the median sequencing coverage of each sample as Table S2 and added sentences describing the median sequencing coverage values in the method section “2.2. Analysis of DNA from Urine and Bladder Tumors”, as follows:

             Before:

[Line 96] The target-enriched DNA libraries were sequenced using the Illumina Novaseq 6000 plat-form (Illumina, USA) to create 150 bp paired-end reads.

             After:

[Line 96] The target-enriched DNA libraries were sequenced using the Illumina Novaseq 6000 plat-form (Illumina, USA) to create 150 bp paired-end reads, with median sequencing depths of 6094× and 7732× for targeted regions in urine and tumor samples, respectively (Table S2).

Comment 2-2

- I suggest not including at the end of the introduction the main results of your results but instead a broad significance of the importance of this study

Authors’ response

We appreciate the Reviewer’s valuable suggestion. We have revised the indicated paragraph by replacing the sentences containing results with sentences that highlight the implications and significance of our method in clinical situations, as follows:

Before:

[Line 64] In this study, we describe uAL100, a urine-based BC detection method that accurately detects utDNA without matched tumor information. To validate the accuracy of uAL100, we applied uAL100 to 43 urine samples and 7 matched tumor samples from patients with BC and 21 urine samples from healthy donors (Figure 1a). We compared the mutations in the paired tumor and urine samples and observed a high level of mutational concordance within matched pairs. Specifically, 82.6% of the mutations identified in the urine samples were also identified in the matched tumor samples. The sensitivity and specificity of uAL100 for BC detection among the 43 patients and 21 controls were 83.7% and 100%, respectively. We confirmed that the mutations identified by uAL100 are significant in BC tumorigenesis and progression and have the potential to serve as biomarkers for BC.

After:

[Line 64] In this study, we describe uAL100, a urine-based BC detection method that accurately detects utDNA without matched tumor information. When applied to urine samples from our cohort, uAL100 successfully suppressed technical errors, resulting in a specificity of 100% for BC detection with a sensitivity of 83.7%. We expect that uAL100 will more reliably detect early BC than other urine based methods, thereby significantly reducing unnecessary tests for patients with hematuria. Also, uAL100 is expected to be helpful in making clinical decisions for patients with BC, since uAL100 provides genotypes of BC including mutations that are significant in BC tumorigenesis and progression. Collectively, we an-ticipate that uAL100 has the potential to greatly improve the efficiency and efficacy of BC treatment.

Also, we moved Figure 1a to the end of the result section “3.1. Patient Characteristics and Study Design” where it was first mentioned. Additionally, the sentences containing results were removed as they overlap with the contents in the result sections.

Reviewer #3

Comment 3-1, 3-2

- Line 54-63: This paragraph should be rewritten and focus on the objective of the work. Elements that are not part of the discussion are developed.

- Figure 1 should be removed and placed in the materials and methods and/or results part.

Authors’ response

We appreciate the Reviewer’s valuable suggestion. We have revised the indicated paragraph by replacing the sentences containing results with sentences that highlight the implications and significance of our method in clinical situations, as follows:

Before:

[Line 64] In this study, we describe uAL100, a urine-based BC detection method that accurately detects utDNA without matched tumor information. To validate the accuracy of uAL100, we applied uAL100 to 43 urine samples and 7 matched tumor samples from patients with BC and 21 urine samples from healthy donors (Figure 1a). We compared the mutations in the paired tumor and urine samples and observed a high level of mutational concordance within matched pairs. Specifically, 82.6% of the mutations identified in the urine samples were also identified in the matched tumor samples. The sensitivity and specificity of uAL100 for BC detection among the 43 patients and 21 controls were 83.7% and 100%, respectively. We confirmed that the mutations identified by uAL100 are significant in BC tumorigenesis and progression and have the potential to serve as biomarkers for BC.

After:

[Line 64] In this study, we describe uAL100, a urine-based BC detection method that accurately detects utDNA without matched tumor information. When applied to urine samples from our cohort, uAL100 successfully suppressed technical errors, resulting in a specificity of 100% for BC detection with a sensitivity of 83.7%. We expect that uAL100 will more reliably detect early BC than other urine based methods, thereby significantly reducing unnecessary tests for patients with hematuria. Also, uAL100 is expected to be helpful in making clinical decisions for patients with BC, since uAL100 provides genotypes of BC including mutations that are significant in BC tumorigenesis and progression. Collectively, we an-ticipate that uAL100 has the potential to greatly improve the efficiency and efficacy of BC treatment.

Also, we moved Figure 1a to the end of the result section “3.1. Patient Characteristics and Study Design” where it was first mentioned. Additionally, the sentences containing results were removed as they overlap with the contents in the result sections.

Comment 3-3

- How was the tumor material obtained? Was it obtained from paraffin-embedded material? Or was it obtained from fresh selection? If fresh material was used, was the presence of tumor confirmed? How was the selection carried out?

Authors’ response

We appreciate the Reviewer’s valuable suggestion. As stated in Material and Methods section, tumor sample was obtained at the time of surgery and stored as a fresh frozen sample. However, pathologic review for fresh frozen sample was not performed in this study.

Comment 3-4

- Why were only 7 paired cases studied? What were the clinicopathological characteristics of these seven patients. For example, pathological stage, histopathological grade, etc.

Authors’ response

We appreciate the Reviewer’s valuable suggestion. Sequencing of tumor DNA was performed only in the initial samples of this study, and 7 paired cases were not selected for this study. After sequencing of 7 tumor DNA for concordance analysis of tumor and urine DNA, we could determine the cutoff VAF for urine DNA for detecting bladder cancer with high sensitivity and specificity. As stated in the result section “3.2. Mutational Concordance in Paired Samples of Tumor and Urine DNA”, although tumor samples were analyzed only in 7 patients, most patients had non-muscle invasive bladder cancer. Therefore, we believe our method is reliable regardless of tumor stage. The patient characteristics of 7 patients are summarized in the table below and shown graphically in Figure 2a.

Comment 3-5

- Table 1. It is noteworthy that most of the cases included in the study were high-grade carcinomas, and that the cytological study was diagnostic in less than half of the cases. What classification system was used to assess the cytologies? Was the cytological material analyzed from a new sample or was the same material used for DNA extraction? Please detail.

Authors’ response

We appreciate the Reviewer’s valuable suggestion. Urine samples for cytological examination was obtained at the time of cystoscopy in clinics (different urine samples for DNA extraction) and the pathologist reported the results according to the Paris System [Acta Cytologica (2016) 60 (3): 185–197.]. If the result was reported as ‘suspicious for high-grade urothelial carcinoma’ or ‘high-grade urothelial carcinoma’, it was categorized as positive and in the case of negative for urothelial carcinoma or no adequate diagnosis possible, then it was categorized as negative. We briefly added a sentence stating about cytology in the method section “2.1. Patient Enrollment and Sample Collection” as follows:

             Before:

[Line 80] All patients provided informed consent for tissue banking and genetic testing.

             After:

[Line 80] All patients provided informed consent for tissue banking and genetic testing. Urine cytology was performed at the time of cystoscopy according to the Paris System [17].

Comment 3-6

- During the presentation of the results, reference is made to their discussion, including bibliographical references. This should be modified and clearly show the results of the study.

Authors’ response

We appreciate the Reviewer’s valuable suggestion. We suspect that the Reviewer raised this comment, especially with regards to the result sections “3.4. Bladder Cancer Genotypes Determined Using uAL100” or “3.5. Information from Peripheral Blood Mononuclear Cells in Urine-Based Bladder Cancer Detection”. Here, we would like to state the validity of using several citations in those result sections.

First, in the result section 3.4, we had to verify the significance of mutations identified by uAL100, and we determined that comparison with the results of previous studies were necessary. Since we analyzed small numbers of tumors (n=7), we believe that comparing our results with those of previous studies was the best way to validate our genotyping results.

For the result section 3.5, we wanted to claim that without matched PBMCs, accurate BC detection is possible using uAL100. However, we did not analyze matched PBMCs for direct validation of our hypothesis. Therefore, we had to indirectly validate our hypothesis, and for inferring processes, cited studies are needed. In summary due to the absence of matched PBMCs, we inferred conclusions that were largely dependent on results of previous studies we cited, which is one of the limitations of our study. While we mentioned this limitation in the end of the result section 3.5, we modified the relevant sentence to make it more clearer, as follows:

Before:

[Line 279] Based on these results, we suggest that PBMCs are not necessary for detecting BC using urine samples; however, further study is required with larger numbers of healthy controls.

After:

[Line 279] Based on these results, we suggest that PBMCs are not necessary for detecting BC using urine samples; however, further study is required with larger numbers of urine samples and matched PBMCs for precise validation of our hypothesis.

We hope that our presented validity can be positively reflected in reconsidering of the Reviewer’s opinions in comment 3-6.

Comment 3-7, 3-8

- The relevant findings of the study are not discussed, nor are the advantages and disadvantages taken into account.

- The discussion must be modified and the analysis supported with adequate bibliographical sources.

Authors’ response

We appreciate the Reviewer’s valuable suggestions. We determined to address both comment 3-7 and comment 3-8 simultaneously, as the Reviewer’s comment ‘The discussion must be modified’ contains both of these comments.

In comment 3-7, the Reviewer recommended to discuss about the advantages and disadvantages of our method. We considered that we have already presented advantages (enhanced accuracy and entirely non-invasiveness due to absence of PBMC) and disadvantages (loss of putative driver mutations due to absence of PBMC) in the first and second paragraphs of unrevised version of our manuscript. However, we also added the advantages of uAL100 in the middle of discussion section in terms of capturing genomic landscape of NMIBC for clinical decisions, since accurate genotyping of NMIBC is difficult due to tumor cell heterogeneity:

[Line 324] As in our cohort, NMIBC is the most common type of bladder cancer at the time of diagnosis. However, unlike similar clinical and histopathological characteristics of NMIBC tumors, their genomic landscapes are variable, resulting in largely different dis-ease progression and response to treatment [33,34]. Consequently, about 50-70% of NMIBC patients experience recurrence and progression to MIBC, necessitating continuous cystoscopy and diverse therapeutic interventions [35,36]. Therefore, accurate capture of the genomic landscape of NMIBC, including its tumor cell heterogeneity, is necessary for the complete cancer remission in early stages of the disease. However, the amount of released tumor-derived materials in bodily-fluids for NMIBC are small, and therefore reflecting integrative information of NMIBC is challenging. To overcome such challenges, we envision that uAL100, with its high accuracy, has the potential to capture genomic landscape of NMIBC containing information from subclonal or metastatic tumors at low levels. Thus, uAL100 is expected to be helpful in early clinical decisions and lifelong surveillance by preventing excessive cystoscopy and therapeutic intervention, which we need to validate in the future studies.

Also, as the Reviewer recommended, we added citations of previous methods when we compared our accuracy with them, as follows:

             Before:

[Line 300] Previous urine-based methods for BC detection showed significant levels of false-positive signals with similar or lower sensitivity compared with our method.

             After:

[Line 300] Previous urine-based methods using deep sequencing for BC detection showed significant levels of false-positive signals with similar or lower sensitivity compared with our method (Table 2) [10,11].

Additionally, we added a table that compared sensitivity and specificity of the methods (Table 2) after discussion section, as follows:

Finally, we added more limitations in terms of low numbers of tumor and urine sample pairs and absence of surveillance study of uAL100 to the final paragraph of the discussion section where we presented the limitations of our study, as follows:

             Before:

[Line 339] We found that uAL100 performed well in detecting BC without tumor or blood samples; however, our study has several limitations. First, our cohort was small, and further validation of our method with larger samples is needed. Second, since the analysis pipe-line of uAL100 was originally developed for plasma samples, the error suppression may not be sufficient to remove all errors in urine DNA. Therefore, although the current version of error suppression in uAL100 is reliable, an error suppression method specifically for urine samples is needed to improve the accuracy of uAL100. Finally, since our target capture panel is not solely composed of genes related to BC, the cost of our method is not fully optimized for BC detection. Furthermore, some known BC-related genes, such as PLEKHS1, are not contained in the target capture panel. Consequently, it might be possible to in-crease the sensitivity for BC detection while reducing the error rate and cost by adjusting the gene panel.

             After:

[Line 339] We found that uAL100 performed well in detecting BC without tumor or blood samples; however, our study has several limitations. First, our cohort was small, and further validation of our method with larger samples is needed. Specifically, more tumor and urine sample pairs are need to be analyzed to obtain statistical significance of mutational concordance study, since we only analyzed 7 sample pairs, which is a small number to provide statistical significance to the results. Second, since the analysis pipeline of uAL100 was originally developed for plasma samples, the error suppression may not be sufficient to re-move all errors in urine DNA. Therefore, although the current version of error suppression in uAL100 is reliable, an error suppression method specifically for urine samples is needed to improve the accuracy of uAL100. Third, since our target capture panel is not solely composed of genes related to BC, the cost of our method is not fully optimized for BC detection. Furthermore, some known BC-related genes, such as PLEKHS1, are not contained in the target capture panel. Consequently, it might be possible to increase the sensitivity for BC detection while reducing the error rate and cost by adjusting the gene panel. Finally, as we mentioned, surveillance of NMIBC patients is important due to their high recurrence rate. Therefore, we need to validate the surveillance ability of uAL100 to make our method more meaningful in real clinical situations.

With mentioned modifications, we believe that we satisfied the Reviewer’s recommendation ‘The discussion must be modified’.

Comment 3-9

- Finally, the references do not have the format required by the journal.

Authors’ response

We appreciate the Reviewer’s valuable suggestion. We changed the style of reference that satisfies criteria of Cancers.

Reviewer #4

Comment 4-1

- How many authors are sure about the accuracy of the screening, especially because urine-based method detection has significant levels of false-positive signals? During disease progression, individual disseminated tumor cells and consecutively metastases can acquire characteristics that do not match those of the corresponding primary tumors, and often are only hardly assessable for further evaluation.

Authors’ response

We appreciate the Reviewer’s valuable suggestion. We would like to clarify that all authors who reviewed the results of bladder cancer screening using urine DNA (including all correspondence and first authors) agreed on the accuracy of our method for BC detection, even though the number of samples we assayed was not large enough. As described in the manuscript, uAL100 was developed based on AL100, which was developed for blood-based cancer screening and AL100 showed significantly low levels of false positive signals in our previous study [Br J Cancer 127, 898–907 (2022)]. When we directly applied AL100 to urine DNA, we observed several significant false positive mutations in healthy control groups, as Reviewer indicated. To remove mutations in healthy control groups, we increased the cutoff level of variant allele frequency from 0.1-0.5% to 1.5% and utilized several oncogene databases, which were the main differences. As a result, we were able to remove false positive mutations in healthy control groups and, through mentioned developing processes, attain certainty in our results.

Regarding the latter statement of this comment, we agreed that obtaining metastatic tumor-derived materials is challenging, especially when metastasis occurred outside of urinary tract. However, we anticipate that metastatic tumor-derived materials can flow in urines and eventually be detected by sampling urine. However, this hypothesis will require further studies to be validated.

Comment 4-2

- Author should describe more about the Alpha Liquid technology if it is only for screening Circulating tumor DNA (ctDNA), how is this platform developed, and other details in the manuscript. This information will be useful for other scientists and clinicians.

Authors’ response

We appreciate the Reviewer’s valuable suggestion. The detailed description about the AL100 was presented in our previous study [Br J Cancer 127, 898–907 (2022)], and we referred this study in the introduction section of our manuscript. Also, another study that utilized blood-based AL100 is preparing for publication, we anticipate that other scientists and clinicians can more easily find and understand principle and properties of AL100.

Comment 4-3, 4-4, 4-6

- Does this technology have a patent, or anyone can use it?

- How much feasible is this technology for the common lab to use?

- Also author should mention the real clinical scenario with reflect on uAL100 technology and whether it has already been approved using blood-serum-based screening for cancer patients such as colorectal or others.

Authors’ response

We appreciate the Reviewer’s valuable suggestion. Here, we would like to reply to comments 4-3, 4-4, and 4-6 simultaneously, since those comments seems closely related. IMBdx, one of the affiliations of authors of this manuscript, is one of the main group (company) that developed the Alpha Liquid technology and has a commercial license for the blood-based AL100 technology. IMBdx is a distributor of the Alpha Liquid technology and provides it in several Asian countries as well as South Korea. Also, IMBdx is preparing to distribute the technology to western countries. As a result, we anticipate that both AL100 and uAL100 can be utilized wordwide, even in common labs.

Regarding the response to comment 4-6, we added a new paragraph in the discussion section, as follows:

[Line 311] For the clinical utilization of liquid biopsy for bladder cancer diagnosis, factors such as cost-effectiveness, test standardization and diagnostic accuracy should be taken into consideration. In this study, we described the process of sample preparation, DNA extraction and genomic analysis for our test and validated the diagnostic accuracy of uAL100 for bladder cancer. During the validation of uAL100, we showed that uAL100 accurately detects utDNA without matched PBMCs, which enables an increase in convenience and a reduction in cost. Regarding cost-effectiveness, although sequencing cost is decreasing, the cost for sequencing may still be more expensive than conventional urine tests for bladder cancer. Therefore, we do not expect uAL100 to fully replace cystoscopy which is the standard procedure for the diagnosis of bladder cancer. However, uAL100 has high diagnostic accuracy, and we expect that uAL100 could identify patients who needs cystoscopy and reduce the number of unnecessary procedures in patients who may have undergone invasive procedures without our test.

In the case of the approval of blood-serum-based cancer screening of AL100, AL100 is already utilized in 23 major cancer centers in South Korea, and we mentioned above, we are preparing approval and commercialization of AL100 in several other countries. However, we determined that these statements are not proper to mention in the manuscript and decided not to mention them.

Comment 4-5

- Indeed, screening BC patients using liquid biopsy samples such as urine is an effective effort. However, the mirror image of tumor cell heterogeneity and real-time reflection in the genomic landscape for non-muscle invasive cancer is difficult. The author’s viewpoint is important in this regard and may include in the conclusion section of the manuscript.

Authors’ response

We appreciate the Reviewer’s valuable suggestion. We agree that obtaining the entire genomic landscape of NMIBC is crucial for making proper clinical decisions and achieving consequent complete remission of the disease. Also, we agree that reflecting information from the whole clones of NMIBC tumors is challenging due to the severely low levels of subclonal-tumor-derived materials in urine. Nevertheless, we believe that still liquid biopsy has advantageous in obtaining integrative tumor information, particularly when considering tumor metastasis and ease of serial sampling. With its high accuracy, we expect that uAL100 has the potential to reflect entire tumor heterogeneity with further developments. Also, as we mentioned earlier, real-time acquisition of genomic landscapes may be obtainable with serial sampling of urine at different time points. We added a new paragraph in the discussion section to address this comment in more detail, as follows:

              [Line 324] As in our cohort, NMIBC is the most common type of bladder cancer at the time of diagnosis. However, unlike similar clinical and histopathological characteristics of NMIBC tumors, their genomic landscapes are variable, resulting in largely different dis-ease progression and response to treatment [33,34]. Consequently, about 50-70% of NMIBC patients experience recurrence and progression to MIBC, necessitating continuous cystoscopy and diverse therapeutic interventions [35,36]. Therefore, accurate capture of the genomic landscape of NMIBC, including its tumor cell heterogeneity, is necessary for the complete cancer remission in early stages of the disease. However, the amount of released tumor-derived materials in bodily-fluids for NMIBC are small, and therefore reflecting integrative information of NMIBC is challenging. To overcome such challenges, we envision that uAL100, with its high accuracy, has the potential to capture genomic landscape of NMIBC containing information from subclonal or metastatic tumors at low levels. Thus, uAL100 is expected to be helpful in early clinical decisions and lifelong surveillance by preventing excessive cystoscopy and therapeutic intervention, which we need to validate in the future studies.

Reviewer #5

Comment 5-1

- The introduction could be a little longer. In the introduction, I would suggest taking information from a recently published article within MDPI (e.g. 10.3390/ijms232113206 or a different analysis) reviewing FDA-approved tests along with many others including ncRNA and cfDNA, and state the actual poor sensitivities and specificities. – also, in this case, lines 37-38 with “… however, and …” is a poor choice of phrasing please rephrase.

Authors’ response

We appreciate the Reviewer’s valuable suggestion. In this comment, the Reviewer recommended revising the introduction section with several opinions: 1) lengthening the introduction section, 2) stating additional urine-based BC detection methods, 3) stating the actual sensitivity and specificity, and 4) rephrasing lines 37-38. We revised the introduction section by reflecting the Reviewer’s opinions with several additional changes as follows:

             Before:

[Page 1, Introduction] Hematuria is a prominent symptom in the early stages of bladder cancer (BC) and is present in 80–90% of patients with BC [1]. However, only 10–22% of patients with hematuria are eventually diagnosed with BC [2, 3]. Cystoscopy, the gold-standard method for detecting BC, is an invasive and costly procedure that carries risks of complication, and its accuracy is highly dependent on the skills of the operator. Therefore, there is a need for non-invasive tests to detect BC with reliable accuracy.

Several non-invasive urine-based tests have been developed to complement invasive cystoscopy for BC diagnosis (e.g. urine cytology, UroVysion, and NMP22) [4-6]. These tests have poor sensitivity, however, and do not provide genotyping information to aid clinical decisions [5, 6]. As an alternative, targeted deep sequencing of cell-free DNA (cfDNA) in blood or urine can detect genomic mutations at low allele frequencies, enabling early diagnosis of BC [7-10]. In patients with BC, DNA in urine has been reported to contain more mutations with higher variant allele frequencies (VAFs) than cfDNA in blood [10]. There-fore, targeted deep sequencing of urine DNA is a promising approach for clinical applications in BC, although accurate detection of urine tumor DNA (utDNA) is challenging due to the presence of false-positive signals at high VAFs [11-14].

In a previous study, we detected circulating tumor DNA in the blood of patients with colorectal cancer using our AlphaLiquid100 target capture panel and its paired analysis pipeline, which we refer to collectively as AL100 [15]. The analysis pipeline suppresses errors in deep sequencing data for 118 cancer-associated gene regions and detects sin-gle-nucleotide variants (SNVs), insertions/deletions (INDELs), copy-number variations (CNVs), and gene fusions with VAFs as low as 0.1%. The AlphaLiquid100 panel includes genes associated with BC, so we decided to utilize AL100 to accurately detect utDNA for the diagnosis of BC.

In this study, we describe uAL100, a urine-based BC detection method that accurately detects utDNA without matched tumor information. To validate the accuracy of uAL100, we applied uAL100 to 43 urine samples and 7 matched tumor samples from patients with BC and 21 urine samples from healthy donors (Figure 1a). We compared the mutations in the paired tumor and urine samples and observed a high level of mutational concordance within matched pairs. Specifically, 82.6% of the mutations identified in the urine samples were also identified in the matched tumor samples. The sensitivity and specificity of uAL100 for BC detection among the 43 patients and 21 controls were 83.7% and 100%, respectively. We confirmed that the mutations identified by uAL100 are significant in BC tumorigenesis and progression and have the potential to serve as biomarkers for BC.

             After:

[Page 1, Introduction] Hematuria is a prominent symptom in the early stages of bladder cancer (BC) and is present in 80–90% of patients with BC [1]. However, only 10–22% of patients with hematuria are eventually diagnosed with BC [2,3]. Cystoscopy, the gold-standard method for detecting BC, is an invasive and costly procedure that carries risks of complication, and its accuracy is highly dependent on the skills of the operator. Therefore, there is a need for non-invasive tests to detect BC with reliable accuracy.

Several non-invasive urine-based tests have been developed to complement invasive cystoscopy for BC diagnosis utilizing materials in urine (e.g. urine cytology, BTA, UroVysion, uCyt+, and NMP22) [4-7]. However, these tests have poor accuracy in detecting BC, with sensitivities between 56% and 83% and specificities between 64% and 88%. Additionally, these urine-based tests do not provide genotyping information to aid clinical decisions [5,6]. Therefore, there is a need for the development of alternative non-invasive methods with enhanced accuracy and genetic information of BC.

As alternative non-invasive methods for detecting BC, targeted deep sequencing of cell-free DNA (cfDNA) in blood or DNA in urine (supernatant or cell pellet) have been developed, as these methods can detect genomic mutations at low allele frequencies, enabling diagnosis and genotyping of BC in early stages of the disease [8-12]. Especially, deep sequencing of urine is more advantageous than blood in BC detection since urine contains relatively more enriched tumor-derived materials than blood due to a lower amount of leukocyte-derived materials in patients with BC [12,13]. Additionally, since urine can be sampled entirely non-invasively with practically unlimited sample volume, urine sampling is more advantageous than blood sampling. Therefore, targeted deep sequencing of DNA in supernatant or cell pellet of urine is a promising approach for clinical applications in BC, although accurate detection of BC is challenging due to the presence of significant levels of false-positive signals (sensitivity: 83-87% and specificity: 85~96%) [10,11,14,15].

In a previous study, we detected circulating tumor DNA in the blood of patients with colorectal cancer using our AlphaLiquid100 target capture panel and its paired analysis pipeline, which we refer to collectively as AL100 [16]. The analysis pipeline suppresses errors in deep sequencing data for 118 cancer-associated gene regions and detects single-nucleotide variants (SNVs), insertions/deletions (INDELs), copy-number variations (CNVs), and gene fusions with variant allele frequencies (VAFs) as low as 0.1%. The Al-phaLiquid100 panel includes genes associated with BC, so we decided to utilize AL100 to accurately detect urine tumor DNA (utDNA) in urine supernatant for the diagnosis of BC.

In this study, we describe uAL100, a urine-based BC detection method that accurately detects utDNA without matched tumor information. When applied to urine samples from our cohort, uAL100 successfully suppressed technical errors, resulting in a specificity of 100% for BC detection with a sensitivity of 83.7%. We expect that uAL100 will more reliably detect early BC than other urine based methods, thereby reducing unnecessary tests for patients with hematuria. Also, uAL100 is expected to be helpful in making clinical decisions for patients with BC since uAL100 provides genotypes of BC including mutations that are significant in BC tumorigenesis and progression. Collectively, we anticipate that uAL100 has the potential to greatly improve the efficiency and efficacy of BC detection and treatment.

We cited a journal that the Reviewer recommended as #Reference 7, and added two more methods, BTA and uCyto+. Also, we stated actual sensitivity and specificity levels of previous methods, and added the advantages of deep sequencing of urine than blood. Finally, we revised last paragraph by replacing results with implication and significance of uAL100.

Comment 5-2

- How cost-effective would your method of diagnosis be compared to today’s golden standard?

Authors’ response

If we consider the golden standard in terms of liquid biopsy methods, we would suggest the blood-based method of Guardant Health that is the most commercially advanced company in the field of liquid biopsy. When compared to Guardant Health’s method, the cost of our method is 1/5 (>$1,000). Therefore, we would like to claim that our method has cost competitiveness. However, we considered that cost comparison between blood-based method is not proper to be discussed in our study, we omitted such a comparison.

In the meanwhile, urine-based commercial non-invasive BC detection methods, such as the ImmunoCyt test of Scimedx, the nuclear matrix protein 22 (NMP22) immunoassay test of Matritech, and multi-target FISH (UroVysion), may be options for BC detection with lower costs than uAL100. However, as we mentioned in our manuscript, since the accuracy of these methods is poor and genotyping of bladder cancer is not available, we consider price comparison with these methods inappropriate.

Comment 5-3

- What was the VAF distribution in the study? Why did you choose this germline filter? Subjects under 40, 40-60, …, above 95 and were there any subjects outside of ranges?

Authors’ response

We appreciate the Reviewer’s valuable suggestion. For the removal of putative germline mutations, we removed mutations with VAFs between 40%-60% or >95%. Although theoretical VAFs of germline mutations are 50% or 100%, we broadened the ranges to sufficiently remove germline mutations since germline mutations showed VAFs between 40%-60% or >95% in real situations, when we analyzed PBMCs in our other studies which we have not published. As we mentioned in the result section “3.5. Information from Peripheral Blood Mononuclear Cells in Urine-Based Bladder Cancer Detection” and the discussion section, VAFs of several mutations were included in the 40%-60% or >95% range and excluded from the final result. Nevertheless, due to this germline filter, the sensitivity was not decreased while the specificity increased. Therefore, we concluded that this germline filter is effective in preventing healthy donors from being classified as bladder cancer patients while preserving sensitivity. However, more urine samples and paired PBMCs need to be analyzed to more profoundly test this germline filter, as discussed in the result section 3.5. In our cohort, we could not observe any subjects outside of the ranges, which was inferred from the absence of any detected mutations in healthy donors.

Comment 5-4

- Line 122-123 T-tests are used to compare or determine the differences between groups.

Authors’ response

We appreciate the Reviewer’s valuable suggestion. We corrected explanation of T-test as the reviewer commented, and changed indicated line, as follows:

             Before:

              [Line 132] Comparisons between two groups were performed using t-tests. ANOVA was used for comparisons of more than two groups.

             After:

[Line 132] T-test was used to test for differences between two groups, and ANOVA was used for more than two groups.

Comment 5-5

- Line 138-139 are there any differences in results for NMIBC and MIBC

Authors’ response

We appreciate the Reviewer’s valuable suggestion. There were differences in the frequency of mutated genes between NMIBC and MIBC, as shown in Figure 4b, although the number of samples from MIBC patients was relatively small compared to samples from NMIBC patients. We presented this result in the result section “3.4. Bladder Cancer Genotypes Determined Using uAL100” in our original manuscript. Also, the mutation counts and percentage of utDNA were higher in urines from MIBC patients than those from NMIBC patients, as shown in Figure 3b and d. However, in terms of sensitivity levels, NMIBC and MIBC were not significantly different (83.3% vs 85.7%). Since NMIBC and MIBC can be distinguished from cancer stages, we considered that presenting sensitivity levels according to tumor stage would sufficiently explain the sensitivity levels of NMIBC and MIBC. However, additional presentation of sensitivity levels in terms of histology would provide a broader perspective. Therefore, we added a relevant sentence in the result section “3.3. Bladder Cancer Detection without Tumor Samples”, as follows:

             Before:

[Line 199] In a subgroup analysis according to tumor stage, the sensitivity of detection in subgroups with stage Ta+Tis, T1, and T2–T4 disease was 71.4%, 90.9%, and 85.7%, respectively, with a specificity of 100% in each subgroup (Figure 3g).

After:

[Line 199] In a subgroup analysis according to tumor stage, the sensitivity of detection in subgroups with stage Ta+Tis, T1, and T2–T4 disease was 71.4%, 90.9%, and 85.7%, respectively, with a specificity of 100% in each subgroup (Figure 3g). When subgroups with stages that cor-respond to NMIBC were combined, the sensitivity of detection was 83.3%, which is similar to those of MIBC (T2-T4).

Comment 5-6

- Lines 156-157 specify the only one found mutation as this part gets distracting when reading.

Authors’ response

We appreciate the Reviewer’s valuable suggestion. We admitted that the indicated line could lead to confusion for readers, and we specified indicated mutation, as follows:

             Before:

[Line 172] Only one mutation was detected in the other patient, and that mutation was detected only in the urine DNA.

             After:

[Line 172] Only one mutation, TP53 M237I, was detected in the other patient, and that mutation was detected only in the urine DNA.

Comment 5-7

- Line 159-161 please rephrase for clarity.

Authors’ response

We appreciate the Reviewer’s valuable suggestion. We admitted that the indicated line can occur confusions for readers, and we changed those line clearly, as follows:

             Before:

              [Line 175] Among the mutations detected in the urine samples, those that were shared with matched tumor samples had higher VAFs than those that were not shared (median VAF = 17.18% vs. 2.06%; p = 0.025; Figure 2c).

After:

[Line 175] Within urine samples, the mutations that were shared with matched tumor samples had a higher median VAF than those that were not shared with matched tumor samples (median VAF = 17.18% vs. 2.06%; p = 0.025; Figure 2c).

Comment 5-8

- Line 163 comparison with 7 samples, ‘6 of 7’ positive is still a low statistical amount to state that these are truly positive. This is a considerable limitation that should be stated in the limitations.

Authors’ response

We appreciate the Reviewer’s valuable suggestion. We agreed with the Reviewer’s opinion and recognized that the low number of available paired samples is one of the limitations of our study when we wrote this manuscript. However, while we were focusing on the performance and limitations of uAL100 in terms of BC detection, we missed stating the limitation of our mutational concordance study. Therefore, we added a statement about the pointed limitation in the result section “3.2. Mutational Concordance in Paired Samples of Tumor and Urine DNA” after the indicated line, as follows:

             Before:

[Line 180] We concluded that most of the mutations detected in the urine samples by uAL100 were true positives, suggesting that uAL100 accurately detects utDNA in urine.

             After:

              [Line 180] We concluded that most of the mutations detected in the urine samples by uAL100 were true positives, suggesting that uAL100 accurately detects utDNA in urine. Nevertheless, to obtain statistical significance of high mutational concordance between paired samples additional sample pairs should be assayed, which is one of our future tasks.

and in discussion section as follows:

             Before:

              [Line 340] First, our cohort was small, and further validation of our method with larger samples is needed.

             After:

[Line 340] First, our cohort was small, and further validation of our method with larger samples is needed. Specifically, more tumor and urine sample pairs are need to be analyzed to obtain statistical significance of mutational concordance study, since we only analyzed 7 sample pairs, which is a small number to provide statistical significance to the results.

Comment 5-9

- Line 271 - Previous urine-based methods – citations.

Authors’ response

We appreciate the Reviewer’s valuable suggestion. We added citations that correspond to previous urine-based methods in indicated lines (reference #10, 11) as follows:

             Before:

[Line 300] Previous urine-based methods for BC detection showed significant levels of false-positive signals with similar or lower sensitivity compared with our method.

After

[Line 300] Previous urine-based methods using deep sequencing for BC detection showed significant levels of false-positive signals with similar or lower sensitivity compared with our method (Table 2) [10,11].

Additionally, we added a table that compared sensitivity and specificity of the methods (Table 2) after the discussion section as follows:

Comment 5-10

- Please revise limitations.

Authors’ response

We appreciate the Reviewer’s valuable suggestion. We revised the limitations of our method in the discussion section as follows:

             Before:

              [Line 339] We found that uAL100 performed well in detecting BC without tumor or blood samples; however, our study has several limitations. First, our cohort was small, and further validation of our method with larger samples is needed. Second, since the analysis pipe-line of uAL100 was originally developed for plasma samples, the error suppression may not be sufficient to remove all errors in urine DNA. Therefore, although the current version of error suppression in uAL100 is reliable, an error suppression method specifically for urine samples is needed to improve the accuracy of uAL100. Finally, since our target capture panel is not solely composed of genes related to BC, the cost of our method is not fully optimized for BC detection. Furthermore, some known BC-related genes, such as PLEKHS1, are not contained in the target capture panel. Consequently, it might be possible to in-crease the sensitivity for BC detection while reducing the error rate and cost by adjusting the gene panel.

             After:

[Line 339] We found that uAL100 performed well in detecting BC without tumor or blood samples; however, our study has several limitations. First, our cohort was small, and further validation of our method with larger samples is needed. Specifically, more tumor and urine sample pairs are need to be analyzed to obtain statistical significance of mutational concordance study, since we only analyzed 7 sample pairs, which is a small number to provide statistical significance to the results. Second, since the analysis pipeline of uAL100 was originally developed for plasma samples, the error suppression may not be sufficient to remove all errors in urine DNA. Therefore, although the current version of error suppression in uAL100 is reliable, an error suppression method specifically for urine samples is needed to improve the accuracy of uAL100. Third, since our target capture panel is not solely composed of genes related to BC, the cost of our method is not fully optimized for BC detection. Furthermore, some known BC-related genes, such as PLEKHS1, are not contained in the target capture panel. Consequently, it might be possible to in-crease the sensitivity for BC detection while reducing the error rate and cost by adjusting the gene panel. Finally, as we mentioned, surveillance of NMIBC patients is important due to their high recurrence rate. Therefore, we need to validate the surveillance ability of uAL100 to make our method more meaningful in real clinical situations.

Round 2

Reviewer 3 Report

I have reviewed the work re-submitted by Lee D. et al, titled: Accurate detection of urothelial bladder cancer using targeted deep sequencing of urine DNA. The authors have made important changes to their study and I consider these modifications appropriate for the paper to be published in its present form.

Reviewer 5 Report

Based on all reviews the article has changed considerably and is now better suitable for publication.